# CHARTGALAXY: A DATASET FOR INFOGRAPHIC CHART UNDERSTANDING AND GENERATION

**Zhen Li**[1,2][*] **Duan Li**[2][*] **Yukai Guo**[2][*] **Xinyuan Guo**[2][*] **Bowen Li**[2][*] **Lanxi Xiao**[2]
**Shenyu Qiao**[2], **Jiashu Chen**[2], **Zijian Wu**[2], **Hui Zhang**[2], **Xinhuan Shu**[3], **Shixia Liu**[2][†]
[1]Tianjin University [2]Tsinghua University [3]Newcastle University

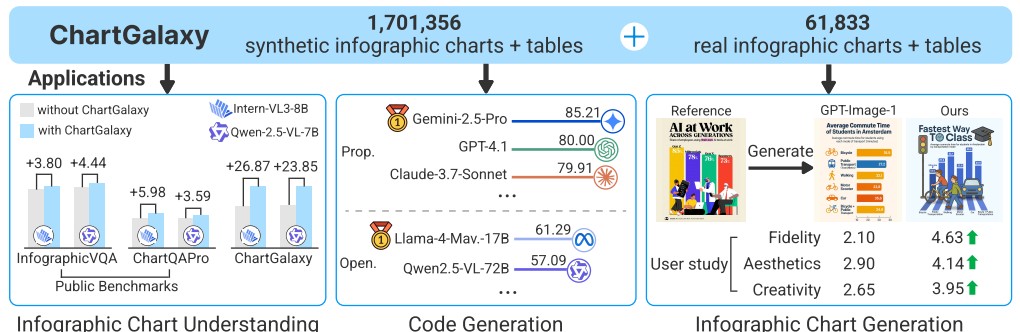

Figure 1: ChartGalaxy, a million-scale dataset of synthetic and real infographic charts with data tables, supporting applications in infographic chart understanding, code generation, and chart generation.

## ABSTRACT

Infographic charts are a powerful medium for communicating abstract data by combining visual elements (*e.g.*, charts, images) with textual information. However, their visual and structural richness poses challenges for large vision-language models (LVLMs), which are typically trained on plain charts. To bridge this gap, we introduce ChartGalaxy, a million-scale dataset designed to advance the understanding and generation of infographic charts. The dataset is constructed through an inductive process that identifies 75 chart types, 440 chart variations, and 68 layout templates from real infographic charts and uses them to create synthetic ones programmatically. We showcase the utility of this dataset through: 1) improving infographic chart understanding via fine-tuning, 2) benchmarking code generation for infographic charts, and 3) enabling example-based infographic chart generation. By capturing the visual and structural complexity of real design, ChartGalaxy provides a useful resource for enhancing multimodal reasoning and generation in LVLMs.

**Code:** https://github.com/ChartGalaxy/ChartGalaxy

**Data & Dataset Card:** https://huggingface.co/datasets/ChartGalaxy/ChartGalaxy

## 1 INTRODUCTION

Infographic charts are widely recognized as an effective form for communicating data and are commonly used in news media, business, and education (Cui et al., 2022; Elaldi & Çifçi, 2021). By integrating imagery (*e.g.*, icons and metaphorical graphics) alongside charts and textual information, they present abstract data in a manner that is both engaging and easy to understand, making data more accessible to a broad audience. Despite their effectiveness for human audiences, foundation models, such as GPT-4 (Achiam et al., 2023), Gemini (Team et al., 2023), and LLaVA (Liu et al., 2023c), face considerable challenges in automatically understanding infographic charts. The intricate interplay between visual and textual elements, diverse layout styles, and the need for cross-modal semantic reasoning pose significant difficulties. For example, Xie et al. (2025) found that MLLMs

---

[*]Equal contribution.

[†]Corresponding author.

perform worse in understanding infographic charts than plain charts constructed from the same data, largely because the imagery elements used in infographics introduce additional reasoning challenges. Moreover, automatically generating high-quality infographic charts remains an open challenge. While human designers can create visually diverse and semantically rich infographic charts, this process is time-consuming and requires expertise. Meanwhile, AI-generated charts often suffer from issues such as low data fidelity, modest visual quality, limited diversity, and a lack of coherence across modalities. This highlights the critical need for a comprehensive dataset of infographic charts that enables model development in both automatic understanding and generation. However, existing efforts focus on constructing datasets that are mostly limited to plain charts, failing to capture the diverse range of design styles and layouts that are key characteristics of infographic charts. This limits the ability of the trained models to generalize across different real-world applications where infographic charts are commonly used.

To address this limitation, we build ChartGalaxy, a million-scale dataset of high-quality real and synthetic infographic charts to facilitate automated understanding and generation. As shown in Fig. 2, we build ChartGalaxy in two steps: 1) collecting real infographic charts; 2) programmatically creating synthetic infographic charts. The real infographic charts are collected from 18 reputable chart-rich websites, such as *Visual Capitalist* and *Statista*. The synthetic infographic charts are created following an inductive structuring process (Schadewitz & Jachna, 2007). Specifically, we identify design patterns grounded in real infographic charts, including **75 chart types** (*e.g.*, bar charts), **440 chart variations** that reflect different visual element styles, and **68 layout templates** that define spatial relationships among elements. Based on these patterns, we then programmatically generate synthetic ones. The core of the generation is a human-in-the-loop pipeline that iteratively extracts and expands layout templates from real infographic charts using a detection model trained on synthetic infographic charts.

The final ChartGalaxy dataset includes 1,701,356 programmatically created infographic charts and 61,833 real infographic charts. It is characterized by two key features. First, the high-quality infographic designs and associated templates from these reputable websites ensure a rich diversity in design styles and structural complexity. Second, each infographic chart, whether real or synthetic, is paired with its source data table, enabling a clear mapping between data and its visual representation. Together, these make ChartGalaxy well-suited for training and evaluating LVLMs for automatic infographic understanding and generation. We demonstrate the utility of ChartGalaxy through three representative applications, each highlighting a distinct aspect of its value (Fig. 1). First, to evaluate and improve the ability of foundation models to understand infographic charts, we introduce a dataset for infographic chart understanding through the task of visual question answering (VQA). Second, to assess the capacity of models to generate executable representations of complex visual layouts, we present a benchmark for infographic chart code generation. Third, to explore the use of ChartGalaxy in creative content generation, we develop an example-based infographic chart generation method.

The main contributions of our work include:

- A pipeline for programmatically creating high-quality synthetic infographic charts based on the extracted layout templates from real designs.
- A comprehensive dataset comprising a large collection of representative and diverse real and synthetic infographic charts paired with tabular data.
- Three applications for showcasing the utility of our dataset in infographic chart understanding, code generation, and example-based infographic chart generation.

## 2 RELATED WORK

Early efforts in chart dataset construction primarily focus on building collections of **plain charts** to support chart understanding and generation (Hu et al., 2024; Yang et al., 2024a). These datasets can be further categorized into three types based on their sources: synthetic datasets, web-collected datasets, and mixed datasets. **Synthetic datasets** are programmatically generated, using tabular data drawn from probability distributions (Kafle et al., 2018; Kahou et al., 2017), collected from online data sources (Hu et al., 2024; Methani et al., 2020; Chaudhry et al., 2020; Xu et al., 2023; Tang et al., 2023; Akhtar et al., 2023), or simulated from large language models (Xu et al., 2023; Xia et al., 2024; Han et al., 2023; Zhao et al., 2025). While this method enables large-scale dataset

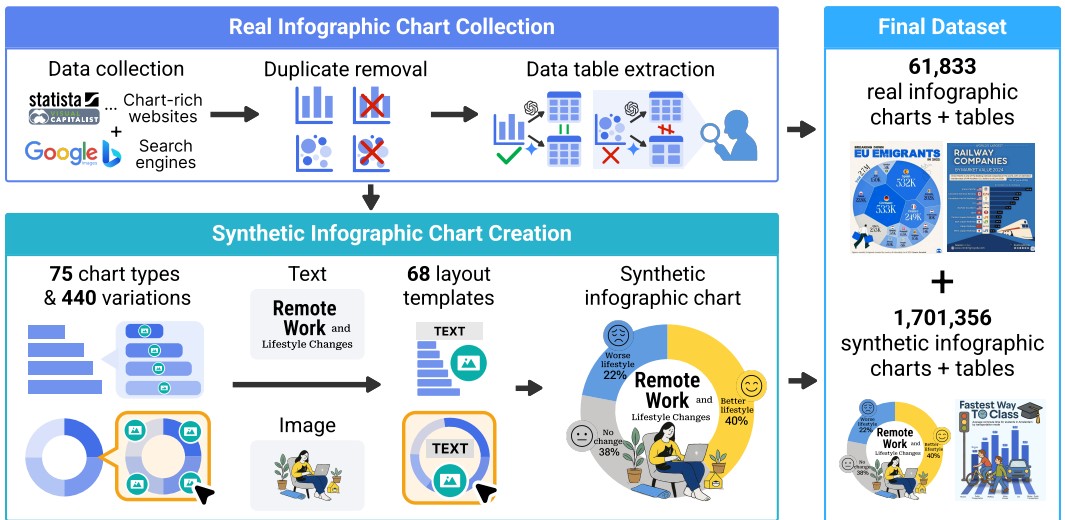

Figure 2: Overview of our dataset construction method.

construction, the controlled generation process often results in a limited diversity of chart types and visual styles, which reduces generalizability to more varied real-world scenarios. To improve diversity, **web-collected datasets** have been introduced, which collect charts from chart-sharing websites (Savva et al., 2011; Choi et al., 2019; Masry et al., 2022; Kantharaj et al., 2022a; Huang et al., 2025b; Obeid & Hoque, 2020; Kantharaj et al., 2022b; Masry et al., 2024a), charting library galleries such as Matplotlib (Wu et al., 2024; Yang et al., 2024b), and publications in academic repositories such as ArXiv (Wang et al., 2024; Hsu et al., 2021; Zhang et al., 2024). These datasets capture a broader variety of human-designed chart styles, but their overall size is often limited due to the time-consuming nature of manual verification and annotation. To overcome the scale limitations while preserving diversity, **mixed datasets** are then introduced. These datasets merge web-collected charts with synthetically generated ones in a two-step workflow: 1) collect real charts from the internet and 2) manually synthesize additional charts that follow the same real-world design patterns (Liu et al., 2023a;b; Masry et al., 2023; Meng et al., 2024; Yang et al., 2025). This hybrid strategy ensures real-world coverage while increasing dataset size.

In contrast, **infographic charts** remain underrepresented in the aforementioned datasets. This creates challenges in evaluating and developing LVLMs' capabilities on infographic charts (Masry et al., 2025). To bridge this gap, recent efforts have focused on building specific datasets for infographic charts. InfographicVQA makes an initial effort by searching "infographics" on the internet and scraping 5,485 infographics (Mathew et al., 2022). More recently, ChartQAPro provides a more challenging benchmark comprising 190 infographic charts, 258 dashboards, and 893 plain charts (Masry et al., 2025). However, these datasets are limited in scale. Moreover, synthesizing infographic charts is challenging, given the intricate interplay between visual and textual elements. To overcome these limitations, we develop an automatic infographic chart generation method that synthesizes infographic charts by leveraging the layout templates and chart variations extracted from real designs.

## 3 DATASET CONSTRUCTION METHOD

### 3.1 METHOD OVERVIEW

Fig. 2 presents an overview of our method, which includes two stages: **real infographic chart collection** and **synthetic infographic chart creation**.

The **real infographic charts** are collected from two main sources. First, we gather infographics from 18 well-known chart-rich websites that permit research use, such as *Visual Capitalist* and *Statista*. The full list of these websites and their licenses is provided in Appendix 5. Second, to further increase the diversity of infographic styles, we retrieve infographic charts via Google Images and Bing Images, following prior work (Masry et al., 2024a). We apply the platforms' built-in license

filters to retain only images available under Creative Commons licenses. The ethical consideration in the data collection process is further discussed in Ethics Statement. To ensure data quality, we remove duplicate images using Perceptual Hashing and CLIP similarity (Radford et al., 2021). Moreover, we extract per-chart tabular data using LVLMs in a multi-step, human-in-the-loop verification pipeline. This pipeline ensures accuracy and reduces model-induced noise. Details of this verification pipeline are discussed in Appendix B. This results in 61,833 real infographic charts with corresponding tables. The **synthetic infographic chart creation** stage follows an inductive structuring process that extracts design patterns, such as layout templates and chart variations, from real infographic charts and then uses these patterns to programmatically create high-quality synthetic charts. It includes three main steps: 1) identifying chart types and their variations, 2) extracting layout templates, and 3) creating synthetic infographic charts.

## 3.2 CHART TYPE AND VARIATION IDENTIFICATION

We first summarize 75 chart types observed in the collected real infographic charts, drawing on two existing taxonomies: *Data Viz Project* and *Datylon*. For each chart type, we extract chart variations featuring diverse visual styles, such as element shapes and icon placement. This results in 440 chart variations in total. The full lists of chart types and variations are provided in Appendix D.1 and D.2. We implement these chart types and variations using the expressive D3.js (Bostock et al., 2011), which supports visual features unavailable in libraries like Matplotlib or Seaborn.

## 3.3 LAYOUT TEMPLATE EXTRACTION

A layout template defines the spatial relationships among the text and visual elements in infographic charts. Example templates are illustrated in the bottom-left corner of each chart in Fig. 3, and a concrete example in JSON format is provided in Appendix F. We adopt a human-in-the-loop pipeline to initialize and expand these templates from real infographic charts.

**Initialization** Three co-authors manually annotate the bounding boxes of the text, images, and charts in 1,500 real infographic charts sampled from two high-quality sources: *Statista* (clean, minimalist designs) and *Visual Capitalist* (denser and more pictorial designs). From these annotations, we summarize an initial set of 55 layout templates that capture elements' relative positions (*e.g.*, title on the top-left, chart on the bottom-right) and pairwise overlaps (overlapping or not).

**Expansion** To ensure the coverage and diversity of templates, we build a detection model to analyze the unlabeled real infographic charts in ChartGalaxy and expand the layout template set. Using the initial templates, we programmatically create 120,000 synthetic infographic charts (Sec. 3.4), each with annotated bounding boxes. We then develop a detection model by fine-tuning InternImage (Wang et al., 2023) along with the DINO (Zhang et al., 2023) detector on these synthetic charts. We use this model to detect chart and image regions (Zhu et al., 2025a) and use PP-OCRv4 to extract text. We then compare the detected layouts with the existing templates using LTSim (Otani et al., 2024), a state-of-the-art method for measuring layout similarity. Layouts with low similarity scores are flagged as potential new templates. Next, we cluster these layouts using k-means ($k = 50$) and manually examine the cluster centroids to identify distinct layouts. This process yields 13 additional layout templates, expanding the set to 68 templates in total. The full list is provided in Appendix F.

## 3.4 TEMPLATE-BASED INFOGRAPHIC CHART CREATION

The creation process involves three steps: 1) curating data tables; 2) generating/recommending elements based on the data table; and 3) optimizing the layout based on the selected template. Fig. 3 shows six examples of the synthetic charts. More examples are provided in Appendix G.

**Tabular data curation** To enhance data diversity for chart generation, we build a rich repository of real and synthetic tabular data. For real data, we collect 200,085 tables from well-established sources, including VizNet (Hu et al., 2019), UN data, Our World in Data, and Papers with Code. For synthetic data, we generate 98,483 tables with Gemini-2.0-Flash following Han et al. (2023). To facilitate downstream processing, we also complement each table with a topic (*e.g.*, "NBA play-offs") extracted by Gemini-2.0-Flash and several data facts (*e.g.*, trends, comparisons) following Wang et al. (2020).

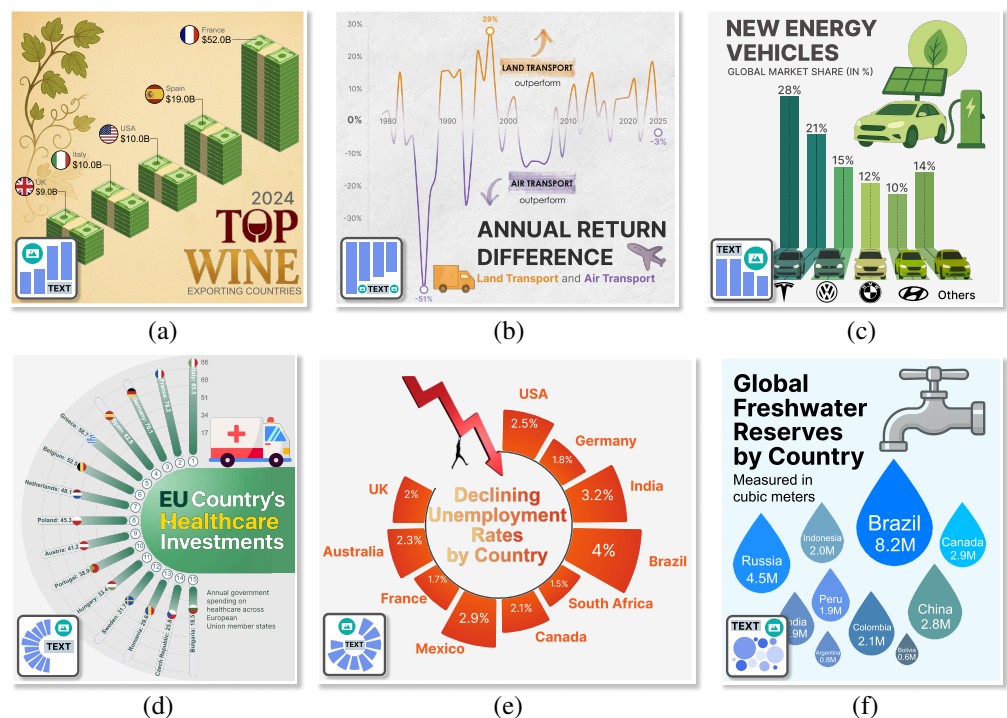

Figure 3: Examples of synthetic infographic charts in ChartGalaxy. The bottom-left illustration on each infographic chart shows the corresponding layout template.

**Element generation/recommendation**    For each data table, we generate/recommend key elements of the infographic chart, including 1) text, 2) image, and 3) chart.

*Text.* The title and subtitle are generated using a retrieval-augmented prompting strategy. Specifically, we use Sentence-BERT (Reimers & Gurevych, 2019) to retrieve the three most relevant real infographic charts according to the data topic and data fact. Using them as in-context references, we prompt Gemini-2.0-Flash to generate the title and subtitle aligned with the data table.

*Image.* To recommend images for infographic charts, we build a repository and retrieve semantically relevant images from it using the associated data tables. Images are collected from publicly available icon resources such as Icon645 (Lu et al., 2021) and Noun Project. We apply heuristic filters to remove low-quality images, excluding those with low ink ratios, poor resolution, or extreme aspect ratios. Using Gemini-2.0-Flash, we then generate descriptive keywords and captions for each image that depict its content and visual style. Images that are overly literal, visually cluttered, or lack symbolic clarity are discarded. This process results in a curated repository of 681,459 images. To retrieve relevant images for each infographic chart, we compute the similarity between the image keywords and the generated chart title using Sentence-BERT embeddings (Reimers & Gurevych, 2019).

*Chart.* Chart generation proceeds in two steps: selecting a suitable chart type and rendering with a specific chart variation. First, based on the attributes (*e.g.*, categorical, numerical) and characteristics (*e.g.*, scales) of the given data table, we identify candidate chart types following the predefined data-to-chart mapping rules. For example, a data table with one categorical column and two numerical columns may be mapped to a scatter plot. When multiple chart types are suitable, we prompt Gemini-2.0-Flash to select the optimal one, considering data types, value distributions, and temporal patterns (Luo et al., 2018). A full list of mapping rules and prompts is provided in Appendix E. Next, we choose a specific variation under the selected chart type using adaptive sampling (Goodfellow et al., 2016) that favors underrepresented variations to maintain distributional balance. For variations requiring additional images (*e.g.*, Fig. 3(a)), we retrieve the relevant ones from our curated image repository. Finally, we apply semantically resonant color palettes for chart rendering based on the generated chart titles and subtitles. These palettes are extracted from the collected real infographic charts (Liu et al., 2024). If a selected palette contains fewer colors than required, we supplement additional harmonic and discriminable colors (Chen et al., 2025).

**Layout optimization** Previous research has shown that a compact layout with appropriate white space enhances both visual appeal and data clarity (Coursaris & Kripintris, 2012). Consequently, we aim to select the template with the highest ink ratios while preserving readability. To this end, we first filter out templates that are incompatible with the generated elements (*e.g.*, unintended overlaps between elements). For each remaining template $t$, we treat its spatial relationships as hard constraints during optimization. We begin by initializing element positions through rejection sampling, repeatedly drawing random candidate layouts and retaining only those that satisfy the spatial relationships specified by the template. This provides a structurally coherent starting point. Next, we refine the layout by adjusting element positions and sizes to reduce unnecessary white space, while preserving all template-specified constraints. We measure white space using the ink ratio and select the optimized result $\mathcal{E}$ with the highest ink ratio as the final layout.

$$\max_{t, \mathcal{E}} |\cup_i e_i| \, / \, |f(\cup_i e_i)|, \qquad \text{s.t.} \quad g(\mathcal{E}, t) = 1; \quad d(\partial e_i, \partial e_j) \geq p, \, \forall \, i \neq j. \tag{1}$$

Here, $e_i$ is the pixel set of an element, $f(e)$ denotes the pixel set within the tight-fitting bounding box of $e$, $\mathcal{E}$ is the set of elements, and $g(\mathcal{E}, t)$ is the indicator function that equals 1 if $\mathcal{E}$ satisfies the spatial relationships specified by the template $t$. We enforce that a minimum pairwise distance $d$ between element contours $\partial e_i$ and $\partial e_j$ is larger than a given threshold $p$. This layout optimization is formulated as a constrained packing problem and solved by grid search (Lodi et al., 1999).

## 3.5 DATA STATISTICS

ChartGalaxy contains 1,701,356 programmatically generated infographic charts and 61,833 real ones, covering 75 chart types, 440 chart variations, and 68 layout templates. Each infographic chart in the dataset is associated with tabular data, providing rich supervision for training and evaluation tasks. Among the 75 chart types, the most frequently occurring ones are horizontal bar charts (11.7%), vertical bar charts (4.9%), and scatterplots (3.5%). Each chart type has up to 27 variations, capturing a wide range of styles in element shapes, icon placement, and rendering effects. We observe differences in the frequency of template usage. For example, the template in Fig. 3(f) is the most common in the synthetic charts (6.9%). Differences in template usage are influenced by each template's structural flexibility, compatibility with chart types, and prevalence in real-world use cases. For more analysis on our dataset, please refer to Appendix C.

## 4 EXPERIMENTS

### 4.1 INSTRUCTION DATASET FOR INFOGRAPHIC CHART UNDERSTANDING

In this experiment, we construct an instruction dataset with ChartGalaxy to enhance model capabilities on infographic chart understanding. We validate its usefulness by fine-tuning two open-source LVLMs, demonstrating improvements on both public benchmarks and our independent evaluation set.

**Dataset construction** To improve LVLMs' data comprehension and visual understanding of info-graphic charts, we construct an instruction dataset comprising 443,455 question-answer pairs based on 70,248 charts sampled from ChartGalaxy, ensuring balanced coverage of diverse chart types for better fine-tuning performance. The questions are classified into three types: 1) **Text-based reasoning**. We incorporate well-established question types from prior work, including open-form questions (Masry et al., 2025) and template-based questions (Meng et al., 2024). These questions cover data identification (DI), data comparison (DC), data extraction with condition (DEC), and fact checking (FC). 2) **Visual-element-based reasoning**. We extend beyond purely text-based reasoning questions by incorporating visual elements from charts, such as icons (*e.g.*, "What was the wine export value of 🇫🇷 in 2024?"). These questions require models to associate visual elements with their corresponding data values, thus testing their ability to conduct more complex cross-modal reasoning (Xie et al., 2025). 3) **Visual understanding**. This type includes style detection (SD), visual encoding analysis (VEA, *e.g.*, "What data dimension is encoded using different colors in this infographic chart?"), and chart classification (CC). The three types of questions evaluate a model's ability to interpret visual elements and underlying structural representations of the data. Detailed prompts and methods for generating question-answer pairs are provided in Appendix I.1. Moreover, we construct an independent, human-verified evaluation set containing 2,176 synthetic charts with

4,975 question-answer pairs. We focus on synthetic charts here, as they provide bounding-box annotations that enable visual-element-based reasoning questions.

**Experimental setup** We fine-tune two representative open-source LVLMs, InternVL3-8B (Zhu et al., 2025b) and Qwen2.5-VL-7B (Bai et al., 2025). Training details are reported in Appendix H.1. Our evaluation benchmark consists of two parts: 1) public benchmarks including Infograph-icVQA (Mathew et al., 2022), which focuses on general infographics with only a subset being infographic charts, and ChartQAPro (Masry et al., 2025), which covers various chart types; and 2) the aforementioned independent evaluation set of 2,176 charts with 4,975 question-answer pairs specifically targeting infographic charts. For the evaluation metrics, we follow previous work on chart question answering (Masry et al., 2025), using relaxed accuracy with a 5% margin for numerical answers, ANLS for textual answers, and exact matching for multiple-choice questions.

**Results and analysis** Tables 1 and 2 show the evaluation results on the public benchmarks and our evaluation set. After fine-tuning with ChartGalaxy, both models demonstrate improved performance gains across all question types. On the public benchmarks (Table 1), InternVL3 improves performance by 3.80% on InfographicVQA and 5.98% on ChartQAPro, while Qwen2.5-VL shows a 4.44% gain on InfographicVQA and a 3.59% improvement on ChartQAPro. On our evaluation set (Table 2), both models show consistent

Table 1: Performance on public benchmarks w/ and w/o ChartGalaxy.

| Model | InfographicVQA | ChartQAPro |
|---|---|---|
| InternVL3-8B | 76.19 | 38.15 |
| + ChartGalaxy | 79.99 | **44.13** |
| (+) | (+3.80) | (+5.98) |
| Qwen2.5-VL-7B | 78.59 | 37.97 |
| + ChartGalaxy | **83.03** | 41.56 |
| (+) | (+4.44) | (+3.59) |

improvements across all question types, with overall gains of +26.87% for InternVL3 and +23.85% for Qwen2.5-VL. The most notable improvements are observed in the visual understanding questions, with increases of up to +60.49% for style detection and +40.78% for visual encoding analysis. These results indicate that existing pre-training routines may underrepresent questions involving chart visual styles and data encoding, an area our dataset helps to supplement. Performance also improves across the text-based and visual-element-based reasoning questions. Qwen2.5-VL performs well on text-based reasoning, while InternVL3 shows relatively stronger gains on visual-element-based reasoning. Additional results, including ablation studies on real/synthetic charts and evaluation on a smaller model (Qwen2.5-VL-3B), are provided in Appendix H.1.

Table 2: Performance on our independent evaluation set w/ and w/o ChartGalaxy.

| Model | Text-Based Reasoning | | | | Visual-Element-Based Reasoning | | | | Visual Understanding | | | Overall |
|---|---|---|---|---|---|---|---|---|---|---|---|---|
| | DI | DC | DEC | FC | DI | DC | DEC | FC | SD | VEA | CC | |
| InternVL3-8B | 85.36 | 55.24 | 51.66 | 75.80 | 33.32 | 18.91 | 37.62 | 61.58 | 30.56 | 50.57 | 73.03 | 53.20 |
| + ChartGalaxy | 91.67 | 74.39 | 75.14 | **89.26** | **69.12** | **42.79** | 58.57 | **80.23** | **91.05** | **91.35** | **99.39** | 80.07 |
| (+) | (+6.31) | (+19.15) | (+23.48) | (+13.46) | (+35.80) | (+23.88) | (+20.95) | (+18.65) | (+60.49) | (+40.78) | (+26.36) | (+26.87) |
| Qwen2.5-VL-7B | 87.45 | 66.32 | 64.44 | 78.53 | 40.76 | 30.65 | 46.00 | 53.95 | 28.70 | 50.08 | 70.91 | 56.50 |
| + ChartGalaxy | **93.28** | **80.98** | **86.31** | 87.34 | 66.15 | 39.80 | **72.38** | 79.38 | 87.65 | 90.86 | 98.18 | **80.35** |
| (+) | (+5.83) | (+14.66) | (+21.87) | (+8.81) | (+25.39) | (+9.15) | (+26.38) | (+25.43) | (+58.95) | (+40.78) | (+27.27) | (+23.85) |

## 4.2 BENCHMARKING INFOGRAPHIC CHART CODE GENERATION

This experiment presents a benchmark to assess LVLMs' code generation for infographic charts.

**Benchmark construction** The benchmark is designed to evaluate the Direct Mimic task (Yang et al., 2025), where an LVLM is prompted to generate the D3.js code for a given infographic chart image. Due to variation in coding styles and implementation strategies (Si et al., 2025), directly comparing code quality is challenging. Therefore, we evaluate the rendered output instead of the code itself. Specifically, we render the output as both an SVG and a PNG: the SVG enables fine-grained analysis, as it contains precise information about visual and textual elements (*e.g.*, positions, colors), while the PNG supports direct visual comparison. To support this task, we randomly sampled 500 synthetic infographic charts, with explicit coverage of all chart types, variations, and layout templates. Each chart is paired with a ground-truth triplet: a PNG image, an SVG, and the corresponding tabular data. Benchmark details are provided in Appendix H.2.

Following previous benchmarks (Si et al., 2025; Yang et al., 2025), we measure the similarity between the ground-truth chart and the one rendered by the generated code at two levels: a **high-level score** (overall visual similarity judged by GPT-4o with the PNG images) and a **low-level score** (average similarity across fine-grained SVG elements). To compute the low-level score, we parse the SVG elements from the rendered chart and the ground-truth one and match them based on attributes such

Table 3: Performance comparison of 17 LVLMs on our proposed code generation benchmark, reporting the code execution success rate (Exec. Rate), low-level, high-level, and overall scores.

| Model | Exec. Rate | Low-Level | | | | | | | High-Level | Overall |
|---|---|---|---|---|---|---|---|---|---|---|
| | | Area | Text | Image | Color | Position | Size | Avg. | GPT-4o | |
| *Proprietary* | | | | | | | | | | |
| Gemini-2.5-Pro | **100.00** | **90.72** | **95.69** | 86.37 | **87.67** | **89.23** | 69.05 | **86.45** | **83.97** | **85.21** |
| GPT-4.1 | **100.00** | 90.58 | 91.58 | **86.53** | 87.52 | 87.13 | 55.61 | 83.16 | 76.84 | 80.00 |
| Claude-3.7-Sonnet | **100.00** | 88.96 | 92.39 | 77.90 | 84.78 | 87.57 | 67.29 | 83.15 | 76.66 | 79.91 |
| GPT-4.1-mini | 99.60 | 88.21 | 88.31 | 79.32 | 86.43 | 85.61 | 62.85 | 81.79 | 77.59 | 79.69 |
| OpenAI-o4-mini | 98.80 | 83.13 | 79.26 | 67.53 | 83.93 | 84.95 | 64.07 | 77.14 | 74.79 | 75.97 |
| Gemini-2.5-Flash | 96.40 | 84.52 | 87.02 | 73.21 | 77.81 | 83.01 | 62.29 | 77.98 | 73.12 | 75.55 |
| OpenAI-o1 | 97.20 | 85.01 | 78.07 | 80.28 | 81.70 | 82.35 | 60.76 | 78.03 | 71.35 | 74.69 |
| OpenAI-o3 | 92.40 | 83.84 | 82.43 | 72.15 | 78.79 | 81.63 | 63.44 | 77.05 | 71.38 | 74.22 |
| GPT-4o | 99.00 | 74.86 | 63.54 | 34.12 | 76.60 | 80.12 | 56.27 | 64.25 | 67.10 | 65.67 |
| GPT-4.1-nano | **100.00** | 74.97 | 59.94 | 36.06 | 69.26 | 75.10 | 47.69 | 60.50 | 59.62 | 60.06 |
| Doubao-1.5-Vision-Pro | 97.20 | 59.05 | 48.08 | 36.63 | 60.09 | 66.02 | 40.00 | 51.65 | 42.58 | 47.11 |
| Moonshot-v1-Vision | 95.20 | 60.86 | 45.54 | 33.68 | 60.86 | 63.33 | 39.81 | 50.68 | 38.11 | 44.39 |
| *Open-Source* | | | | | | | | | | |
| Llama-4-Maverick-17B | **99.60** | **76.59** | 56.37 | 59.24 | **69.59** | 75.39 | 49.87 | **64.51** | **58.06** | **61.29** |
| Qwen2.5-VL-72B | 92.60 | 73.50 | **63.22** | 49.33 | 65.06 | 73.14 | 47.53 | 61.96 | 52.21 | 57.09 |
| InternVL3-78B | 95.60 | 73.36 | 47.98 | **62.19** | 65.76 | 70.35 | 43.74 | 60.56 | 49.58 | 55.07 |
| Llama-4-Scout-17B | 98.20 | 71.08 | 51.81 | 45.27 | 63.48 | 69.95 | 42.28 | 57.31 | 46.50 | 51.91 |
| Qwen2.5-VL-32B | 81.60 | 60.37 | 49.37 | 28.19 | 54.37 | 61.37 | 37.60 | 48.55 | 44.42 | 46.48 |

as tag types and positions. This matching is formulated as a linear assignment problem and solved using the Jonker-Volgenant algorithm (Crouse, 2016; Si et al., 2025; Chen et al., 2024). Based on the matching results, we compute a low-level score by averaging six metrics: area, text, image, color, position, and size. The area metric captures the ratio of matched element area to the total element area. The text and image metrics assess the similarity of generated text and image elements, respectively. The color, position, and size metrics evaluate visual consistency in these attributes among matched elements. Details of the evaluation metrics are provided in Appendix H.2. Following Yang et al. (2025), we also calculate the **overall** score as the average of the high-level and low-level scores, ranging from 0 to 100. Notably, if the code fails to render the chart, both scores are set to 0.

**Experimental setup**  We benchmark 17 widely used LVLMs, including 12 proprietary ones and 5 open-source ones, as shown in Table 3. Model configurations and detailed prompts are provided in Appendix H.2 and I.2, respectively.

**Results and analysis**  Table 3 presents the results of the 17 LVLMs on our benchmark. We highlight key findings below. First, among proprietary models, Gemini-2.5-Pro achieves the highest overall score of 85.21. Among open-source models, Llama-4-Maverick-17B performs the best with a score of 61.29, outperforming the proprietary GPT-4.1-nano (60.06). However, it still lags behind top-performing proprietary models. Within the GPT-4.1 series, GPT-4.1 and GPT-4.1-mini achieve nearly identical overall scores (80.00 vs. 79.69). However, GPT-4.1 consistently outperforms GPT-4.1-mini across all individual low-level metrics, except the size metric, where it scores notably lower (55.61 vs. 62.85). This drop suggests that GPT-4.1 may struggle to preserve element size accurately, highlighting the need for further investigation. Additional results and analysis are provided in Appendix H.2.

### 4.3 EXAMPLE-BASED INFOGRAPHIC CHART GENERATION

This experiment demonstrates how ChartGalaxy can be used to support the generation of infographic charts through layout and style adaptation.

**Method**  We develop an example-based method that transforms user-provided tabular data into an infographic chart, aligning with the layout and visual style of a given example infographic chart. This example is either provided by the user or automatically retrieved from the ChartGalaxy dataset by selecting the chart most relevant to the user-provided tabular data and its column descriptions. The key feature of this method is its ability to generate visually coherent infographic charts by reusing the layout of well-designed examples and leveraging powerful detection and vision-language models. To enable this capability, we first use the detection model described in Sec. 3.3 to detect key

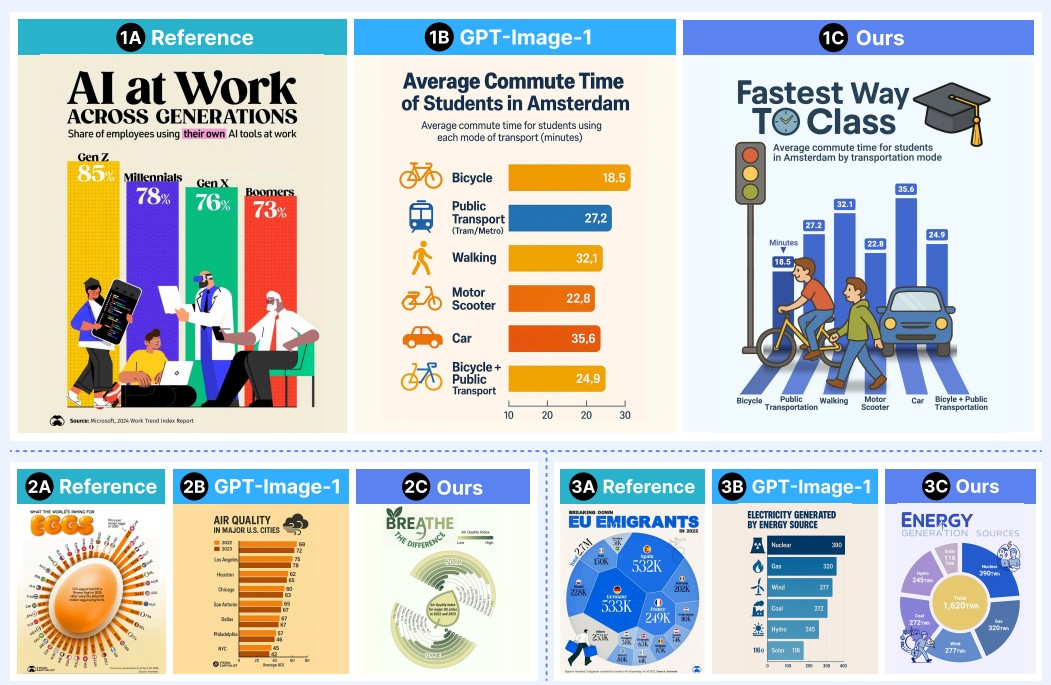

Figure 4: Three examples of infographic charts used in Sec. 4.3. In each example, A is the reference chart, B and C are generated by GPT-Image-1 and our method, respectively, using the same data.

Table 4: Performance comparison between our method and GPT-Image-1 (Mean, [95% CI]).

| Method | Fidelity | Aesthetics | Creativity |
|---|---|---|---|
| Ours | 4.63, [4.51, 4.75] | 4.14, [3.95, 4.33] | 3.95, [3.77, 4.13] |
| GPT-Image-1 | 2.10, [1.71, 2.50] | 2.90, [2.48, 3.33] | 2.65, [2.28, 3.03] |

elements (*e.g.*, icons, text blocks) in the example infographic chart. We then generate/recommend new elements from the provided tabular data (Sec. 3.4), initialize their layout using the extracted positional information from the example, and refine the arrangement through the layout optimization module (Sec. 3.4).

**User study setup**   We conducted a user study with 16 experts in design or visualization to evaluate the quality of the generated infographic charts using our method and GPT-Image-1, a state-of-the-art image generation model. The evaluation focused on three key metrics: **fidelity** (how accurately the data is represented), **aesthetics** (how appealing the infographic chart is), and **creativity** (how innovative the design is). Experts were asked to rate 30 pairs of infographic charts generated by our method and GPT-Image-1 based on the same tabular data and reference infographic chart, using a five-point Likert scale (1=poor, 5=excellent) for each metric. Fig. 4 shows examples of the generated infographic charts and the reference. The prompt used for GPT-Image-1 is provided in Appendix I.3.

**Results and analysis**   Table 4 shows that our method outperforms GPT-Image-1 across all three metrics based on the Wilcoxon signed-rank test on the user rating data ($p < 0.01$): **fidelity** (average: 4.63 vs. 2.10), **aesthetics** (4.14 vs. 2.90), and **creativity** (3.95 vs. 2.65). Particularly, our method achieves high fidelity by accurately representing data through carefully implemented chart variations. Some scores are slightly lower due to expert preferences for alternative chart types in certain cases. In contrast, GPT-Image-1 often exhibits serious fidelity-related issues, such as incorrect labels, disproportionate representations, and mismatched data elements. In terms of aesthetics and creativity, our method benefits from accurately extracting layout templates from reference infographic charts and supporting a wider variety of chart types beyond basic chart types, such as bar charts and line charts. By comparison, GPT-Image-1 tends to use basic chart types with limited variations, leading to outputs that are visually simple and monotonous. The detailed analysis is provided in Appendix H.3.

## 5 CONCLUSION

We echo the growing interest in multimodal understanding and generation in LVLMs by introducing ChartGalaxy, a million-scale, high-quality dataset of 61,833 real and 1,701,356 synthetic infographic charts. Grounded in real designs, our structured synthesis pipeline enables the scalable creation of diverse infographic charts. By providing aligned data-chart pairs, extracted layout templates, and three representative applications, we aim to advance the development of foundation models capable of interpreting, reasoning, and generating complex infographic charts.

At the same time, we acknowledge that the current ChartGalaxy dataset primarily focuses on single-chart infographics, limiting its ability to capture the complexity of multi-chart narratives. As a result, future work should explore the generation and analysis of multi-chart infographics, which emphasize storytelling through coordinated visual elements. Additionally, enriching the interplay between text and visuals could further enhance models' capacity for nuanced multimodal understanding.

## ACKNOWLEDGMENTS

This work was supported by National NSF of China (No. U21A20469). The authors would like to thank J. Zhu, T. Xie, M. Lin, Z. Wang, and Z. Shen for their contributions to the dataset construction.

## ETHICS STATEMENT

We strictly comply with the terms of use of the original sources and standard research ethics in dataset construction and evaluation. **1. Real infographic charts**. We collect infographic charts from publicly accessible sources with identifiable copyright/licensing terms: i) 18 chart-rich websites (*e.g.*, Visual Capitalist, Statista), each of which provides clear copyright or license terms (see Table 5 for the full source list and licensing links); ii) Google Images and Bing Images, where we apply built-in license filters to retain only charts explicitly marked under Creative Commons licenses. To further reduce copyright risks, we release only the URLs of real charts, following prior work (Masry et al., 2024a; Schuhmann et al., 2022). Prior to releasing our dataset, we consulted a licensed IP attorney, who reviewed the licensing pages of all 18 websites (accessed on 2025-09-16 and re-checked on 2025-12-02) as well as our dataset release format. The legal assessment is that our release does not violate the rights of these websites, because it does not republish, redistribute, or reproduce third-party charts and we do not host or distribute any chart images, including thumbnails or cached copies. Any downstream access must comply with each source's terms. This legal assessment informed the design of our dataset and release protocol. We provide additional details in Appendix A. **2. Synthetic infographic charts**. The synthetic infographic charts are generated using resources that are permitted for academic use, including tabular data from openly licensed repositories: UN data (Public domain), Our World in Data (CC BY), VizNet (Hu et al., 2019) (CC BY-NC-SA), and Papers with Code (CC BY-SA); as well as images from publicly available resources with explicit reuse permissions: Icon645 (Lu et al., 2021) (CC BY-NC-SA), Flaticon (Royalty-Free License), Iconshock (Royalty-Free License), Heroicons (MIT License), and Google Material Icons (Apache License 2.0). **3. Sensitive content**. To mitigate the risk of exposing sensitive and personal content, we ensure that all real infographic charts, tabular data, and image resources are sourced exclusively from reputable public platforms or collected via search engines with built-in safe-search functionality, both of which generally employ safeguards. We further applied Gemini-2.0-Flash filtering to exclude sensitive topics such as religious conflicts and individually identifiable medical records. **4. Human subjects**. The user study (Sec. 4.3) was approved by the Institutional Review Board of the first author's university. Each participant was compensated with 30 USD for their participation. The study did not involve exposure to emotionally charged, political, or misleading content. Participants were shown infographic charts on neutral topics (*e.g.*, bird population growth, energy sources) and asked to evaluate their quality. No sensitive or personally identifiable data was collected during the study. Participants were fully informed of their rights and were free to withdraw at any time.

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

# A    DATA SOURCES AND USE POLICY DETAILS

Table 5 lists the 18 chart-rich websites in our corpus together with links to their publicly available terms/licensing pages (accessed on 2025-09-16 and re-checked on 2025-12-02). As stated in the Ethics Statement, ChartGalaxy is released in a URL-only format and does not include, host, cache, or redistribute any chart/image files (including thumbnails or cached copies).

Table 5: Chart-rich websites and their licenses.

| Name | URL | License |
|------|-----|---------|
| Statista | statista.com | CC BY-ND 4.0 |
| Visual Capitalist | visualcapitalist.com | Customized |
| World Statistics | world-statistics.org | CC BY SA-3.0 |
| Our World in Data | ourworldindata.org | CC BY |
| OECD | oecd.org | CC BY 4.0 |
| Openverse | openverse.org | CC |
| The Conversation | theconversation.com | CC BY-ND |
| Kaiser Family Foundation | kff.org | CC BY-NC-ND 4.0 |
| World Economic Forum | weforum.org | CC BY-NC-ND 4.0 |
| Office for National Statistics | ons.gov.uk | OGL |
| Information is Beautiful | informationisbeautiful.net | Customized |
| Pew Research Center | pewresearch.org | Customized |
| MarketingCharts | marketingcharts.com | Customized |
| Chit Chart | chitchart.com | Customized |
| Wikimedia | wikimediafoundation.org | Customized |
| Hikaku Sitatter | hikaku-sitatter.com | Customized |
| Eurostat | ec.europa.eu | Customized |
| European Parliament | europarl.europa.eu | Customized |

**Attorney review and scope**    Prior to releasing the dataset, we consulted a licensed Intellectual Property (IP) attorney. The attorney reviewed (i) the terms and licensing/usage pages of all 18 websites, and (ii) our dataset release format. The attorney's assessment is that, under this URL-only release format, our dataset **does not constitute republishing, redistribution, or reproduction of the underlying charts**, and therefore does not violate the reviewed website terms. Users of this dataset are responsible for ensuring that any downstream access via these URLs complies with each source's terms.

**Source-specific conditions**    Based on the attorney's review of the 18 sources, the relevant conditions can be summarized as follows:

- **Attribution and copyright notice requirements** (6 sources). *Chit Chart* requires that any shared use clearly indicate its copyright ownership and retain the required trademark/copyright notice. *MarketingCharts* requires attribution to the source website and acknowledgment of the original data source(s) associated with the chart. *Eurostat* authorizes reuse provided that the source is acknowledged. *European Parliament* permits further dissemination provided that the entire item is reproduced and the source is acknowledged. *Pew Research Center* requires that copies display the full copyright notice. *Kaiser Family Foundation* permits citation or reprinting of specific charts, and notes that its original content remains copyrighted.

- **Use limited to non-commercial or personal purposes** (3 sources). *Hikaku Sitatter* permits use only for not-for-profit purposes. *Visual Capitalist* permits use only for personal, non-commercial purposes and prohibits resale or redistribution. *World Economic Forum* permits sharing or redistribution only for non-commercial purposes with proper attribution and prohibits distributing modified versions.

- **No additional source-specific constraints** (9 sources). For the remaining 9 sources listed in Table 5, the posted terms reviewed by the IP attorney do not specify additional source-specific constraints that would restrict our URL-only release format beyond standard attribution and copyright notice requirements.

## B    PIPELINE FOR DATA TABLE EXTRACTION

For real infographic charts, we extract a per-chart data table using LVLMs in a multi-step verification pipeline, where humans are involved in resolving inconsistent tabular results. Specifically, we first run Gemini-2.0-Flash and GPT-4o-mini in parallel and retain tables where both models produce consistent outputs. Two tables are considered consistent if they have 1) the same number of data points, 2) identical column names and categorical values, and 3) numerical values differing by no more than 5%. The threshold of 5% is set by following the common practices (Methani et al., 2020; Masry et al., 2022; 2025). If the two tables are consistent but not exactly matched, we randomly select one for use. We also reviewed 200 randomly selected exact matches and found that all of them were correct. In contrast, if the two tables are inconsistent, we perform an additional verification step where GPT-4o also extracts a table. If GPT-4o's output agrees with either of the models, we accept it. Otherwise, trained annotators from the service provider verify and correct the table. The success rates of this pipeline are:

- 43.6% of charts are accepted directly when GPT-4o-mini and Gemini-2.0-Flash agree.

- Among the remaining 56.4%, GPT-4o resolves 76.2% of the inconsistencies, resulting in 86.6% of charts being automatically processed by LVLMs.

- The remaining 13.4% are manually verified and corrected.

This pipeline greatly reduces the need for manual annotation while ensuring reliable outputs.

## C    DATASET DIVERSITY

**Geographic diversity**    We evaluated the geographic diversity of the whole dataset using Gemini-2.5-Flash to extract country and region references based on the UN geoscheme. Infographic charts with geographic information account for 59.4% of the total, with the most frequently referenced regions being North America (33.0%), Europe (23.1%), Asia (19.3%), and Africa (10.6%). While the dataset spans a broad range of regions, there is a moderate skew toward North America and Europe, reflecting the distribution of original sources (e.g., Visual Capitalist, Statista).

**Domain diversity**    Likewise, we analyzed domain diversity in our dataset by prompting Gemini-2.5-Flash to assign each chart a topic based on the IPTC Media Topics taxonomy, a widely used hierarchical ontology for classifying news content. The result shows that the infographic charts in ChartGalaxy span 16 out of 17 top-level categories (excluding a sensitive category on religion conflict) and 99 out of 120 topics defined in the taxonomy. The most frequently occurring topics include public health (7.3%), business enterprise (6.0%), and environmental conservation (3.9%). These findings indicate that our dataset captures a wide and balanced distribution of real-world domains.

**Linguistic diversity**    Following ChartQAPro (Masry et al., 2025), we measured linguistic diversity using the average pairwise cosine distance between Sentence-BERT embeddings extracted by the all-MiniLM-L6-v2 model (Reimers & Gurevych, 2019). Our dataset achieves a linguistic diversity score of 0.8754, higher than ChartQAPro (Masry et al., 2025) (0.8439) and Chartxiv (Wang et al., 2024) (0.7831). This result indicates that ChartGalaxy exhibits greater linguistic diversity than existing datasets. The real infographic charts predominantly use English, with additional content in languages such as Spanish, German, and French, reflecting the linguistic diversity of online data. All synthetic data in our dataset is in English, which can be easily translated into other languages upon request, offering flexibility to meet users' diverse linguistic needs.

**Representativeness of synthetic infographic charts**    To evaluate the distributional alignment of our synthetic infographic charts with real infographic charts, we conducted an additional quantitative analysis. Specifically, we extracted DreamSim (Fu et al., 2023) features (dimension = 1792) from both real and synthetic infographics, applied UMAP for projection with $n\_neighbors = 100$, and measured grid-based coverage over a 20×20 spatial grid. We define coverage as the percentage of grid cells occupied by real infographic charts that are also covered by synthetic ones. The result shows that our synthetic charts cover 97.62% of the feature space occupied by real infographic charts, demonstrating that our synthetic infographics are highly representative of real infographics.

## D    CHART TYPES AND VARIATIONS

We provide a full list of 75 chart types and 440 chart variations.

### D.1    CHART TYPES

We summarize chart types observed in the collected real infographic charts, drawing on two existing taxonomies: Data Viz Project and Datylon. The diversity of chart types ensures that our synthetic infographic charts can adapt to a wider range of scenarios and data representations, making them valuable for model training and evaluation. The full list of 75 chart types is in Figs. 5-20.

### D.2    CHART VARIATIONS

For each chart type, we include multiple stylistic variations, designed along the following dimensions:

- Element shapes, such as rounded bars and curved bars.
- Icon placement, such as positioned above data elements and beside labels.
- Rendering effects, such as hand-drawn style and 3D style.
- Element alignment, such as center-aligned layout and edge-aligned layout.
- Gridline and axis design, ranging from minimalist to detailed ticks and axes.

These variations were observed from real-world infographics and verified by design experts, who also added complementary styles to ensure coverage and diversity. This results in a variation space that reflects real-world design patterns and supports diverse chart generation. We provide illustrations of the variations under each chart type (Figs. 5-20).

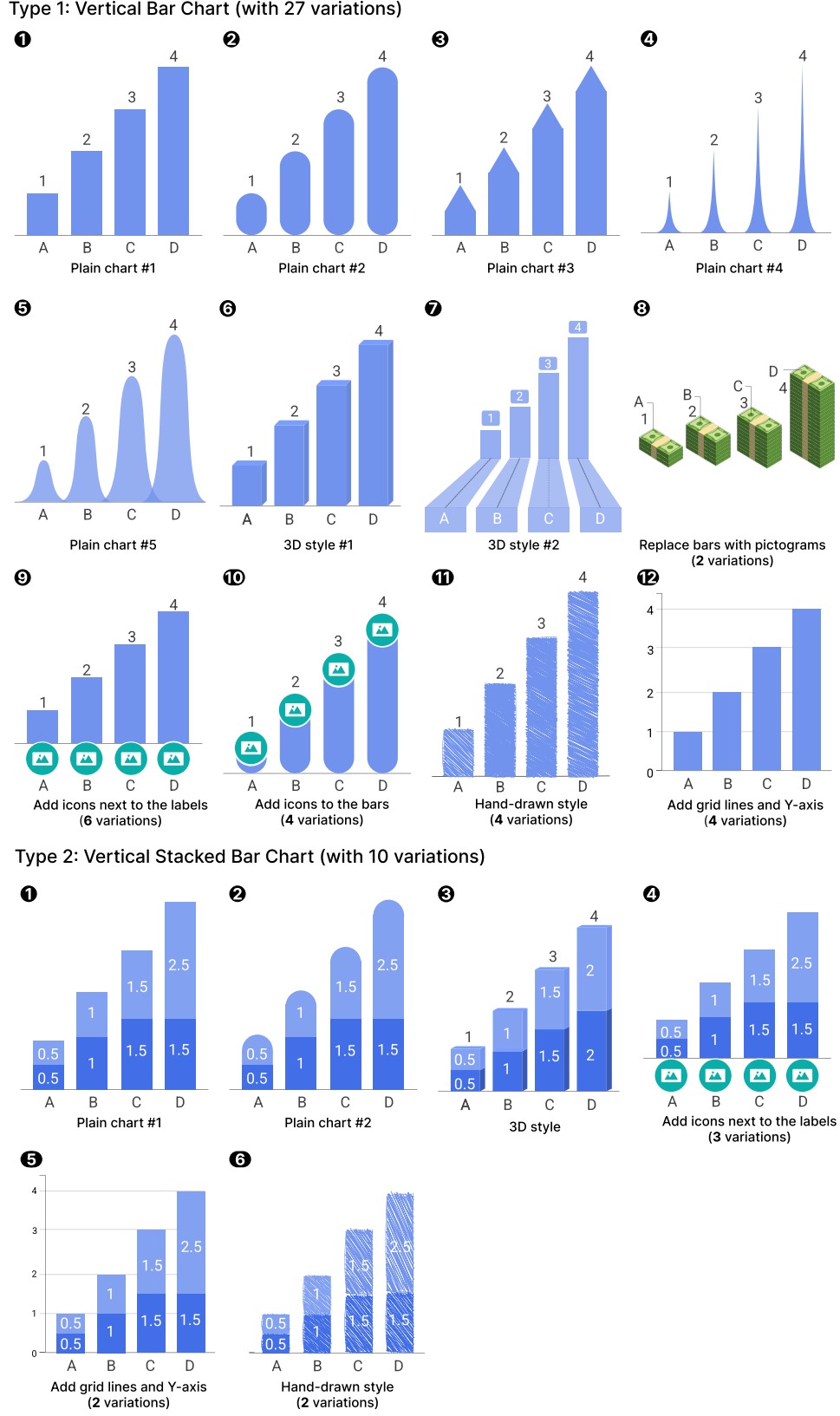

Figure 5: 75 chart types and 440 chart variations (Part 1).

Type 3: Vertical Grouped Bar Chart (with 24 variations)

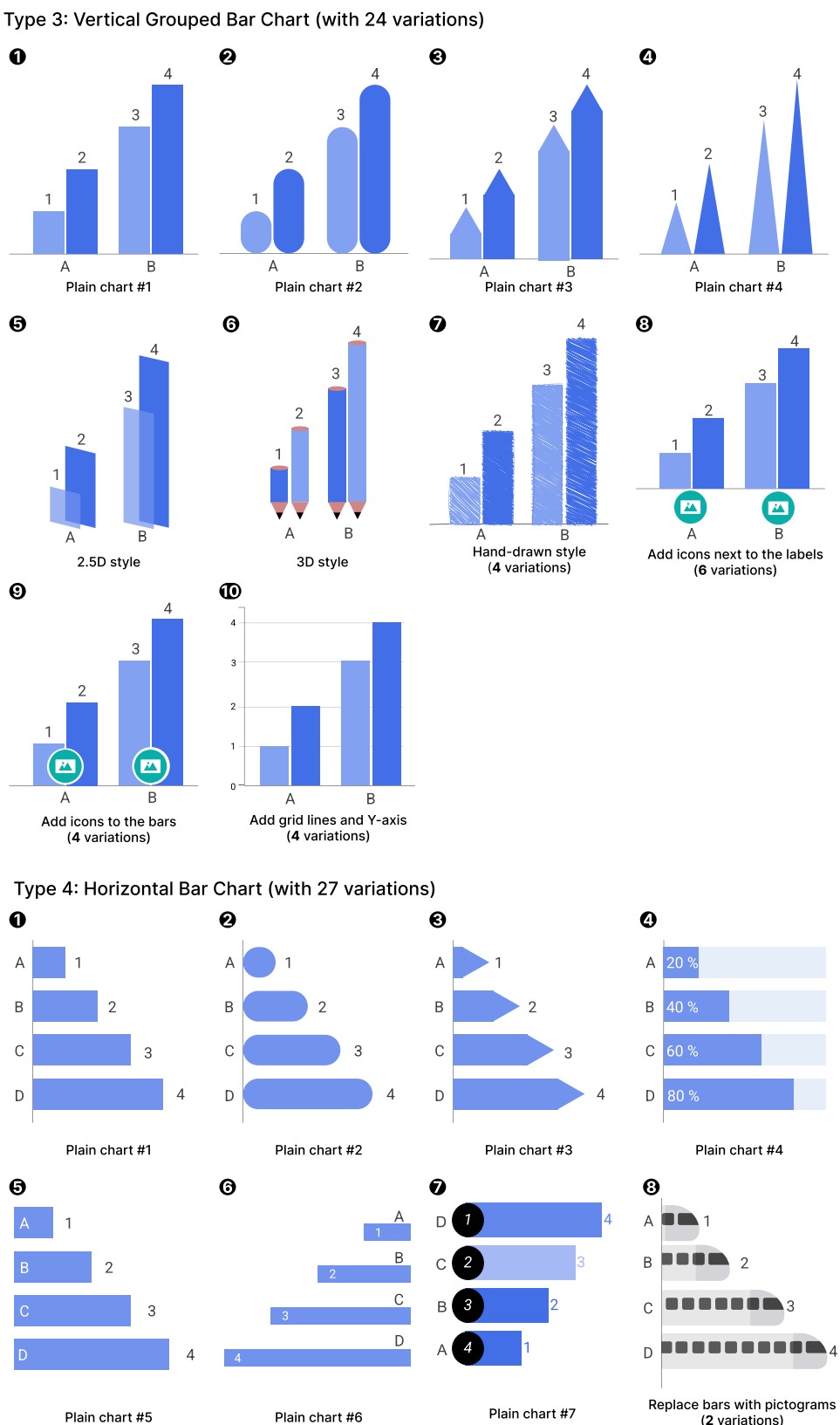

Type 4: Horizontal Bar Chart (with 27 variations)

Figure 6: 75 chart types and 440 chart variations (Part 2).

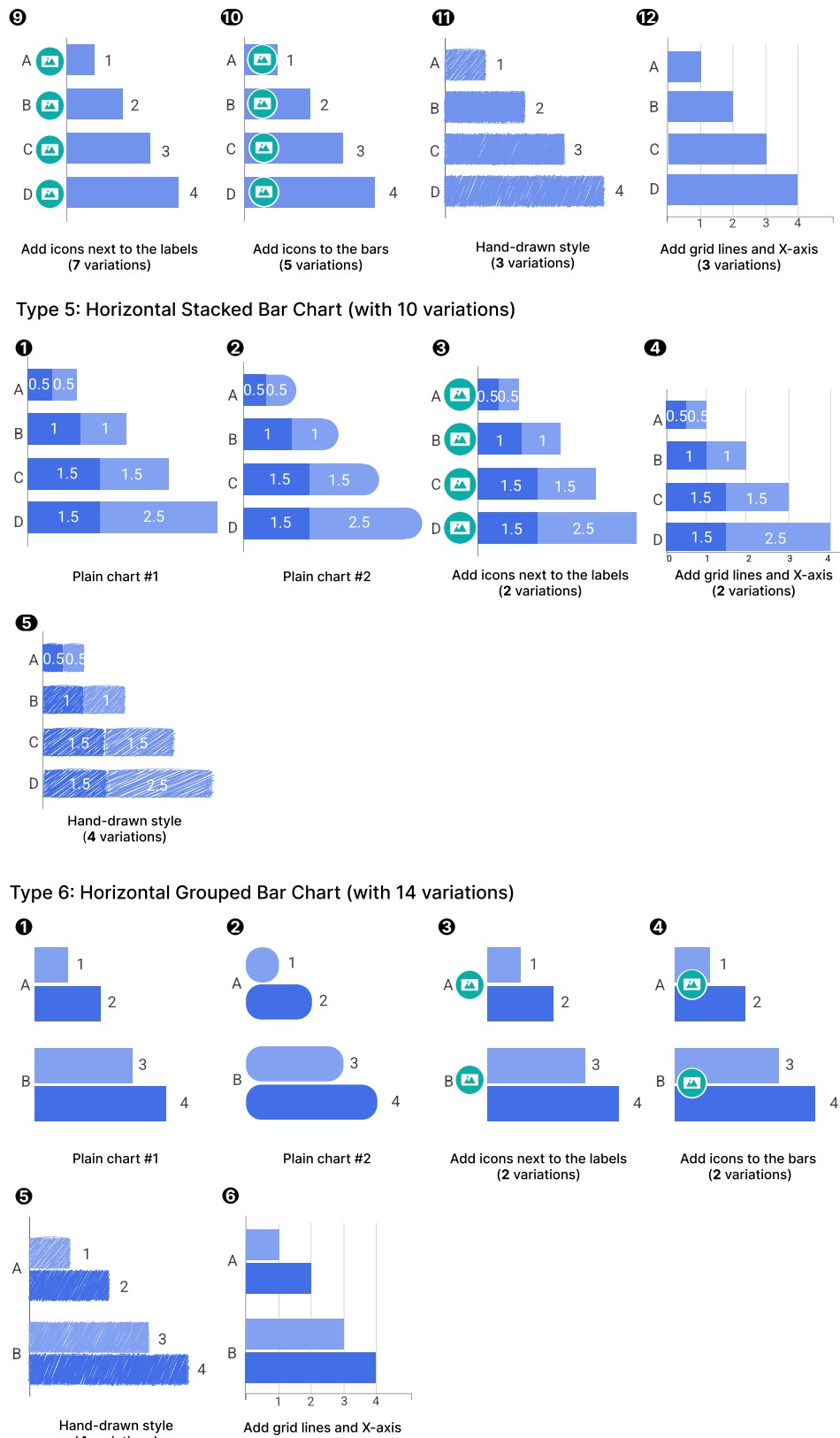

Figure 7: 75 chart types and 440 chart variations (Part 3).

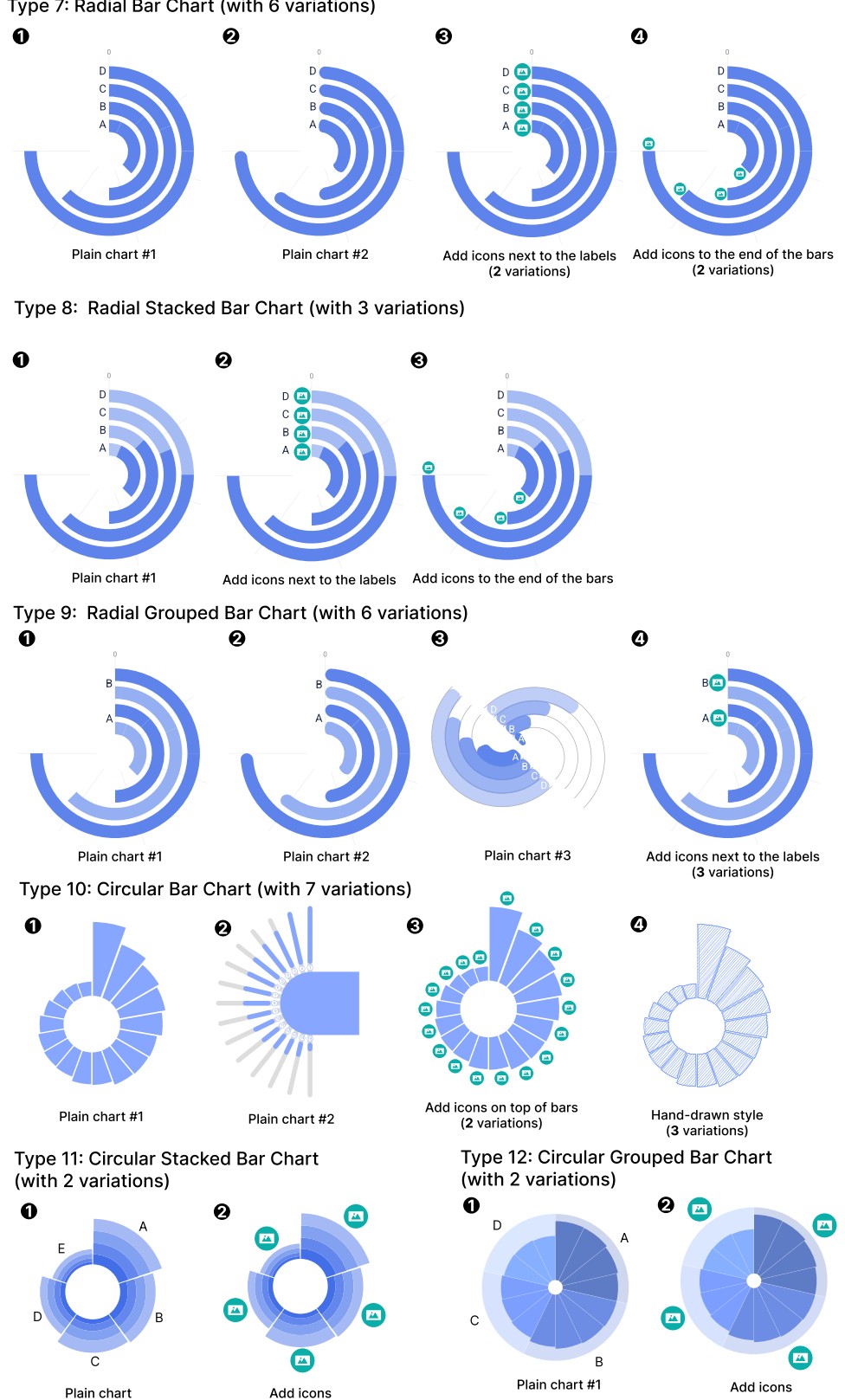

Figure 8: 75 chart types and 440 chart variations (Part 4).

Type 13: Pictorial Percentage Bar Chart (with 1 variation)

Type 14: Histogram (with 3 variations)

Type 15: Lollipop Chart (with 4 variations)

Type 16: Dot Chart (with 3 variations)

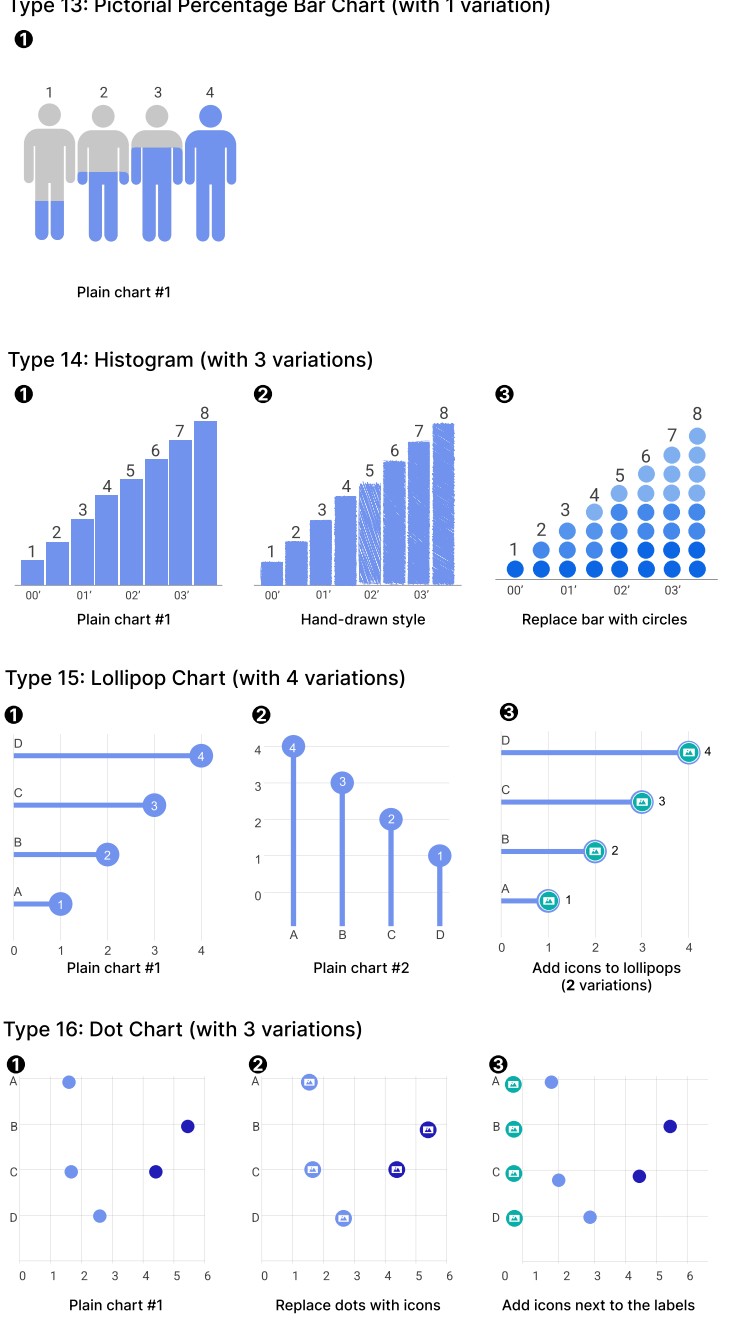

Figure 9: 75 chart types and 440 chart variations (Part 5).

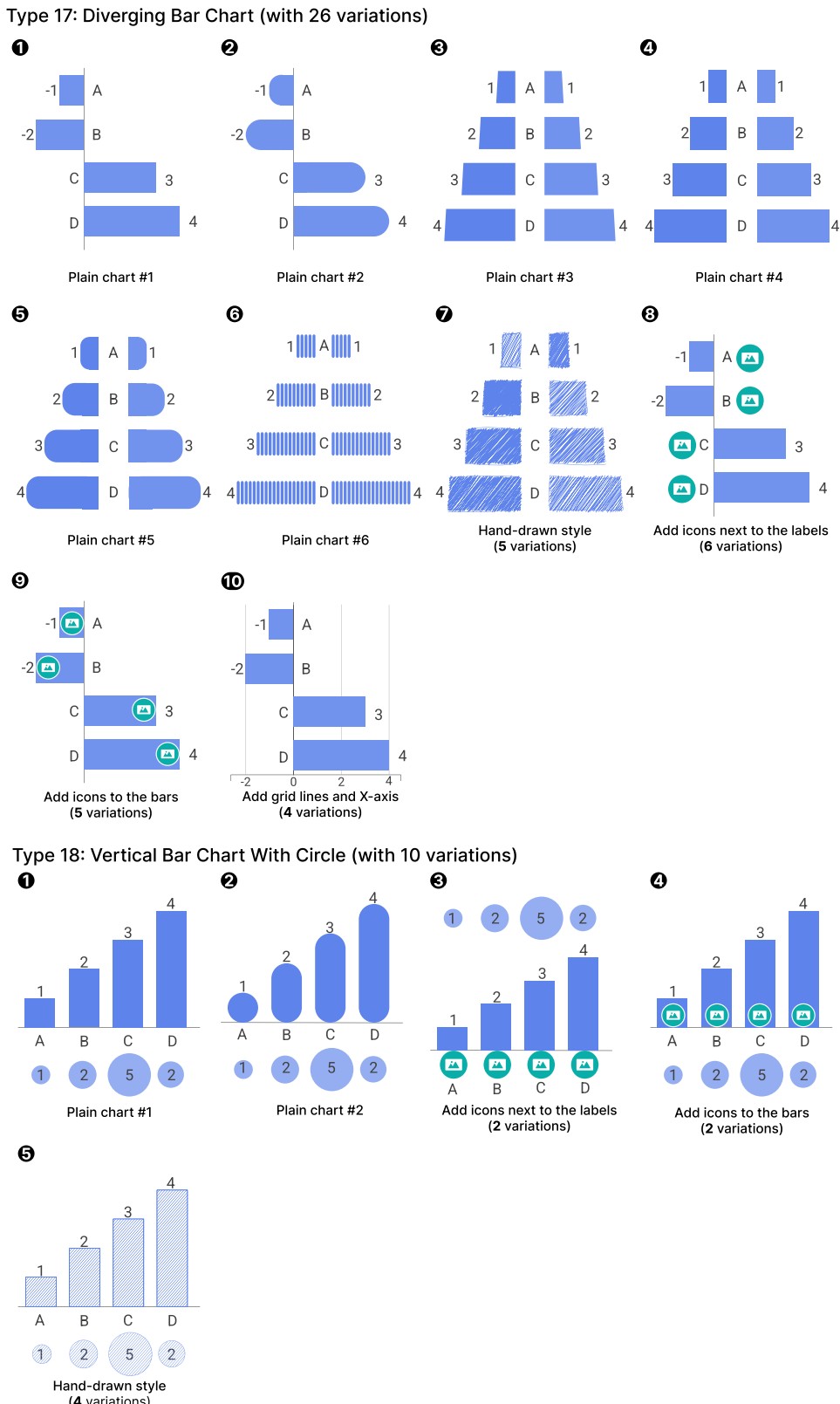

Figure 10: 75 chart types and 440 chart variations (Part 6).

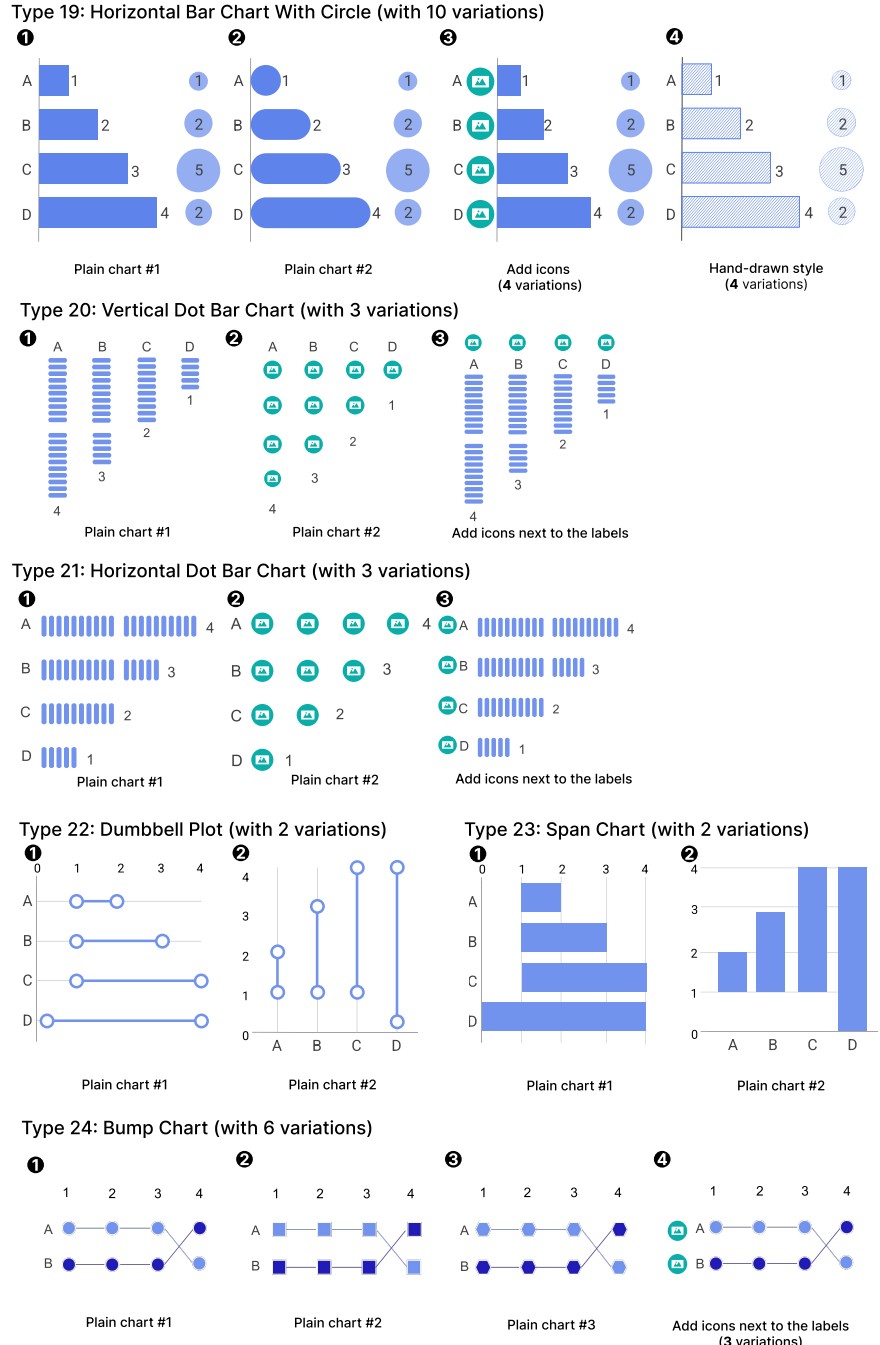

Figure 11: 75 chart types and 440 chart variations (Part 7).

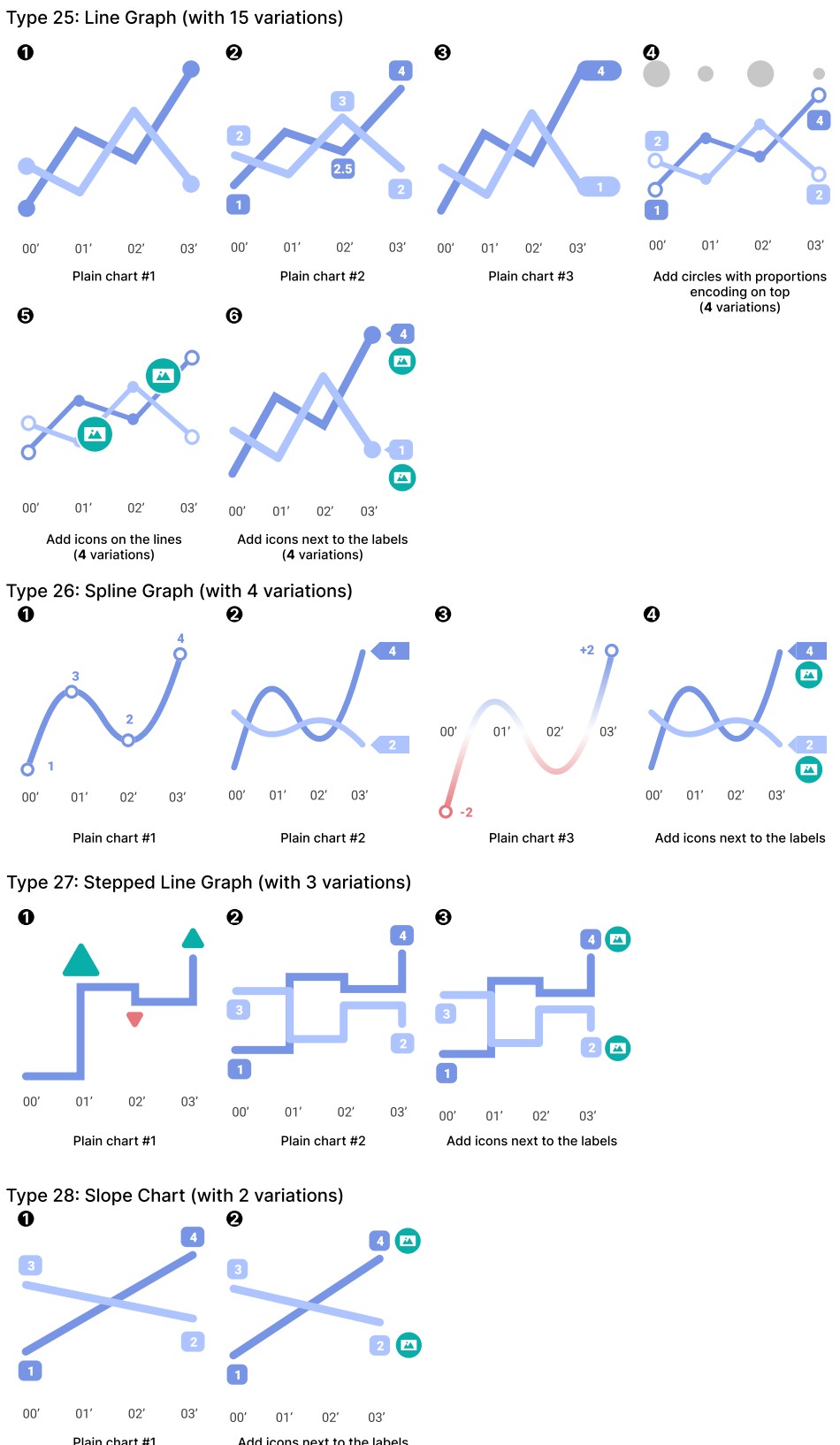

Figure 12: 75 chart types and 440 chart variations (Part 8).

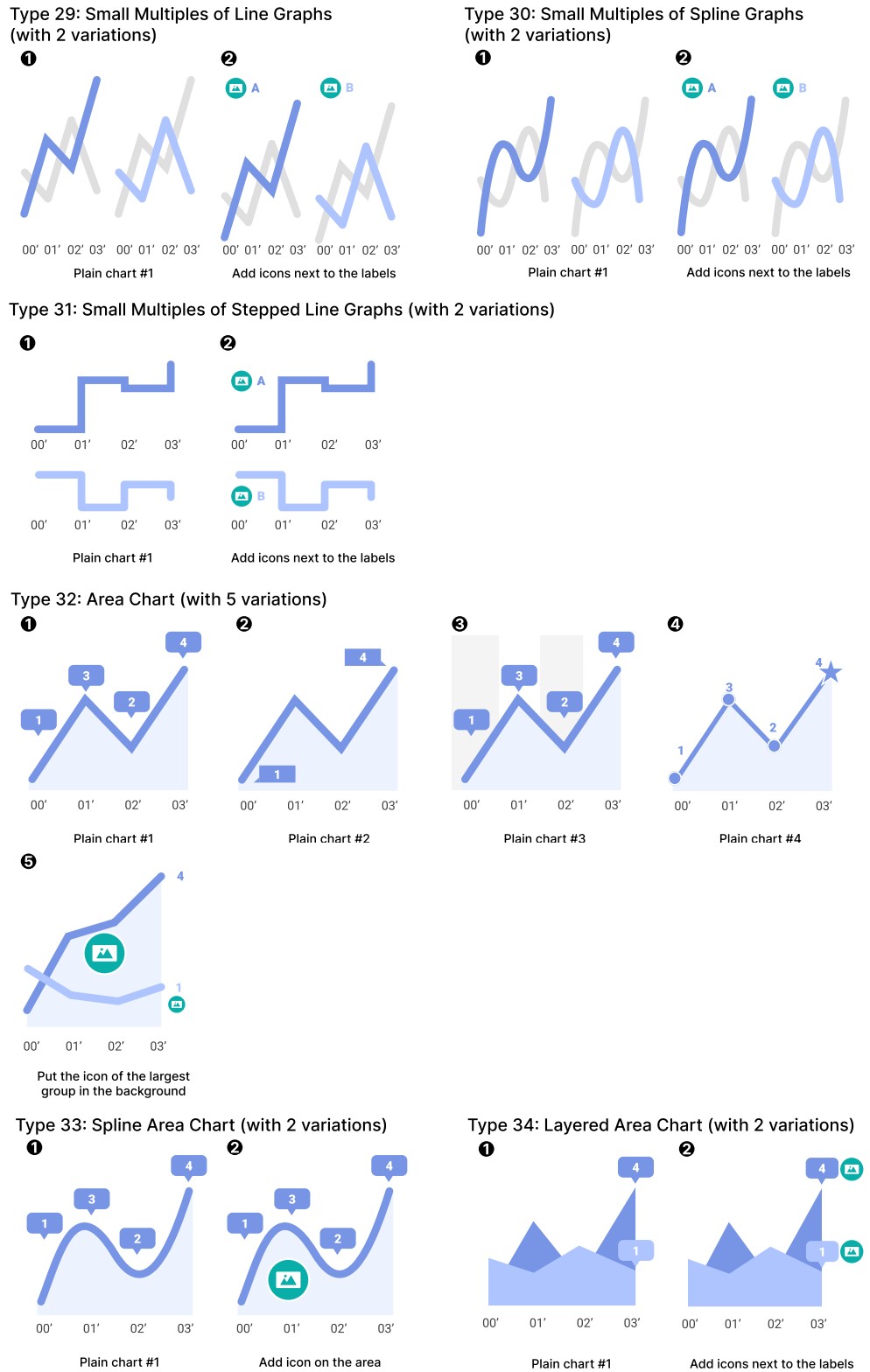

Figure 13: 75 chart types and 440 chart variations (Part 9).

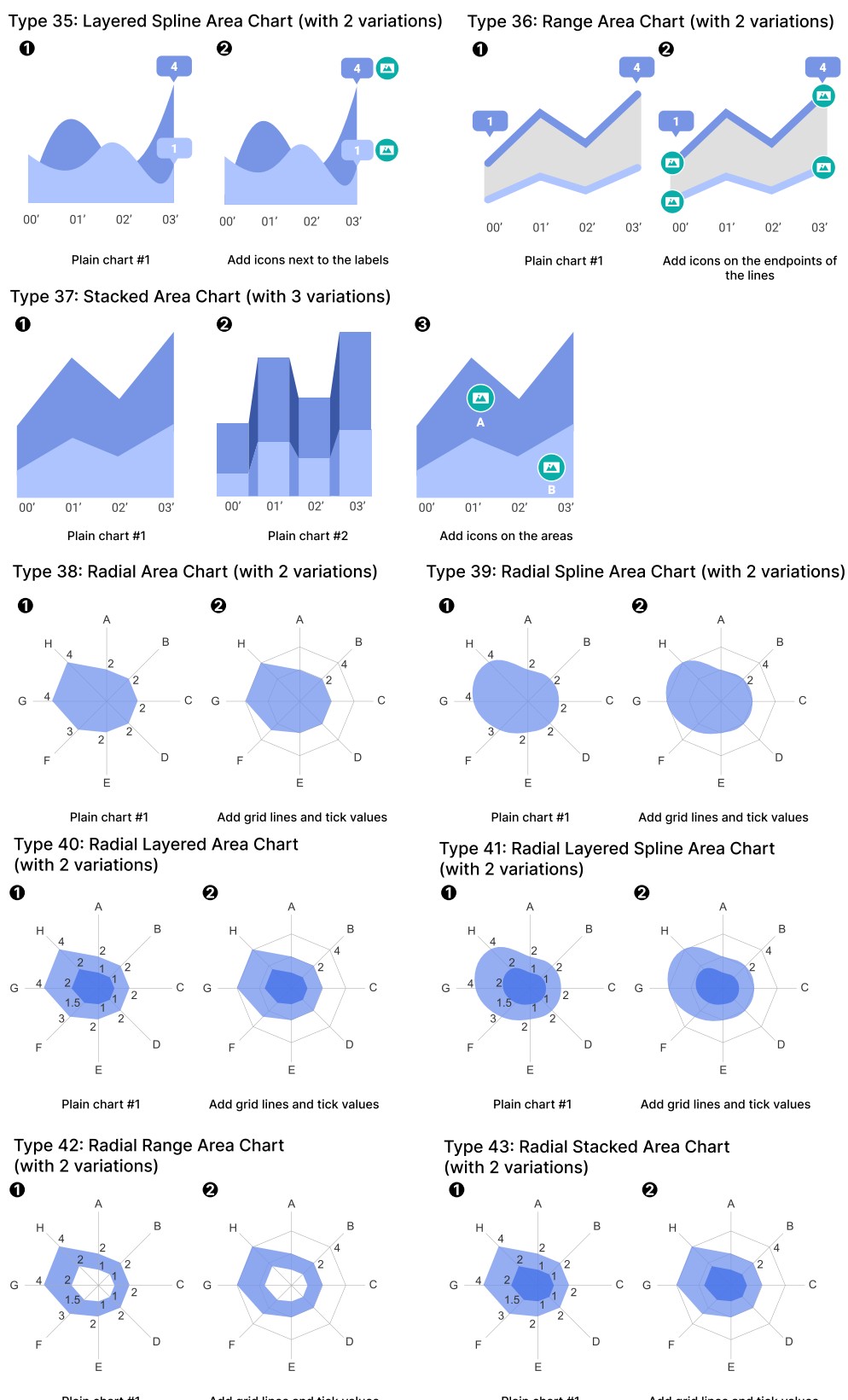

Figure 14: 75 chart types and 440 chart variations (Part 10).

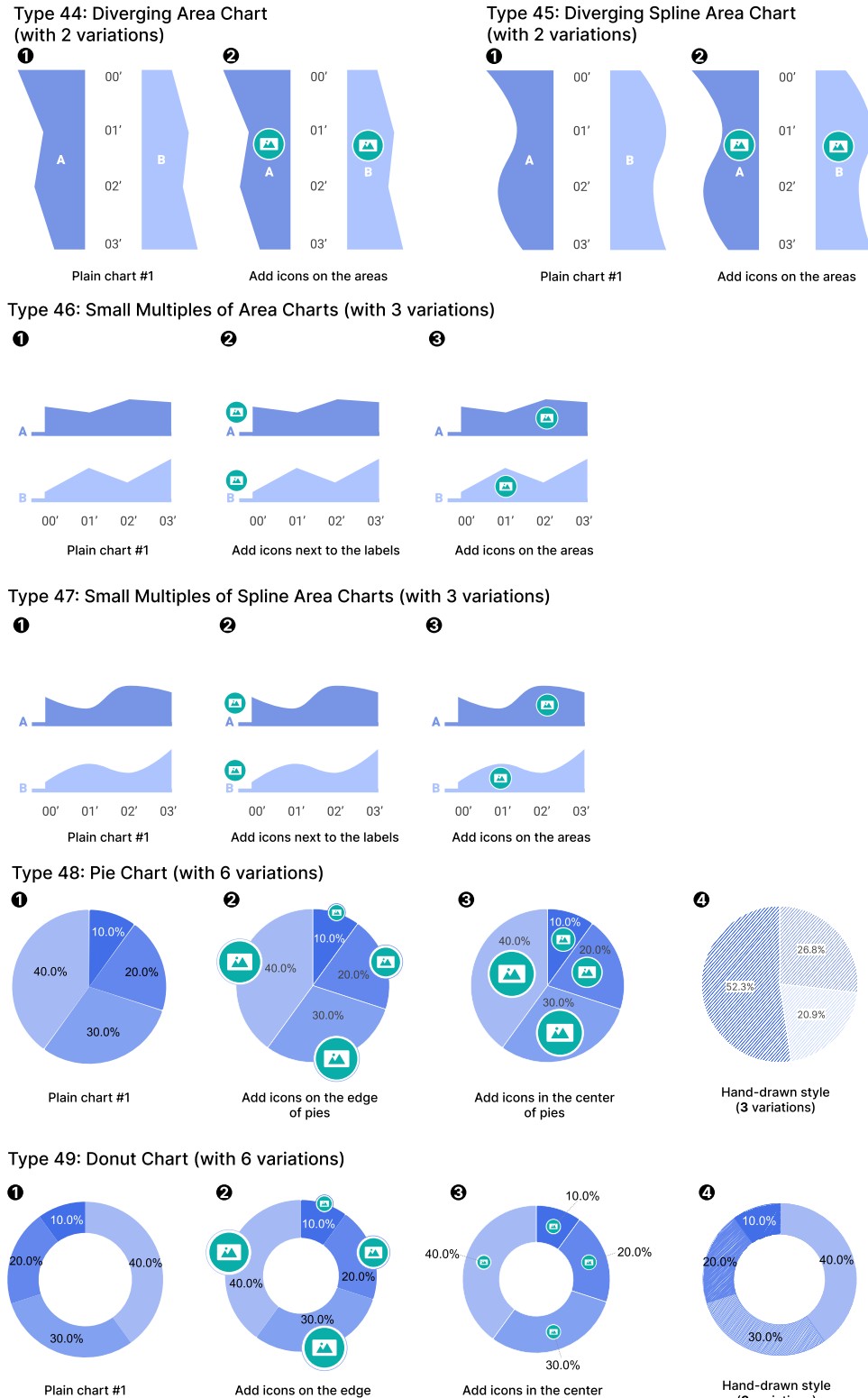

Figure 15: 75 chart types and 440 chart variations (Part 11).

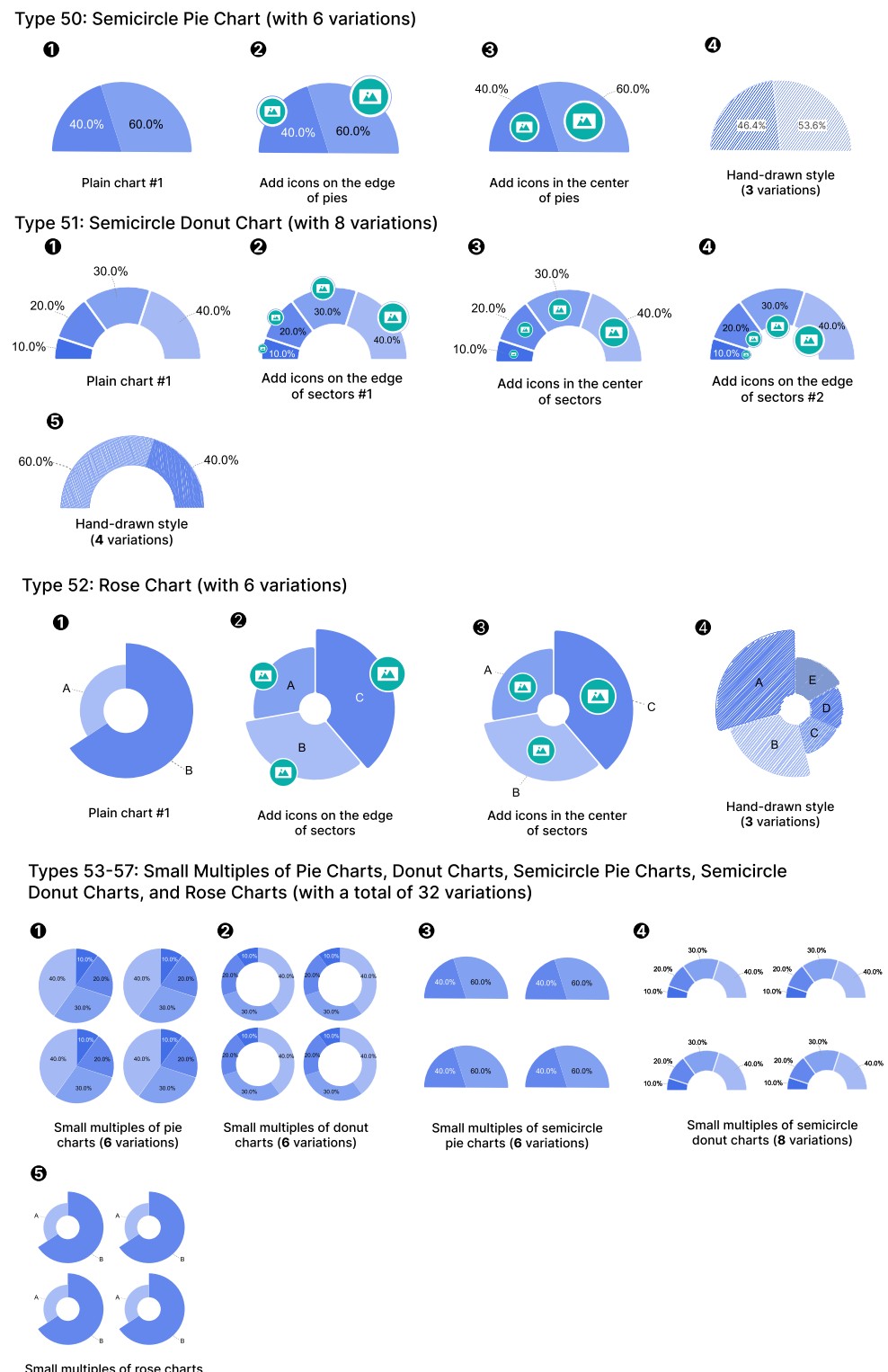

Figure 16: 75 chart types and 440 chart variations (Part 12).

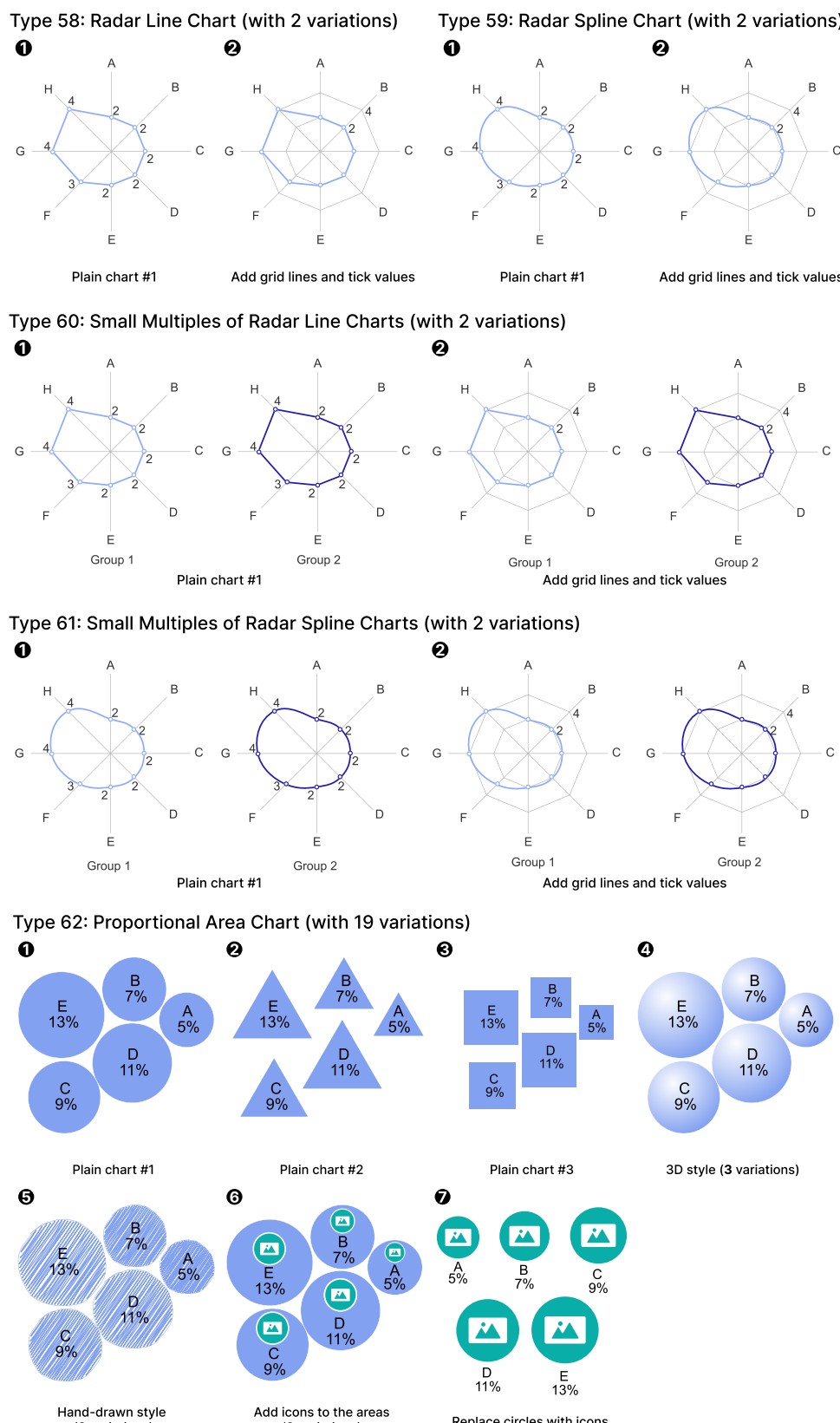

Figure 17: 75 chart types and 440 chart variations (Part 13).

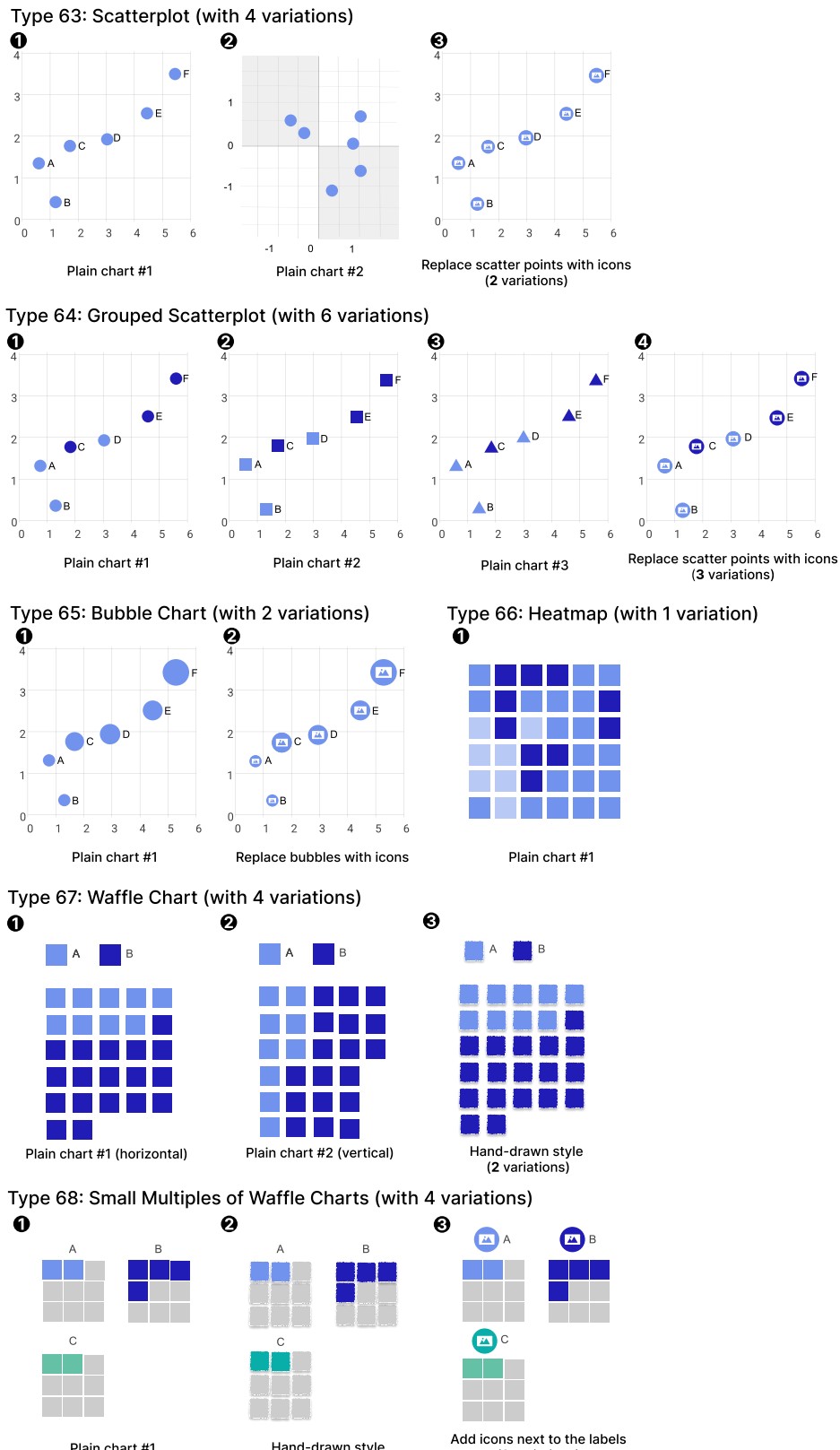

Figure 18: 75 chart types and 440 chart variations (Part 14).

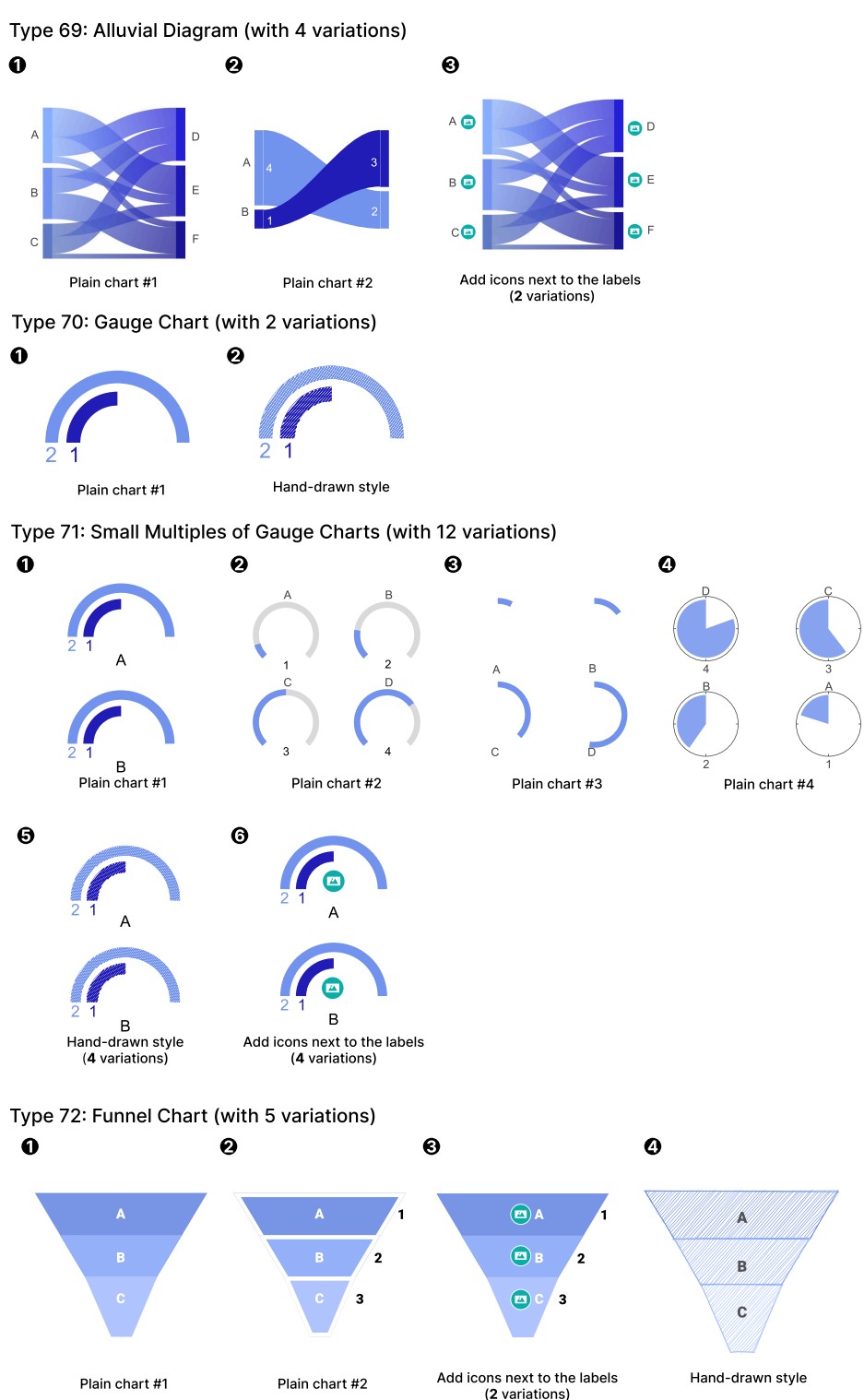

Figure 19: 75 chart types and 440 chart variations (Part 15).

Type 73: Pyramid Chart (with 7 variations)

Type 74: Treemap (with 12 variations)

Type 75: Voronoi Treemap (with 7 variations)

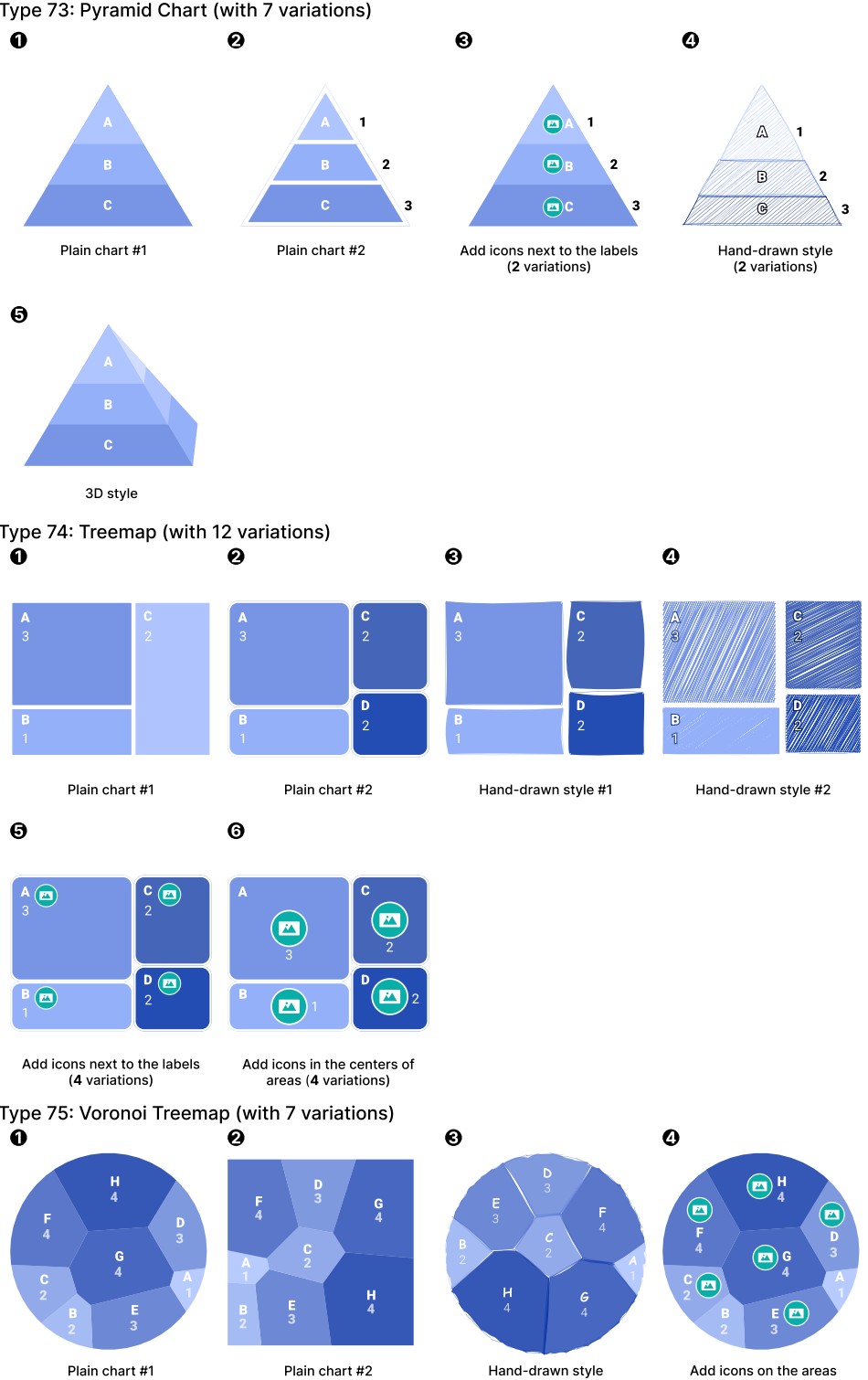

Figure 20: 75 chart types and 440 chart variations (Part 16).

# E CHART TYPE SELECTION

We determine chart types by analyzing the data attributes and their characteristics. First, we apply rule-based mapping (Table 6) that identifies candidate chart types based on data attribute combinations. If multiple candidates remain, we instruct Gemini-2.0-Flash to select the best fit by evaluating the compatibility between the data and these chart types. The selection prompt shown below provides candidate chart types, their descriptions, and key data statistics to inform the decision.

Table 6: Mapping rules defining attribute combinations for chart type selection. C, N, and T are abbreviations for Categorical, Numeric, and Temporal attributes, respectively. The notation $X \times k$ indicates $k$ distinct attributes of type $X$. When a symbol $*$ is specified (e.g., for the Diverging Bar Chart), it indicates that the first categorical attribute must contain exactly two distinct values.

| Chart Type | Attribute Combinations |
|---|---|
| Vertical Bar Chart | $C \times 1 + N \times 1$ |
| Vertical Stacked Bar Chart | $C \times 2 + N \times 1$ |
| Vertical Grouped Bar Chart | $C \times 2 + N \times 1$ |
| Horizontal Bar Chart | $C \times 1 + N \times 1$ |
| Horizontal Stacked Bar Chart | $C \times 2 + N \times 1$ |
| Horizontal Grouped Bar Chart | $C \times 2 + N \times 1$ |
| Radial Bar Chart | $C \times 1 + N \times 1$ |
| Radial Stacked Bar Chart | $C \times 2 + N \times 1$ |
| Radial Grouped Bar Chart | $C \times 2 + N \times 1$ |
| Circular Bar Chart | $C \times 1 + N \times 1$ |
| Circular Stacked Bar Chart | $C \times 2 + N \times 1$ |
| Circular Grouped Bar Chart | $C \times 2 + N \times 1$ |
| Pictorial Percentage Bar Chart | $C \times 1 + N \times 1$ |
| Histogram | $C \times 1 + N \times 1, T \times 1 + N \times 1$ |
| Lollipop Chart | $C \times 1 + N \times 1$ |
| Dot chart | $C \times 1 + N \times 1, C \times 2 + N \times 1$ |
| Diverging Bar Chart | $C \times 2 + N \times 1 *$ |
| Vertical Bar Chart With Circle | $C \times 1 + N \times 2$ |
| Horizontal Bar Chart With Circle | $C \times 1 + N \times 2$ |
| Vertical Dot Bar Chart | $C \times 1 + N \times 1$ |
| Horizontal Dot Bar Chart | $C \times 1 + N \times 1$ |
| Dumbbell Plot | $T \times 1 + N \times 1 + C \times 1 *$ |
| Span Chart | $C \times 1 + N \times 2$ |
| Bump Chart | $T \times 1 + N \times 1 + C \times 1$ |
| Line Graph | $T \times 1 + N \times 1, T \times 1 + N \times 1 + C \times 1$ |
| Spline Graph | $T \times 1 + N \times 1, T \times 1 + N \times 1 + C \times 1$ |
| Stepped Line Graph | $T \times 1 + N \times 1, T \times 1 + N \times 1 + C \times 1$ |
| Slope Chart | $T \times 1 + N \times 1 + C \times 1$ |
| Small Multiples of Line Graphs | $T \times 1 + N \times 1 + C \times 1$ |
| Small Multiples of Spline Graphs | $T \times 1 + N \times 1 + C \times 1$ |
| Small Multiples of Stepped Line Graphs | $T \times 1 + N \times 1 + C \times 1$ |
| Area Chart | $T \times 1 + N \times 1, T \times 1 + N \times 1 + C \times 1$ |
| Spline Area Chart | $T \times 1 + N \times 1, T \times 1 + N \times 1 + C \times 1$ |
| Layered Area Chart | $T \times 1 + N \times 1 + C \times 1$ |
| Layered Spline Area Chart | $T \times 1 + N \times 1 + C \times 1$ |
| Range Area Chart | $T \times 1 + N \times 1 + C \times 1 *$ |
| Stacked Area Chart | $T \times 1 + N \times 1 + C \times 1$ |
| Radial Area Chart | $T \times 1 + N \times 1 + C \times 1$ |
| Radial Spline Area Chart | $T \times 1 + N \times 1 + C \times 1$ |
| Radial Layered Area Chart | $T \times 1 + N \times 1 + C \times 1$ |
| Radial Layered Spline Area Chart | $T \times 1 + N \times 1 + C \times 1$ |
| Radial Range Area Chart | $T \times 1 + N \times 1 + C \times 1 *$ |
| Radial Stacked Area Chart | $T \times 1 + N \times 1 + C \times 1$ |
| Diverging Area Chart | $T \times 1 + N \times 1 + C \times 1 *$ |
| Diverging Spline Area Chart | $T \times 1 + N \times 1 + C \times 1 *$ |
| Small Multiples of Area Charts | $T \times 1 + N \times 1 + C \times 1$ |
| Small Multiples of Spline Area Charts | $T \times 1 + N \times 1 + C \times 1$ |
| Pie Chart | $C \times 1 + N \times 1$ |

Table 6: (Continued): Mapping rules defining attribute combinations for chart type selection.

| Chart Type | Attribute Combinations |
| --- | --- |
| Donut Chart | $C \times 1 + N \times 1$ |
| Semicircle Pie Chart | $C \times 1 + N \times 1$ |
| Semicircle Donut Chart | $C \times 1 + N \times 1$ |
| Rose Chart | $C \times 1 + N \times 1$ |
| Small Multiples of Pie Charts | $C \times 2 + N \times 1$ |
| Small Multiples of Donut Charts | $C \times 2 + N \times 1$ |
| Small Multiples of Semicircle Pie Charts | $C \times 2 + N \times 1$ |
| Small Multiples of Semicircle Donut Charts | $C \times 2 + N \times 1$ |
| Small Multiples of Rose Charts | $C \times 2 + N \times 1$ |
| Radar Line Chart | $C \times 1 + N \times 1$ |
| Radar Spline Chart | $C \times 1 + N \times 1$ |
| Small Multiples of Radar Line Charts | $C \times 2 + N \times 1$ |
| Small Multiples of Radar Spline Charts | $C \times 2 + N \times 1$ |
| Proportional Area Chart | $C \times 1 + N \times 1$ |
| Scatterplot | $C \times 1 + N \times 2$ |
| Grouped Scatterplot | $C \times 2 + N \times 2$ |
| Bubble Chart | $C \times 1 + N \times 2$ |
| Heatmap | $N \times 2$ |
| Waffle Chart | $N \times 1$ |
| Small Multiples of Waffle Charts | $C \times 1 + N \times 1$ |
| Alluvial Diagram | $C \times 1 + N \times 1 + T \times 1, C \times 2 + N \times 1$ |
| Gauge Chart | $N \times 1$ |
| Small Multiples of Gauge Charts | $C \times 1 + N \times 1$ |
| Funnel Chart | $C \times 1 + N \times 1$ |
| Pyramid Chart | $C \times 1 + N \times 1$ |
| Treemap | $C \times 2 + N \times 1$ |
| Voronoi Treemap | $C \times 2 + N \times 1$ |

# INPUT
**Candidate Chart Types:** {Candidate Chart Types}
**Chart Descriptions:** {Chart Descriptions}
**Attribute Statistics:** {Attribute Statistics}

Keep answers concise and direct.

# INSTRUCTION
**Role:** You are an expert assistant in data visualization and chart selection.
**Task:** Select the *single most optimal* chart type from {Candidate Chart Types} using their {Chart Descriptions} and {Attribute Statistics}. Prioritize **data-chart compatibility**: the chart must clearly and accurately represent key data insights.
**Instructions for Selection:**

1. Analyze all provided inputs.

2. Leverage your expertise to interpret how {Attribute Statistics} (e.g., categorical cardinality, number of temporal points, numerical value ranges, cumulative nature of data) impact the effectiveness and clarity of each candidate chart type, considering their {Chart Descriptions}.

3. Evaluate chart types based on descriptions, common visualization best practices, and your interpretation of {Attribute Statistics} to identify the most insightful and unambiguous visualization.

4. Primary goal: maximize data-chart compatibility.

**Output Format:** Return *only* the name of the single selected chart type.

# EXAMPLE OF TASK EXECUTION
**Input:**
**Candidate Chart Types:** ["Line Graph", "Area Chart", "Spline Graph"]
**Chart Descriptions:** "Line Graph: Emphasizes trends and rate of change over time; best for non-cumulative data with sufficient points. Area Chart: Shows trends and volume/magnitude over time; good for cumulative data or showing part-to-whole over time. Spline Graph: A Line Graph with smoothed curves, visually softens trends, suitable for data with many points or where a less angular look is desired."
**Attribute Statistics:** { "temporal_points": 30, "numeric_min_value": 15, "numeric_max_value": 450 }

**Output:**
Line Graph

## F  LAYOUT TEMPLATES

We summarize 68 layout templates from real infographic charts. We represent the layout template using the JSON format, which describes (1) the existence of elements (chart, image, and text), (2) their pairwise positional relationships, and (3) overlap relationships between their bounding boxes. A concrete example is shown below.

```
{
  # Element Existence
  "title": "yes",      # "yes" or "no"
  "image": "yes",
  "chart": "yes",

  # Position Relationships
  "title-to-chart": "left-top",    # top, bottom, and 7 other options
  "image-to-chart": "right-top",
  "title-to-image": "left",

  # Overlap Relationships Between Bounding Boxes
  "chart-title-overlap": "no",     # "yes" or "no"
  "chart-image-overlap": "no",
  "title-image-overlap": "no"
}
```

The illustrations of all 68 templates are shown in Figs. 21 and 22.

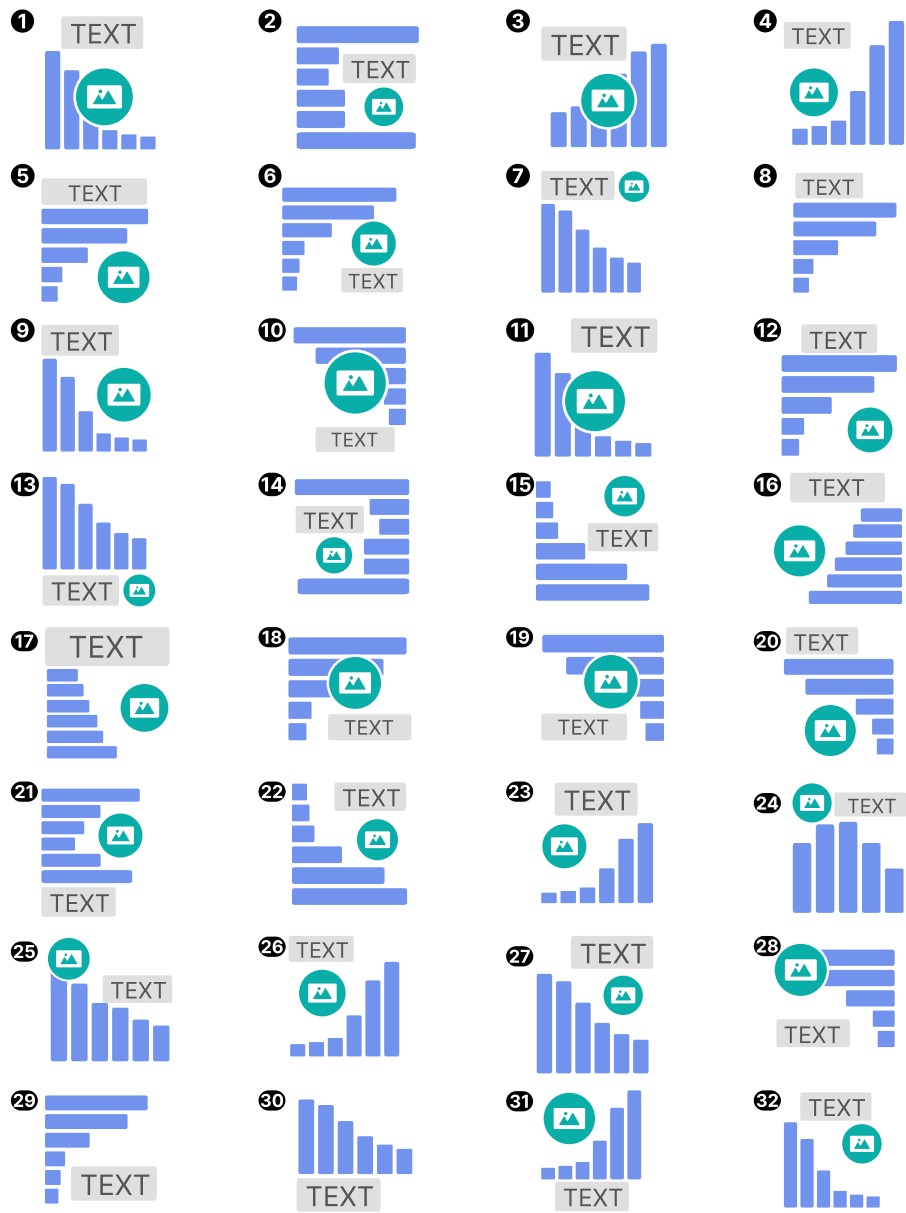

Figure 21: 68 layout templates (Part 1).

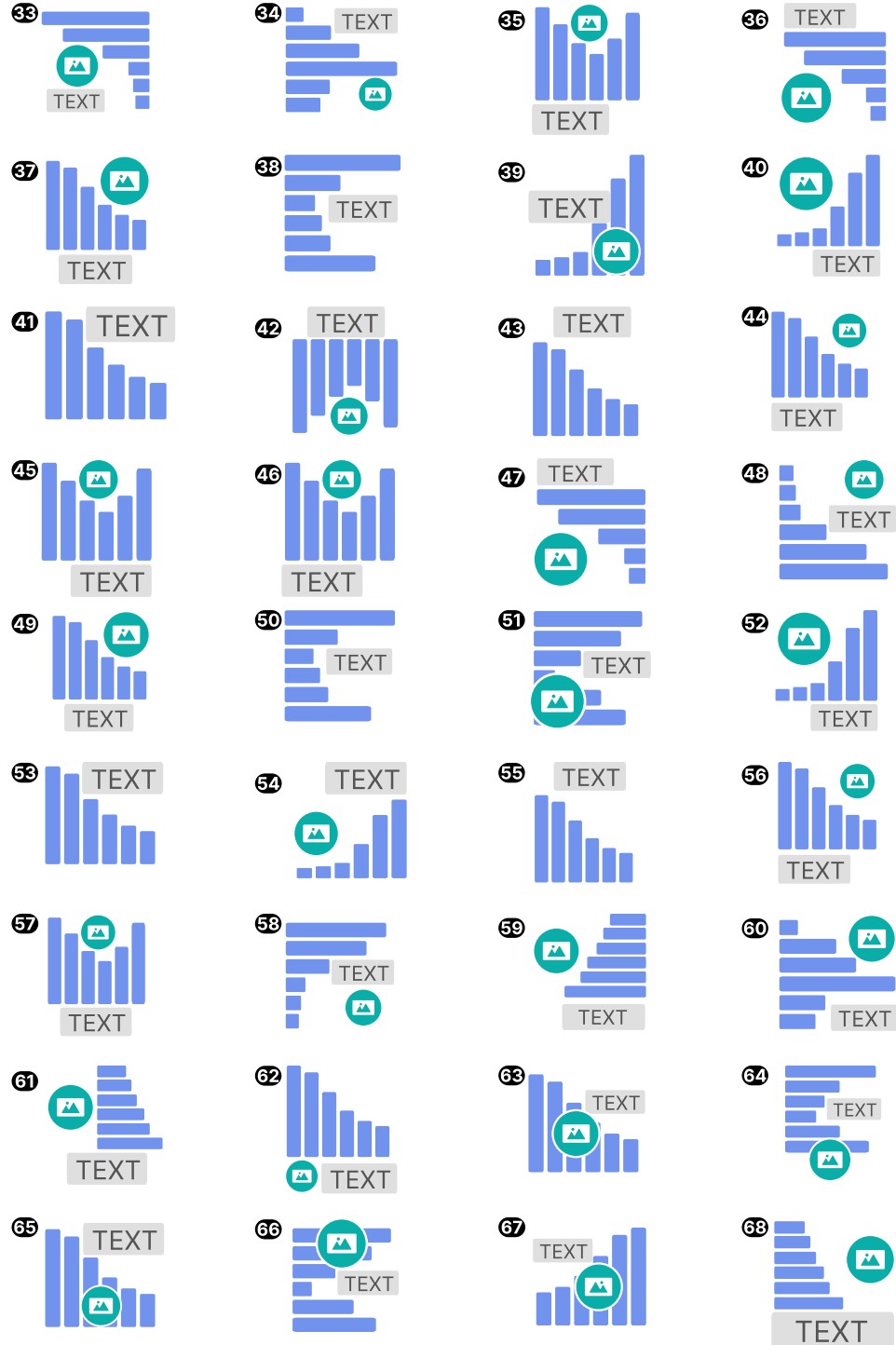

Figure 22: 68 layout templates (Part 2).

# G EXAMPLES OF SYNTHETIC INFOGRAPHIC CHARTS

Figs. 23 and 24 consist of synthetic infographic chart examples that offer a quick preview of ChartGalaxy. To access the full dataset, please visit our dataset repository[1].

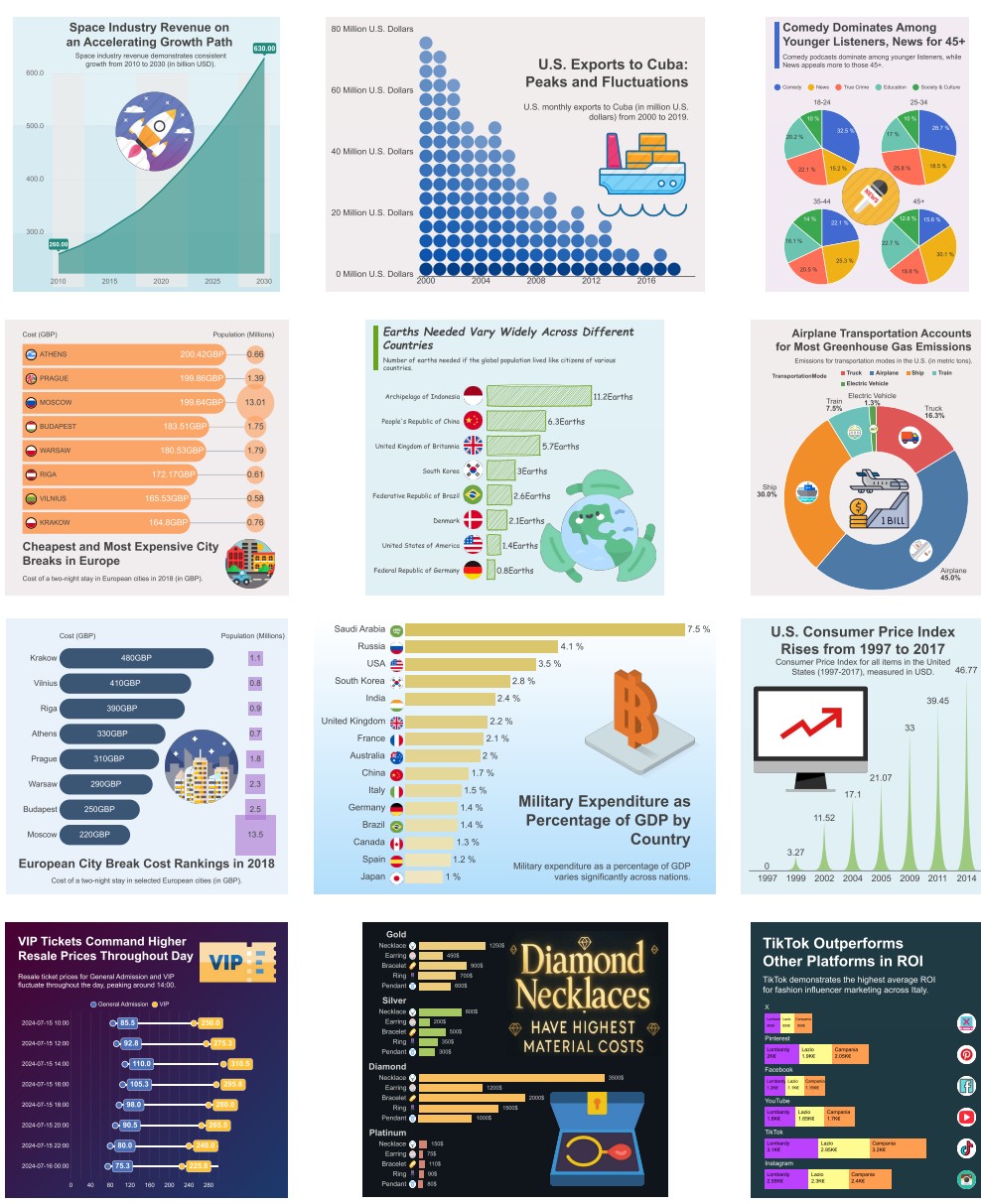

Figure 23: Synthetic infographic chart examples (Part 1).

[1]https://huggingface.co/datasets/ChartGalaxy/ChartGalaxy

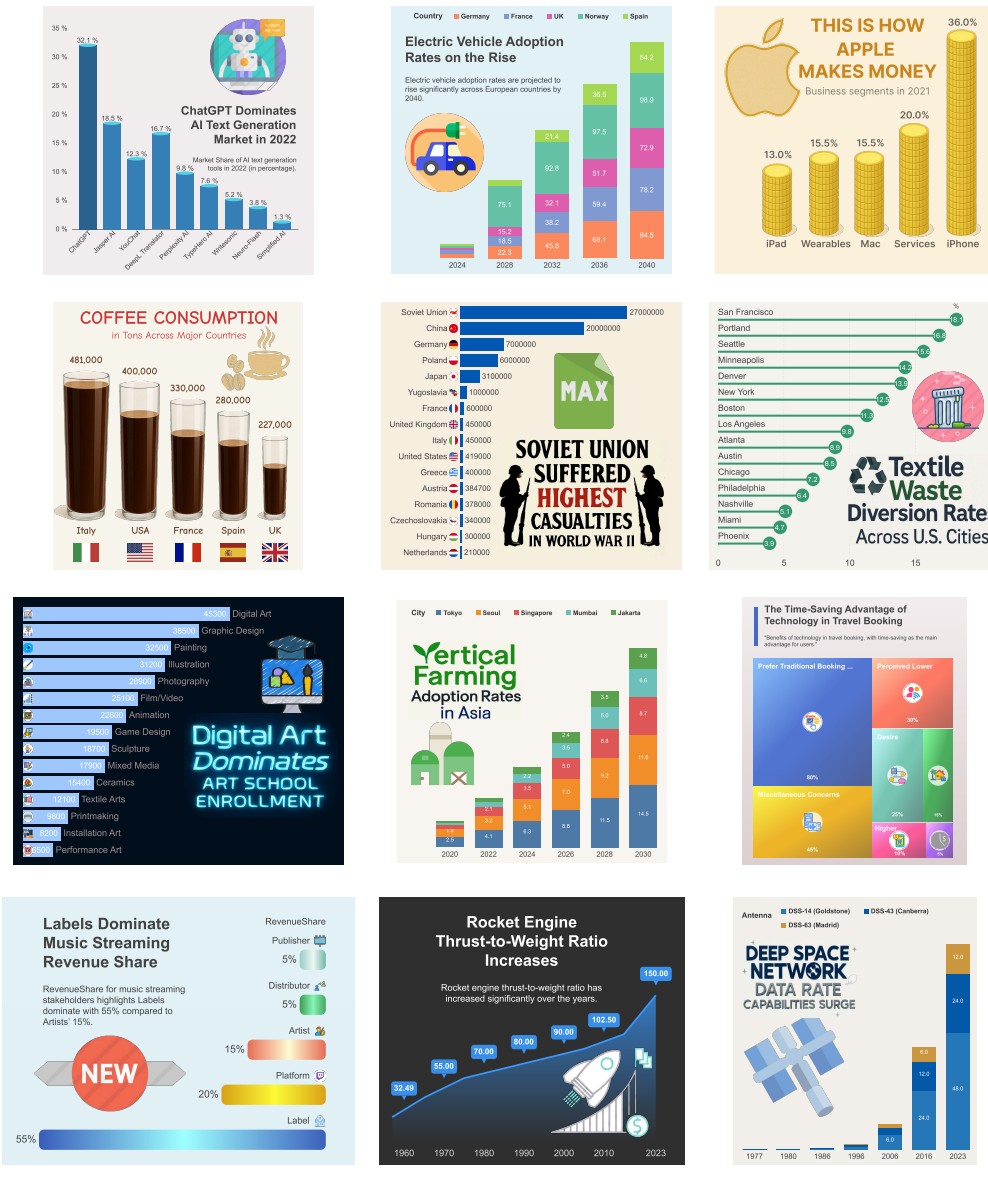

Figure 24: Synthetic infographic chart examples (Part 2).

# H  EXTENDED EVALUATION AND RESULTS

## H.1  INSTRUCTION DATASET FOR INFOGRAPHIC CHART UNDERSTANDING

Table 7: All infographic chart understanding questions with definitions and examples included in our test set.

| Question | Description | Example |
|---|---|---|
| **Text-Based Reasoning** | | |
| Data Identification (DI) | Identify and report specific data values from the chart based on textual references. | Which Country has an AdoptionRate closest to 29.9? Select the correct answer from the following options: A. India, B. China, C. Germany, D. South Africa |
| Data Comparison (DC) | Compare multiple data points and their relationships to analyze relative values and derive insights. | What is the difference between the highest and lowest MedalCount? Please provide a numerical answer. |
| Data Extraction with Condition (DEC) | Extract specific data points from the chart that meet certain conditions. | What is the total PollutionLevel of Beijing in the given image? Please provide a numerical answer. |
| Fact Checking (FC) | Verify statements about data by cross-checking information against the visual representation. | Is the ApplicationRate of Spain consistently less than the ApplicationRate of Italy across all Years? Answer with exactly 'Yes' or 'No'. |
| **Visual-Element-Based Reasoning** | | |
| Data Identification (DI) | Identify data values associated with specific visual elements (e.g., icons, symbols) in the chart. | What is the WaterConsumption for 🟢 in the Industrial group? Please provide a numerical answer. |
| Data Comparison (DC) | Compare data values associated with different visual elements, requiring cross-modal reasoning. | What is the difference between the AverageTeacherSalary of 🇮🇳 and 🇸🇬? Please provide a numerical answer. |
| Data Extraction with Condition (DEC) | Extract data values from visual elements when specific conditions are met. | What is the total Population of 🗼 in the given image? Please provide a numerical answer. |
| Fact Checking (FC) | Verify claims about data represented by visual elements, integrating visual and textual information. | Is the InterceptionRate of 2021 in 🔴 less than that in 🇺🇸? Answer with exactly 'Yes' or 'No'. |
| **Visual Understanding** | | |
| Style Detection (SD) | Identify design styles and formatting choices used in the chart. | What is the alignment style of the main text content (chart title and subtitle)? Select the correct answer from the following options: A. left-aligned, B. center-aligned, C. right-aligned, D. justified |
| Visual Encoding Analysis (VEA) | Analyze how data dimensions are encoded using visual properties (e.g., color, size, position, icons). | What data attribute or dimension is represented by icons in the given infographic chart? Select the correct answer from the following options: A. Location, B. WinPercentage, C. Team, D. None |
| Chart Classification (CC) | Identify the type of chart based on its visual characteristics and structure. | What types of charts are included in this infographic? Select the correct answer from the following options: A. Radial Grouped Bar Chart, B. Circular Grouped Bar Chart, C. Stacked Bar Chart, D. Diverging Bar Chart |

**Training details**  We fine-tune two LVLMs in our experiments. The first is InternVL3-8B (Zhu et al., 2025b), which combines the Qwen2.5-7B language model with the InternViT-300M-448px-V2.5 visual encoder. The

second is Qwen2.5-VL-7B (Bai et al., 2025), which integrates the same Qwen2.5-7B language model with a Vision Transformer architecture optimized by Qwen. These models are fine-tuned on our instruction dataset for 2 epochs using a global batch size of 128. For parameter-efficient tuning, we utilize LoRA (Hu et al., 2022) with a rank of 8. This training process is conducted on 8 NVIDIA 4090D 48G GPUs, leveraging Deepspeed ZerO for parallelized training. For both InternVL3-8B and Qwen2.5-VL-7B, the learning rate is set to $5 \times 10^{-5}$, and we employ a cosine learning rate scheduler with a warm-up ratio of 0.1. The LoRA parameters, $\alpha$ and $r$, are both set to 32.

**Evaluation details**    Our test set is designed to evaluate chart understanding capabilities on infographic charts. It includes a variety of question types, which are categorized into text-based reasoning, visual-element-based reasoning, and visual understanding. Table 7 provides a detailed breakdown of these categories, including the definition and an illustrative example for each specific question type.

**Limitations of pre-trained models in visual design literacy**    As shown in Table 2, fine-tuning on our dataset leads to notably larger improvements in visual understanding tasks, such as Style Detection (SD) and Visual Encoding Analysis (VEA), with average gains of +42.5% for InternVL3 and +42.3% for Qwen2.5-VL. In contrast, the improvements on the remaining tasks are more modest (+20.2% for InternVL3 and +17.2% for Qwen2.5-VL). This gap indicates that existing pre-training pipelines pay relatively little attention to visual design literacy: understanding how visual attributes like color, style, and other encodings convey information. Our dataset helps compensate for this missing capability by providing tasks that require such understanding. Incorporating such tasks and data into pre-training could meaningfully strengthen models' visual design literacy.

**Analysis of performance gap between two benchmarks**    As shown in Table 1, there is a performance gap of nearly 40 percent between InfographicVQA and ChartQAPro. As we can only infer from the design characteristics of the two benchmarks, we attribute the performance gap between InfographicVQA [2] and ChartQAPro [3] to differences in task and question design. InfographicVQA mainly focuses on visual extraction tasks, while ChartQAPro emphasizes "complex analytical reasoning," introducing more challenging question types such as "hypothetical questions," "conversational questions," and "unanswerable questions" that require reasoning beyond simple data retrieval. This distinction is also reflected in human performance reported in their original papers: InfographicVQA reaches 95.70% accuracy, while ChartQAPro only achieves 85.02%. Overall, these results suggest that current models handle recognition well, but analytical reasoning remains a major gap.

**Ablation results of fine-tuning with different datasets**    We conducted additional fine-tuning experiments on two datasets, CogAlign (Huang et al., 2025a) and ChartGemma (Masry et al., 2024b). We fine-tuned on each dataset using the same LoRA adapters, hyperparameters, and training setup as in Sec. 4.1, and trained for one epoch. We used the same two models as in Sec. 4.1, InternVL3-8B and Qwen2.5-VL-7B, and evaluated performance on ChartQAPro and InfographicVQA. The results are shown in Table 8.

Table 8: Model fine-tuning results with ChartGemma, CogAlign, and ChartGalaxy on two benchmarks.

| Model | ChartQAPro | InfographicVQA |
|---|---|---|
| InternVL3-8B | 38.15 | 76.19 |
| + ChartGemma | 38.60 | 76.23 |
| + CogAlign | 37.24 | 74.19 |
| + ChartGalaxy | **44.13** | **79.99** |
| Qwen2.5-VL-7B | 37.97 | 78.59 |
| + ChartGemma | 38.26 | 78.94 |
| + CogAlign | 37.48 | 77.73 |
| + ChartGalaxy | **41.56** | **83.03** |

Among the three datasets, **ChartGalaxy** yields the largest performance improvements for both models across both benchmarks.

**ChartGemma** brings relatively modest improvements, likely because its design, optimized for tasks such as chart summarization, chart-to-Markdown extraction, and program-aided solving, aligns only loosely with the complex analytical reasoning required by **ChartQAPro** and the visual extraction required for **InfographicVQA**.

**CogAlign** results in a performance drop, indicating negative transfer from task discrepancies. It emphasizes low-level geometric cues typical of visual arithmetic, and the resulting gradients likely dominate the shared LoRA updates, thereby suppressing the higher-level reasoning capabilities required for complex QA. This observation is consistent with the findings of Li et al. (2024), who highlight that the limited trainable capacity of LoRA-based methods can hinder cross-task generalization under divergent task distributions.

**Ablation Study on Real and Synthetic Data**    To further understand the role of synthetic data in chart understanding, we conducted an ablation experiment on chart understanding. Specifically, we fine-tuned InternVL3-8B and Qwen2.5-VL-7B on the real and synthetic infographics separately, and compared their performance on three benchmarks: the ChartGalaxy evaluation set, ChartQAPro, and InfographicsVQA.

As shown in Table 9, the models fine-tuned on both real and synthetic infographics perform the best, with observable advantages over those fine-tuned solely on real or synthetic data. We further observe that the gains from synthetic data are particularly notable on our evaluation set. This improvement arises because synthetic infographics provide bounding-box annotations for charts, text, and images, which facilitates the generation of complex instructions, especially for visual-element-based reasoning.

Table 9: Comparison between real, synthetic, and combined data on three benchmarks.

| Model | ChartGalaxy evaluation set | ChartQAPro | InfographicsVQA |
|---|---|---|---|
| InternVL3-8B | 53.20 | 38.15 | 76.19 |
| + Real | 55.74 | 42.58 | 78.67 |
| + Synthetic | 76.08 | 41.76 | 77.92 |
| + Both | **80.07** | **44.13** | **79.99** |
| Qwen2.5-VL-7B | 56.50 | 37.97 | 78.59 |
| + Real | 58.76 | 40.47 | 81.76 |
| + Synthetic | 77.80 | 40.07 | 81.01 |
| + Both | **80.35** | **41.56** | **83.03** |

**Additional Analysis on a Smaller Model**    We conducted additional experiments with a smaller model, Qwen2.5-VL-3B, and fine-tuned it on the ChartGalaxy QA dataset (Sec. 4.1). The evaluation was performed on three benchmarks: the ChartGalaxy evaluation set, ChartQAPro, and InfograpicVQA.

Table 10: Evaluation of Qwen2.5-VL-3B with and without fine-tuning on ChartGalaxy QA dataset.

| Model | ChartGalaxy evaluation set | ChartQAPro | InfograpicVQA |
|---|---|---|---|
| Qwen2.5-VL-3B | 52.42 | 30.89 | 74.39 |
| **+ ChartGalaxy** | **71.01** | **37.08** | **78.43** |

The results show consistent improvements across all benchmarks after fine-tuning. Notably, the model achieves a +18.59 gain on the ChartGalaxy evaluation set and +6.19 on ChartQAPro, demonstrating that smaller models can also effectively benefit from training on ChartGalaxy. These findings suggest that our dataset is broadly applicable and valuable even for resource-constrained VLMs.

## H.2    BENCHMARKING INFOGRAPHIC CHART CODE GENERATION

**Settings**    Table 12 lists the 17 LVLMs along with their API names used in this experiment. Since our task involves generating executable code with relatively long outputs, we conducted a small-scale pilot study to assess the basic code generation capability of each model. Based on this, we excluded models that consistently failed to produce meaningful or complete outputs under our task setting—for example, Phi-4—due to their limited capacity or inability to handle long sequences. For illustration, we present the results of two smaller models (Qwen2.5-VL-7B, Phi-4-6B) on our benchmark in Table 11. We use greedy decoding (temperature $\tau = 0$) across all models to ensure deterministic outputs. To maximize the chance of obtaining complete and executable code, we configure each model to generate as many tokens as possible, setting the maximum generation length

to $min(16384, A)$, where $A$ denotes the model's maximum generation limit. This helps mitigate the risk of incomplete outputs, which result in non-executable code.

Table 11: Evaluation results of Qwen2.5-VL-7B and Phi-4-6B.

| Model | Exec. Rate | Low-Level | High-Level | Overall |
|---|---|---|---|---|
| Qwen2.5-VL-7B | 43.80 | 13.85 | 9.25 | 11.55 |
| Phi-4-6B | 1.60 | 0.29 | 0.07 | 0.18 |

Table 12: API names of the evaluated LVLMs.

| Model | Type | API name |
|---|---|---|
| Gemini-2.5-Pro (Google, 2025b) | Proprietary | `gemini-2.5-pro-preview-05-06` |
| Gemini-2.5-Flash (Google, 2025a) | Proprietary | `gemini-2.5-flash-preview-04-17` |
| Claude-3.7-Sonnet (Anthropic, 2025) | Proprietary | `claude-3-7-sonnet-20250219` |
| GPT-4.1 (OpenAI, 2025a) | Proprietary | `gpt-4.1` |
| GPT-4.1-mini (OpenAI, 2025a) | Proprietary | `gpt-4.1-mini` |
| GPT-4.1-nano (OpenAI, 2025a) | Proprietary | `gpt-4.1-nano` |
| OpenAI-o4-mini (OpenAI, 2025d) | Proprietary | `o4-mini` |
| OpenAI-o3 (OpenAI, 2025d) | Proprietary | `o3` |
| OpenAI-o1 (OpenAI, 2025c) | Proprietary | `o1` |
| GPT-4o (OpenAI, 2025b) | Proprietary | `gpt-4o-2024-11-20` |
| Doubao-1.5-Vision-Pro (ByteDance, 2025) | Proprietary | `Doubao-1.5-vision-pro-32k` |
| Moonshot-v1-Vision (Moonshot AI, 2025) | Proprietary | `moonshot-v1-32k-vision-preview` |
| Llama-4-Maverick-17B (Meta, 2025) | Open-Source | `chutesai/Llama-4-Maverick-17B-128E-Instruct-FP8` |
| Llama-4-Scout-17B (Meta, 2025) | Open-Source | `chutesai/Llama-4-Scout-17B-16E-Instruct` |
| Qwen2.5-VL-72B (Bai et al., 2025) | Open-Source | `Qwen/Qwen2.5-VL-72B-Instruct` |
| Qwen2.5-VL-32B (Bai et al., 2025) | Open-Source | `Qwen/Qwen2.5-VL-32B-Instruct` |
| InternVL3-78B (Zhu et al., 2025b) | Open-Source | `internvl3-78b` |

**Benchmark details**   Our benchmark includes 75 chart types and 68 layout templates in ChartGalaxy. The associated tabular data contains an average of 15.02 data points. We also compute statistics on the number of SVG elements in all infographic charts, as this metric partially reflects the complexity of reproducing a given chart. On average, each chart contains 77.93 SVG elements, including 28.52 `text` elements and 8.07 `image` elements. Other commonly used visual elements include `rect` ($M = 24.36$), `circle` ($M = 6.21$), `path` ($M = 5.78$), and `line` ($M = 3.21$). Among all chart types, waffle charts are the most element-dense, with an average of 677.55 elements, while funnel charts are the simplest, with an average of only 14.60 elements.

**Evaluation Metrics**   We present the details of the evaluation metrics of our benchmark, including the high-level score and the low-level score.

For the high-level score, we employ GPT-4o (OpenAI, 2025b) to assess the visual similarity between the PNG image rendered by the generated code and the ground-truth one. We instruct GPT-4o to evaluate the similarity along six dimensions: data element, layout, text, image, color, and validity. The model outputs a score for each of the six dimensions, which are then summed to produce a total score ranging from 0 to 100. The detailed prompt for this evaluation is provided in Supp. I.2.

The low-level score evaluates the fine-grained similarity between SVG elements of the rendered chart and the corresponding ground-truth chart. This evaluation is conducted through three steps: 1) decomposing both charts into SVG elements, 2) matching elements between the two charts, and 3) computing similarity metrics based on the matching results (Si et al., 2025; Chen et al., 2024). Algorithm 1 presents the pseudo-code for the matching procedure. The full implementation details are available in our publicly accessible code repository[2].

Based on the matching results, the low-level score is computed as the average of six similarity metrics: area, text, image, color, position, and size. Let the parsed SVG elements of the ground-truth chart and the generated chart be denoted by $G = \{g_1, g_2, \ldots, g_m\}$ and $P = \{p_1, p_2, \ldots, p_n\}$, respectively, and let the set of matching pairs between $G$ and $P$ be $M$, where $(i, j) \in M$ indicates that $g_i$ is matched with $p_j$. The detailed definitions and calculations of these metrics are provided below.

The area metric quantifies the proportion of the matched element areas relative to the total element areas:

$$\text{Area} = \frac{\sum_{(i,j)\in M}\big(S(g_i) + S(p_j)\big)}{\sum_{i=1}^{m} S(g_i) + \sum_{j=1}^{n} S(p_j)}, \tag{1}$$

where $S(\cdot)$ denotes the size of an element.

---

[2]`https://github.com/ChartGalaxy/ChartGalaxy`

---

**Algorithm 1** SVG Element Matching Algorithm

---

**Require:** $gt\_leafs$: Ground truth SVG elements.
**Require:** $pr\_leafs$: Predicted SVG elements.
**Require:** $gt\_matched$: Array for ground truth matches (init with -1).
**Require:** $pr\_matched$: Array for prediction matches (init with -1).
**Ensure:** Updated matching information between elements.
1: $m \leftarrow |gt\_leafs|, n \leftarrow |pr\_leafs|$
2: $CostMatrix \leftarrow$ ZeroMatrix$(m, n)$
3: **for** $i \leftarrow 0 \ldots m - 1$ **do**
4:     **for** $j \leftarrow 0 \ldots n - 1$ **do**
5:         $CostMatrix[i][j] \leftarrow$ LeafCost$(gt\_leafs[i], pr\_leafs[j])$
6:     **end for**
7: **end for**
8: $(rows, cols) \leftarrow$ HungarianAlgorithm$(CostMatrix)$      ▷ Returns optimal row-column pairs
9: **for** each pair $(i, j)$ in $(rows, cols)$ **do**
10:     **if** $CostMatrix[i][j] \leq 1$ AND $gt\_matched[i] = -1$ AND $pr\_matched[j] = -1$ **then**
11:         $gt\_matched[i] \leftarrow j$
12:         $pr\_matched[j] \leftarrow i$
13:     **end if**
14: **end for**

---

The text and image metrics evaluate the similarity of generated text and image elements, respectively, by averaging the similarity scores over all matched pairs of `text` and `image` elements. Unmatched `text` and `image` elements in ground-truth charts are assigned a similarity score of 0 to penalize generation failures. For `text` elements, similarity is computed using the character-level Sørensen-Dice coefficient, defined as twice the number of overlapping characters divided by the total number of characters in the two strings (Si et al., 2025). For `image` elements, similarity is measured using the CLIP embedding-based similarity (Radford et al., 2021).

The color, position, and size metrics assess visual consistency across matched elements with respect to their respective attributes. The color metric employs the CIEDE2000 formula (Luo et al., 2001) to measure the perceptual difference between the colors of matched elements. The position and size metrics are defined as follows:

$$\text{Position} = \frac{1}{|M|} \sum_{(i,j) \in M} \left[1 - \max\left(|X(g_i) - X(p_j)|, |Y(g_i) - Y(p_j)|\right)\right], \tag{2}$$

where $(X(e), Y(e))$ denotes the normalized coordinates of the center of element $e$,

$$\text{Size} = \frac{1}{|M|} \sum_{(i,j) \in M} \left[1 - \frac{|S(g_i) - S(p_j)|}{\max\left(S(g_i), S(p_j)\right)}\right], \tag{3}$$

where $S(\cdot)$ denotes the size of an element.

**Additional Analysis on LVLM performance** We present additional results and analysis of LVLM performance on our benchmark, focusing on generated code length, performance across varying levels of complexity, and qualitative outcomes from the three top-performing models.

*Generated code length and model performance.* Table 13 reports the generated code length and overall performance of 17 LVLMs, with corresponding visualizations in Fig. 25. Code length is measured by token count using the GPT-2 tokenizer across all executable code segments. The statistics reveal significant variation in generated code length among different LVLMs. Among the 12 proprietary models, GPT-4.1-mini produces the longest code on average (7,656.71 tokens), surpassing both GPT-4.1 (5,237.78 tokens) and GPT-4.1-nano (2,387.05 tokens). This longer code length may partly explain GPT-4.1-mini's comparable performance to GPT-4.1. In contrast, Gemini-2.5-Pro and Gemini-2.5-Flash generate code of similar average length (6,662.45 vs. 6,508.22 tokens). OpenAI's o-series models produce the shortest average code length among top performers (2,886.75 for OpenAI-o4-mini, 2,662.11 for OpenAI-o1, and 3,114.74 for OpenAI-o3), which may reflect their ability to generate more concise solutions. Among the five open-source models, Qwen2.5-VL-32B has the longest average code length despite achieving the lowest overall score, while the remaining models exhibit comparable average lengths. These findings highlight the distinct coding styles of different LVLMs when generating extended code sequences.

*Model performance across different complexity levels.* To comprehensively assess LVLM performance across varying degrees of difficulty, we divide our benchmark into three splits corresponding to different complexity

Table 13: Generated code length and overall scores of different LVLMs. We measure code length in terms of tokens, utilizing the GPT-2 tokenizer.

| Model | Length (AVG.) | Length (STD.) | Overall |
|---|---|---|---|
| *Proprietary* | | | |
| Gemini-2.5-Pro | 6,662.45 | 1,674.01 | **85.21** |
| GPT-4.1 | 5,237.78 | 1,675.36 | 80.00 |
| Claude-3.7-Sonnet | 5,870.50 | 1,552.48 | 79.91 |
| GPT-4.1-mini | **7,656.71** | **2,488.46** | 79.69 |
| OpenAI-o4-mini | 2,886.75 | 735.19 | 75.97 |
| Gemini-2.5-Flash | 6,508.22 | 1,934.79 | 75.55 |
| OpenAI-o1 | 2,662.11 | 1,018.11 | 74.69 |
| OpenAI-o3 | 3,114.74 | 999.66 | 74.22 |
| GPT-4o | 2,962.72 | 613.42 | 65.67 |
| GPT-4.1-nano | 2,387.05 | 1,306.72 | 60.06 |
| Doubao-1.5-Vision-Pro | 3,878.70 | 1,347.10 | 47.11 |
| Moonshot-v1-Vision | 3,087.17 | 791.16 | 44.39 |
| *Open-Source* | | | |
| Llama-4-Maverick-17B | 3,460.76 | 856.62 | **61.29** |
| Qwen2.5-VL-72B | 3,512.20 | 1,236.29 | 57.09 |
| InternVL3-78B | 3,376.52 | 758.71 | 55.07 |
| Llama-4-Scout-17B | 3,247.25 | 826.56 | 51.91 |
| Qwen2.5-VL-32B | **4,960.82** | **1,264.43** | 46.48 |

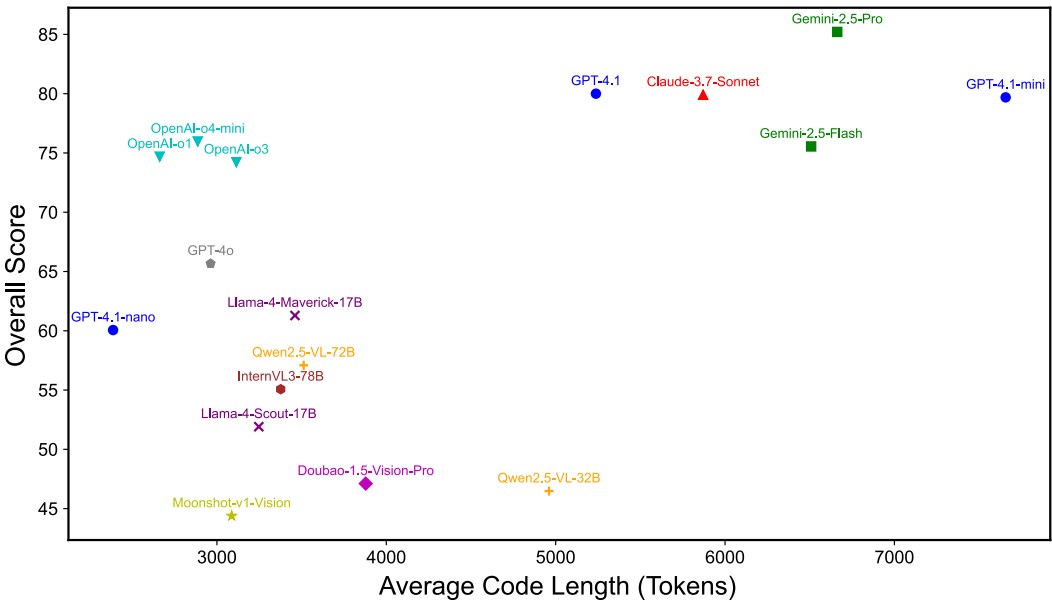

Figure 25: The average code length and overall scores of the evaluated LVLMs.

levels. We adopt a straightforward heuristic based on the number of elements in an infographic chart to define complexity levels. While this simple approach does not consider layout complexity or chart types, it provides a practical and quantifiable basis for stratification, with more sophisticated measures left for future work. Based on this criterion, we split the benchmark into easy (240 infographic charts), medium (169), and hard (91) subsets. Fig. 26 presents the overall LVLM performance across these splits. Gemini-2.5-Pro consistently demonstrates superior performance at all levels of complexity. For most models, performance declines predictably as difficulty

increases. Interestingly, Gemini-2.5-Flash and OpenAI-o3 exhibit improved results on the hard split, suggesting enhanced capability in understanding complex relationships among a large number of elements.

Figure 26: Overall scores of LVLMs across different complexity levels.

*Qualitative examples*. To illustrate model performance on our benchmark, we present five examples in Fig. 27, showing ground-truth charts alongside rendered ones from the generated code of the three top-performing LVLMs (Gemini-2.5-Pro, GPT-4.1, and Claude-3.7-Sonnet). These examples demonstrate the models' relatively strong ability to generate code for infographic charts. Nevertheless, substantial room for improvement remains. Notably, for specialized chart variations (shown in the last four rows of Fig. 27), the models struggle to accurately reproduce spatial arrangement and data encoding.

**Extended experiments**   We conduct two extended experiments to analyze the effects of different prompting methods and the thinking budget parameter on model performance. Both experiments are performed on a randomly sampled subset of 100 infographic charts from the benchmark dataset.

*Prompting methods*.   In addition to the direct prompting method, we evaluate three alternative prompting strategies following prior work (Yang et al., 2025; Si et al., 2025):

- **HintEnhanced**, which guides the model to focus on key aspects of the given infographic chart, such as layout, chart type, and data;
- **TableAug**, which provides auxiliary tabular data to the model;
- **SelfReflection**, where the model is provided the ground-truth chart, previously generated code from direct prompting, and the rendered chart, and is instructed to revise the generated code.

Detailed prompts are available in our code repository[3]. The results, summarized in Table 14, show that the **SelfReflection** method consistently achieves the best performance across models.

*Thinking budget.* Recent LVLMs have incorporated internal reasoning mechanisms that allow them to perform intermediate "thinking" steps prior to final output generation. This process is governed by the thinking budget parameter, which controls the token budget allocated for internal reasoning and is supported only by some

---

[3]https://github.com/ChartGalaxy/ChartGalaxy

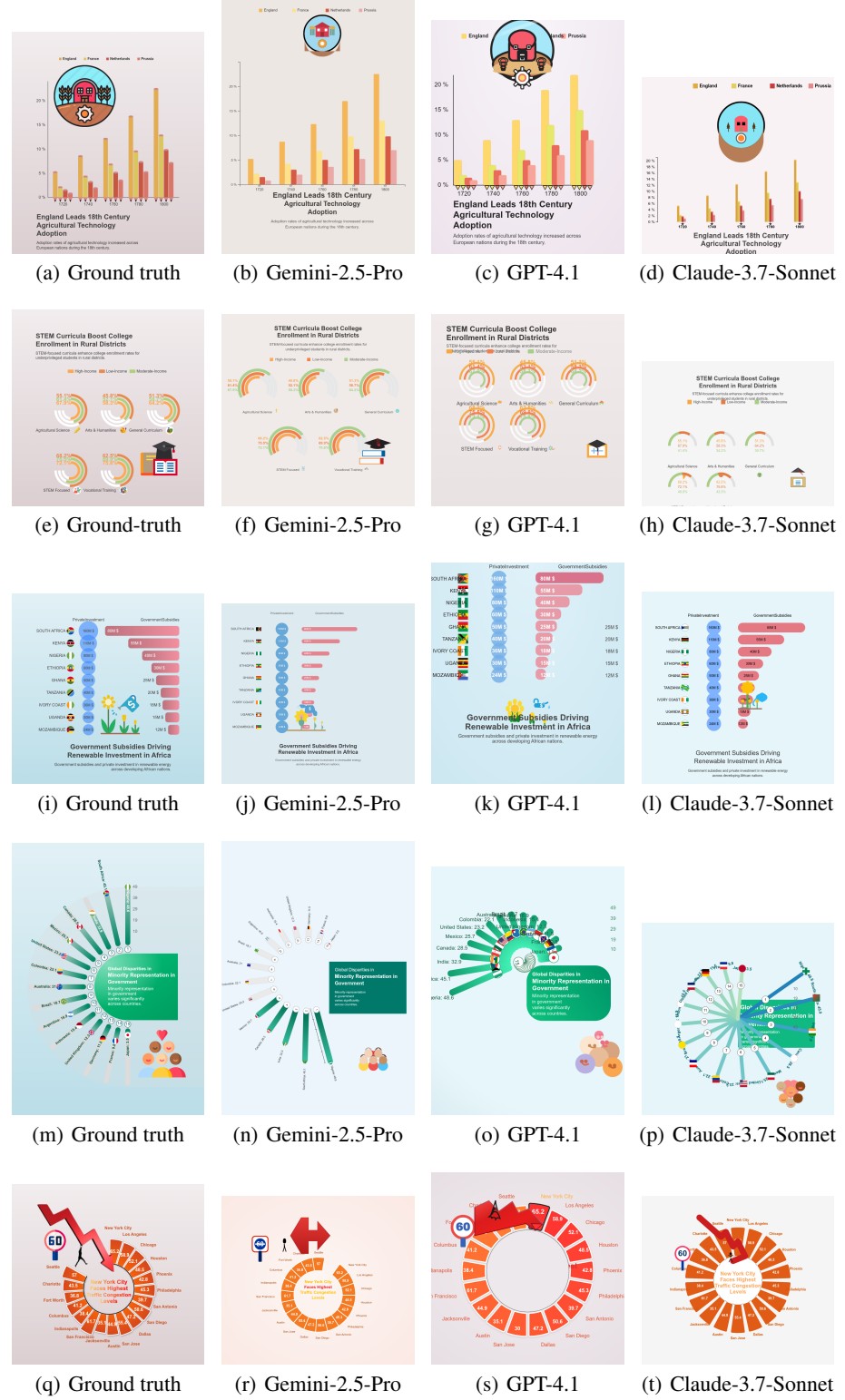

(a) Ground truth     (b) Gemini-2.5-Pro     (c) GPT-4.1     (d) Claude-3.7-Sonnet

(e) Ground-truth     (f) Gemini-2.5-Pro     (g) GPT-4.1     (h) Claude-3.7-Sonnet

(i) Ground truth     (j) Gemini-2.5-Pro     (k) GPT-4.1     (l) Claude-3.7-Sonnet

(m) Ground truth     (n) Gemini-2.5-Pro     (o) GPT-4.1     (p) Claude-3.7-Sonnet

(q) Ground truth     (r) Gemini-2.5-Pro     (s) GPT-4.1     (t) Claude-3.7-Sonnet

Figure 27: Qualitative comparison of the generated infographic charts by Gemini-2.5-Pro, GPT-4.1, and Claude-3.7-Sonnet with the ground-truth ones.

Table 14: Performance comparison of LVLMs with different prompting methods.

| Model | Method | Exec. Rate | Low-Level | | | | | | | High-Level | Overall |
| | | | Area | Text | Image | Color | Position | Size | Avg. | GPT-4o | |
|---|---|---|---|---|---|---|---|---|---|---|---|
| Gemini-2.5-Pro | Direct | 100.00 | **94.88** | **95.62** | 86.95 | 88.46 | **89.53** | 68.09 | 87.25 | 83.59 | 85.42 |
| | HintEnhanced | 100.00 | 94.18 | 95.26 | 86.28 | 87.88 | 88.72 | 68.44 | 86.79 | **84.36** | 85.58 |
| | TableAug | 100.00 | 93.91 | 95.55 | 87.01 | 87.05 | 88.54 | **68.75** | 86.80 | 83.67 | 85.23 |
| | SelfReflection | 100.00 | 93.89 | 95.25 | **88.68** | **88.51** | 89.45 | 68.65 | **87.40** | 84.19 | **85.80** |
| Llama-4-Maverick-17B | Direct | 100.00 | 78.24 | 58.51 | 59.53 | 66.35 | 74.00 | 47.74 | 64.06 | 58.42 | 61.24 |
| | HintEnhanced | 100.00 | 79.31 | 56.13 | 52.06 | 67.01 | 75.14 | 48.47 | 63.02 | 55.56 | 59.29 |
| | TableAug | 100.00 | 81.64 | **61.24** | 61.75 | **68.66** | 74.44 | **48.97** | 66.12 | 59.23 | 62.67 |
| | SelfReflection | 100.00 | 82.44 | 58.93 | **66.19** | 68.11 | **75.45** | 48.93 | **66.67** | **59.46** | **63.06** |

Table 15: Performance comparison of LVLMs with different thinking budgets.

| Model | Thinking Budget | Exec. Rate | Low-Level | | | | | | | High-Level | Overall |
| | | | Area | Text | Image | Color | Position | Size | Avg. | GPT-4o | |
|---|---|---|---|---|---|---|---|---|---|---|---|
| Claude-3.7-Sonnet | 1024 | **100.00** | 92.29 | **94.64** | 78.82 | 85.55 | 87.28 | 66.62 | 84.20 | 76.89 | 80.55 |
| | 4096 | **100.00** | 92.91 | 93.28 | 82.42 | 86.42 | 88.10 | **67.22** | 85.06 | 76.12 | 80.59 |
| | 8192 | **100.00** | **93.78** | 93.85 | **87.12** | **87.24** | **88.40** | **67.22** | **86.27** | **78.71** | **82.49** |
| Gemini-2.5-Flash | 1024 | 97.00 | **90.18** | **89.60** | **78.51** | 80.57 | 84.88 | 62.13 | **80.98** | **75.69** | **78.34** |
| | 4096 | **98.00** | 86.74 | 84.66 | 72.57 | **82.33** | **85.91** | **64.59** | 79.47 | 74.51 | 76.99 |
| | 8192 | 97.00 | 86.45 | 87.19 | 74.05 | 78.31 | 84.09 | 62.82 | 78.82 | 75.55 | 77.19 |

reasoning-enabled models, such as Claude-3.7-Sonnet and Gemini-2.5-Flash. In our earlier experiments, we set the thinking budget to 1024 tokens for these models. For other reasoning models without explicit support for this parameter, we simulate the budget constraint by instructing them to limit internal thinking to 1024 tokens via prompt design. To investigate the effect of the thinking budget in greater detail, we conduct additional experiments varying this parameter for Claude-3.7-Sonnet and Gemini-2.5-Flash. As shown in Table 15, increasing the thinking budget for Claude-3.7-Sonnet leads to clear improvements in the image similarity metric and overall score. Conversely, for Gemini-2.5-Flash, a larger thinking budget correlates with declines in multiple metrics, including the area, text, and image metrics. These contrasting behaviors indicate that the impact of the thinking budget parameter on LVLM performance is model-dependent and warrants further investigation.

**Human Evaluation** To evaluate the reliability of our metric, we conduct a within-subject human study and measure the degree to which the metric's assessments agree with human judgments.

*Participants*. We recruited three participants, all graduate students in computer science with extensive experience in visualization and machine learning. Upon completion, each participant received a $15 compensation.

*Data*. We sampled 600 pairs of generated charts and their corresponding overall scores from the dataset used in Sec. 4.2. Each pair contains two infographic charts produced by different LLMs for the same ground-truth chart. To obtain a representative and well-balanced sample, we examined the full distribution of model scores and score differences across all generated results. We then drew samples to ensure uniform coverage across both score levels and score-difference ranges.

*Task Design*. For each pair, participants were required to choose the chart that is more similar to the ground-truth chart, with both the ground-truth chart and the two options displayed on the screen. The order of the two options was randomized, and participants were unaware of which model produced which chart. When evaluating similarity, participants were instructed to consider both the low-level and high-level aspects outlined in Sec 4.2. Each participant finished the comparison of all 600 pairs, leading to 1,800 total trials.

*Result Analysis*. We analyzed both the consistency between participants to ensure the quality of human preferences and the consistency between the metric and participants to validate the effectiveness of the metric. Three participants reached agreement on 537 out of 600 pairs (89.5%), and Fleiss' Kappa was 0.8598 (almost perfect agreement), indicating strong consistency among participants. The agreement rates between the model and the three participants were 93.17%, 93.17%, and 90.67%, respectively. The corresponding Cohen's Kappa values were 0.8633, 0.8633, and 0.8133 (almost perfect agreement). On average, the agreement rate between the metric and participants was 92.33%, with a Cohen's Kappa of 0.847. This demonstrates that our metric aligns well with human judgment.

## H.3 EXAMPLE-BASED INFOGRAPHIC CHART GENERATION

The study was approved by the Institutional Review Board of the first author's university. In the user study, each participant was compensated with 30 USD for their participation. The study did not involve exposure to emotionally charged, political, or misleading content. Participants were shown infographic charts on neutral topics (e.g., bird population growth, energy sources) and asked to evaluate their quality. No sensitive or personally identifiable data was collected during the study. Participants were fully informed of their rights and were free to withdraw at any time. Fig. 28 illustrates the user study interface using a specific example. Figs. 29-30 present all 30 triplets of infographic charts: one reference, two infographic charts to be rated that are generated by GPT-Image-1 and our method, respectively.

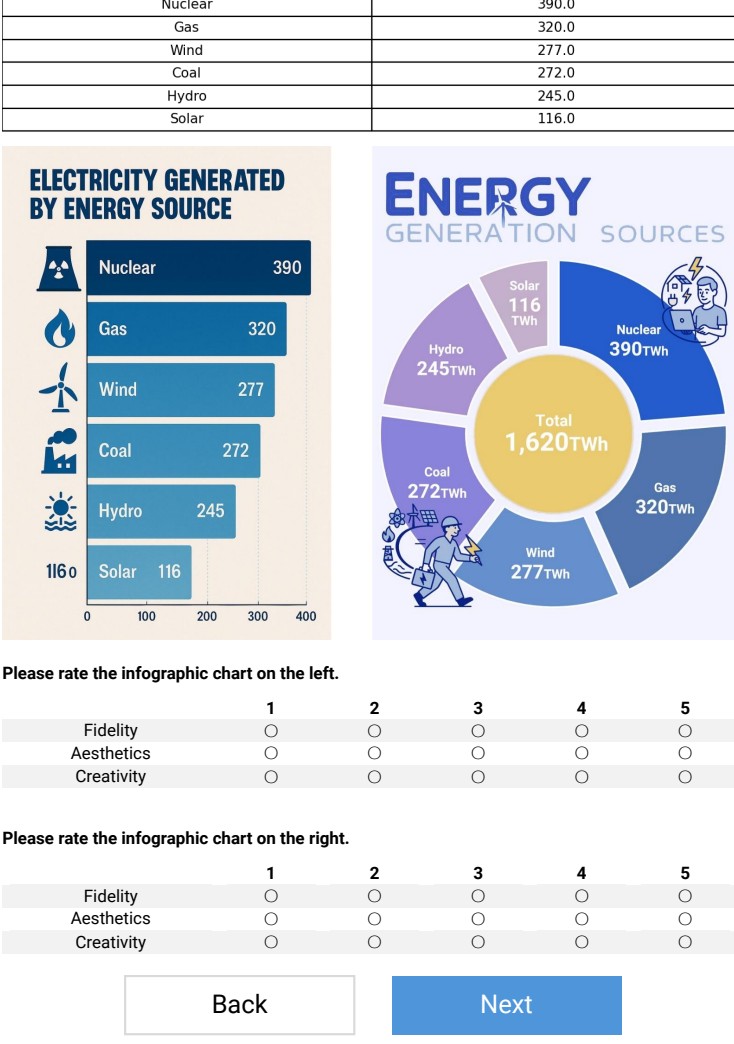

Figure 28: Screenshot of the user study interface. Participants were asked to compare two infographic charts generated by our method and GPT-Image-1 based on the same dataset and reference infographic chart, and rate each chart on three metrics: fidelity, aesthetics, and creativity, using a 5-point Likert scale.

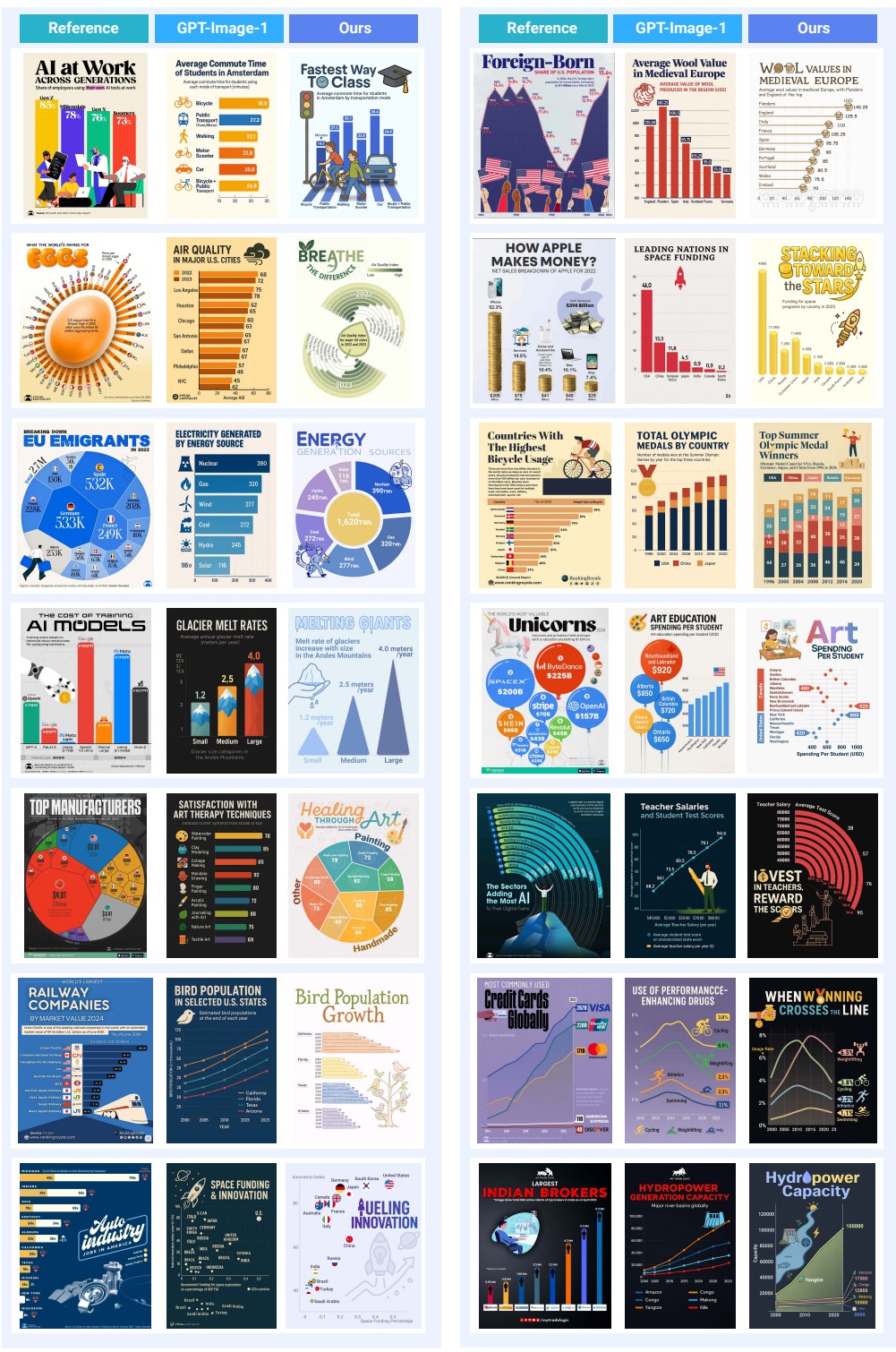

Figure 29: Infographic charts used in the user study: 30 triplets, each comprising a reference, a GPT-generated, and a chart generated by our method (Part 1).

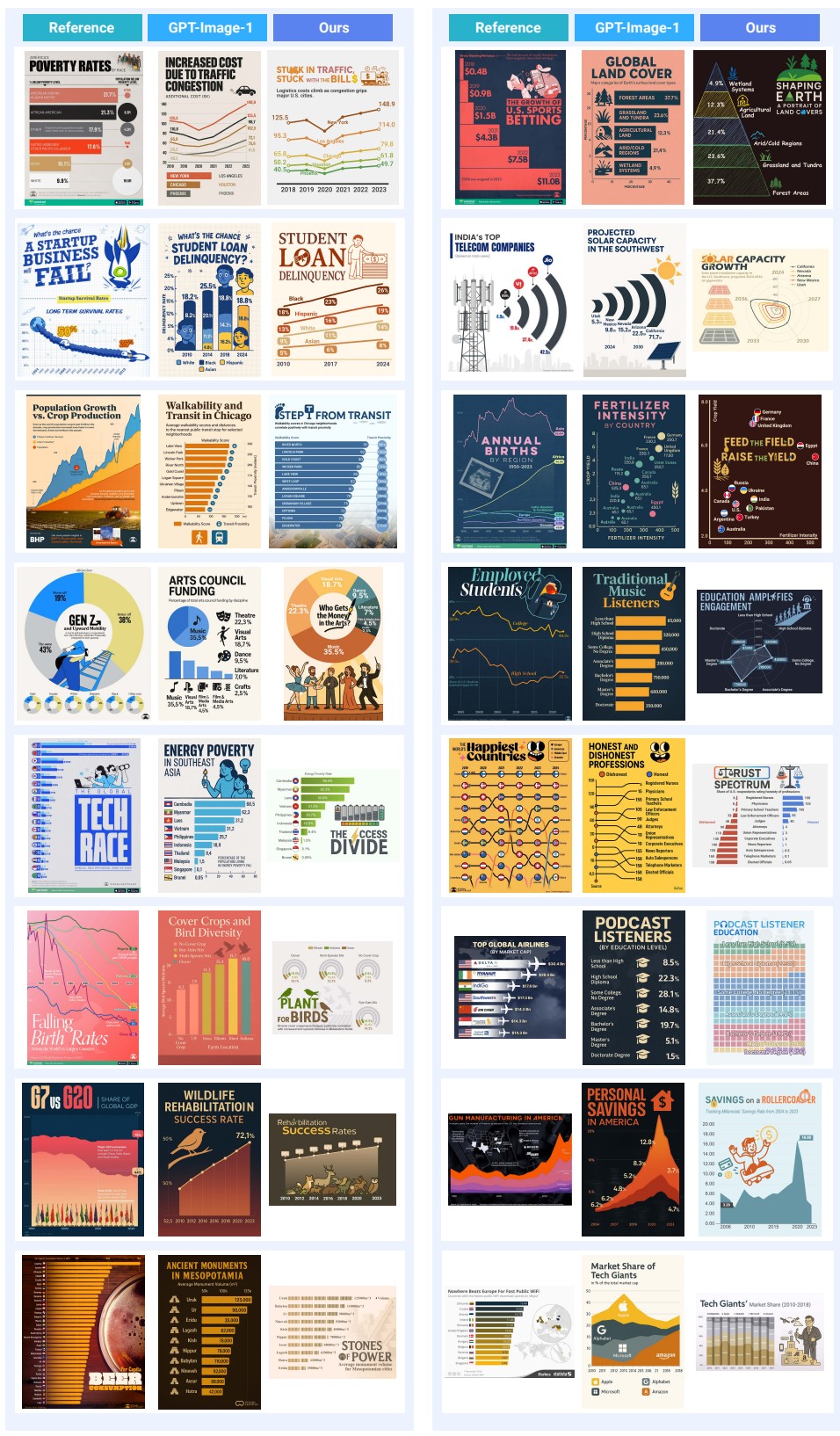

Figure 30: Infographic charts used in the user study: 30 triplets, each comprising a reference, a GPT-generated, and a chart generated by our method (Part 2).

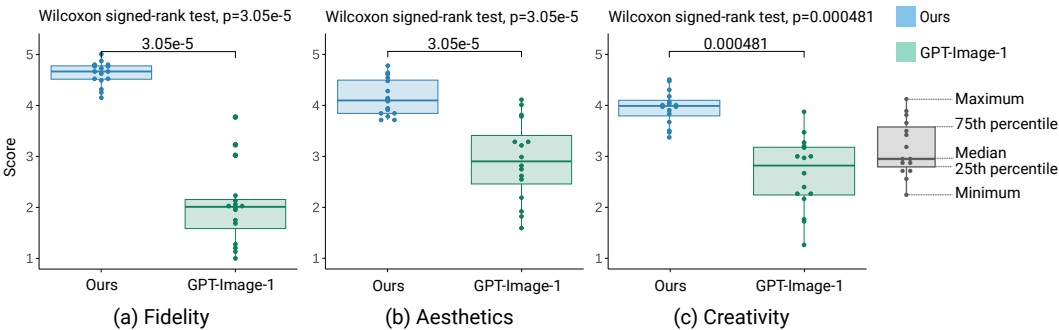

Figure 31: Performance comparison between our method and GPT-Image-1.

Fig. 31 presents the results of the Wilcoxon signed-rank test conducted on the user study data.

*Fidelity*. Ours significantly outperforms GPT-Image-1 in terms of fidelity. The average score for Ours is 4.63, compared to 2.10 for GPT-Image-1. The difference is statistically significant (V = 136, p = 3.05e-5), indicating that our method produces infographic charts that participants perceive as more accurate and faithful to the underlying data. This result can be attributed to our use of the template-based infographic chart creation method, which ensures numerical correctness and consistency between visual encodings and underlying values. In contrast, GPT-Image-1 often exhibits fidelity-related issues, such as incorrect labels, mismatched bar heights, and duplicated or omitted data elements.

*Aesthetics*. In terms of aesthetics, Ours receives an average score of 4.14, while GPT-Image-1 scored 2.90. The difference is statistically significant (V = 136, p = 3.05e-5). This difference is likely due to our method leveraging high-quality layout templates extracted from reference infographic charts, along with carefully designed color palettes to enhance visual harmony. In contrast, GPT-Image-1 occasionally produces unbalanced compositions or relies on overly simplistic and repetitive color palettes, reducing the overall visual appeal.

*Creativity*. Ours also achieves higher scores in creativity, with an average of 3.95 compared to 2.65 for GPT-Image-1. The difference is statistically significant (V = 136, p = 0.000481), suggesting that our method yields designs that are seen as more original and creative. This may be explained by our efforts to incorporate creative elements, such as embedding meaningful icons into the titles, and by exploring less conventional chart types beyond basic bar and line charts. In contrast, GPT-Image-1 tends to generate conventional titles and favors basic chart types, leading to lower perceived creativity.

## H.4 ABLATION STUDY OF THE INFOGRAPHIC CHART CREATION PIPELINE

We conducted an ablation study to assess the contribution of key modules in our generation pipeline. We constructed four ablation variants, each removing one key module and replacing it with a simpler baseline:

1) Title generation: replace the RAG-based method (Sec. 3.4) with direct LLM generation.

2) Image recommendation: replace recommended images with randomly selected ones.

3) Chart rendering: replace programatically implemented charts with chart images generated by GPT-Image-1 using only the chart type and tabular data.

4) Layout optimization: initialize and adjust layouts without enforcing template-specified spatial constraints.

We evaluated fidelity and semantic consistency with 5 participants on 40 pairs of infographics, each consisting of one output from our full pipeline and one from an ablation variant (balanced across settings). Participants rated each infographic on a 5-point Likert scale: fidelity reflects how accurately the chart represents the underlying data, while semantic consistency measures how well the visual content aligns with the intended message.

The results show our full pipeline achieves high fidelity (4.82) and semantic consistency (4.50). The chart rendering module is most critical for fidelity: replacing it with GPT-Image-1 reduces the score to 3.08 due to label errors, misalignment, and incorrect data encoding in LLM-generated charts. Semantic consistency is most affected by layout optimization, image recommendation, and title generation, with scores dropping to 3.12, 2.90, and 3.66, respectively. These results confirm that all four modules play essential roles in the quality and interpretability of the synthesized infographic charts.

Table 16: Ablation results on key modules of the infographic generation pipeline (Mean, [95% CI]).

| Method | Fidelity | Semantic consistency |
|---|---|---|
| Full pipeline | 4.82, [4.71, 4.93] | 4.50, [4.30, 4.70] |
| w/o Title generation | 4.62, [4.46, 4.78] | 3.66, [3.45, 3.87] |
| w/o Image recommendation | 4.56, [4.40, 4.72] | 2.90, [2.70, 3.10] |
| w/o Chart rendering | 3.08, [2.77, 3.39] | 2.58, [2.30, 2.86] |
| w/o Layout optimization | 4.60, [4.44, 4.76] | 3.12, [2.90, 3.34] |

# I    PROMPTS FOR DATA PROCESSING

## I.1    INSTRUCTION DATASET FOR INFOGRAPHIC CHART UNDERSTANDING

This section details the construction of our instruction dataset for infographic chart understanding. We categorize the generated question–answer (QA) pairs into two types: **prompt-based** pairs and **template-based** pairs.

The **prompt-based** pairs are produced through LLM prompting. Specifically, Gemini-2.0-Flash is provided with both the tabular data underlying the chart and the corresponding chart image to produce question–answer pairs. These pairs cover two forms of reasoning: text-based reasoning and visual-element-based reasoning. They further span several categories, including Data Identification (DI), Data Comparison (DC), Data Extraction with Condition (DEC), and Fact Checking (FC). For text-based reasoning, the generated questions directly reference the textual data attributes in the chart. For visual-element-based reasoning, questions are created by first determining whether the data attributes referenced in a generated question correspond to icon representations in the chart image; when such icons are present, the textual attributes are replaced by their respective icons.

The **template-based** pairs are mainly derived by adapting templates from ChartAssistant (Meng et al., 2024). In addition, style-related questions for Visual Understanding (e.g., Style Detection, Visual Encoding Analysis, and Chart Classification) are also template-based, since they are derived directly from the chart styles used in our generation process and do not rely on LLM prompting.

To generate question-answer pairs for Data Identification (DI), where the goal is to identify and report specific data values from the chart based on textual references, the following prompt is used:

> **# DATA**
> {Tabular data}
>
> Follow the data shown in the table strictly; keep answers concise and direct; avoid contradicting the table data.
>
> **# INSTRUCTIONS**
> Generate straightforward Factoid Questions alongside their Corresponding Answers for the given image. The questions should focus on direct identification and extraction of explicit information such as specific data values, labels, titles, axis information, or quantities directly readable from the chart or its textual components. Avoid questions requiring inference, multi-step calculation, or comparison between multiple distinct data points. The Answers should be a number, text label, or a common phrase (Yes, No) found directly in the data. Respond in an Array of JSON objects format with the following keys: (i) Question, and (ii) Answer.
>
> **# EXAMPLES**
> "According to the line graph, what was the 'Population' in 'New York' during '2010'?"
> "What are the units indicated on the Y-axis of the 'Sales Performance' chart?"
> "In the pie chart legend, which category does the color blue represent?"

For Data Comparison (DC) question-answer pairs, which involve comparing different data points or trends, this prompt is utilized:

# DATA
{Tabular data}

Follow the data shown in the table strictly; keep answers concise and direct; avoid contradicting the table data.

# INSTRUCTIONS
Generate some of the most difficult Factoid Questions alongside the Corresponding Answers for the given image. The questions could be related to numerical or visual reasoning. These questions should focus on making comparisons between different data points, categories, or time periods, and identifying significant differences or relationships between multiple elements in the data. The Answers could be a number, text label, or a common phrase (Yes, No). You should respond in an Array of JSON objects format with the following keys: (i) Question, and (ii) Answer.

# EXAMPLES
"Which year had the highest gap between the headline inflation and core inflation?"
"In the years in which the red line was higher than the blue line, which year had the smallest difference between the red and green lines?"
"Which country had the highest increase in the number of cases between Jun and Jul?"
"Which country had the most significant drop in its share of the global hashrate between Aug 2021 and Sep 2021?"

For Data Extraction with Condition (DEC) question-answer pairs, which require extracting specific data points from the chart that meet certain conditions, the following prompt is used:

> **# DATA**
> {Tabular data}
>
> Follow the data shown in the table strictly; keep answers concise and direct; avoid contradicting the table data.
>
> **# INSTRUCTIONS**
> Generate some of the most difficult Factoid Questions alongside the Corresponding Answers for the given image. The questions could be related to numerical or visual reasoning. These questions should focus on identifying trends, making comparisons, finding threshold crossings, analyzing patterns of change, or identifying significant events in the data. The Answers could be a number, text label, or a common phrase (Yes, No). You should respond in an Array of JSON objects format with the following keys: (i) Question, and (ii) Answer.
>
> **# EXAMPLES**
> "Estimate the year in which wind capacity first exceeds 100 gw based on the trend shown in the chart."
> "Determine the airline with the highest increase in ghg emissions from 2008 to 2014."
> "How many times the retail sales growth went below the average annual percentage change from 2002 to 2010 by more than 2%?"
> "Which event caused the most significant drop followed by quick recovery for both lines?"

The following prompt is used to facilitate Data Extraction with Condition (DEC) by generating questions that require calculations based on specific data points extracted from the chart under certain conditions:

> **# DATA**
> {Tabular data}
>
> Follow the data shown in the table strictly; keep answers concise and direct; avoid contradicting the table data.
>
> **# INSTRUCTIONS**
> Generate some of the most difficult Factoid Questions alongside the Corresponding Answers for the given image. The questions could be related to numerical or visual reasoning. These questions should focus on performing calculations based on the data, such as computing percentages, averages, rates of change, or other mathematical operations on the values presented. The Answers could be a number, text label, or a common phrase (Yes, No). You should respond in an Array of JSON objects format with the following keys: (i) Question, and (ii) Answer.
>
> **# EXAMPLES**
> "What is the average growth rate of renewable energy capacity between 2010 and 2015?"
> "If the total investment in 2019 was $100 million, how much would be allocated to the healthcare sector based on the percentage shown?"
> "Calculate the compound annual growth rate (CAGR) of smartphone sales from 2015 to 2020."

The following prompt is used for Data Extraction with Condition (DEC) in the context of hypothetical scenarios, where questions require extrapolations based on data points extracted under specific assumed conditions:

---

**# DATA**
{Tabular data}

Follow the data shown in the table strictly; keep answers concise and direct; avoid contradicting the table data.

**# INSTRUCTIONS**
You are an AI that generates concise and specific hypothetical questions based on chart images. Your task is to analyze the chart and generate a short, data-driven hypothetical question that explores future trends, impacts, or extrapolations based on the data. Avoid adding unnecessary explanations or context like "Based on the chart data..." or "A meaningful hypothetical question could be...". Keep the question focused and directly related to the chart. The question should make an assumption about future trends, impacts, or extrapolations based on the data.

**# EXAMPLES**
"If the average wealth per person in Asia increases by 50%, what will be the new average wealth per person in Asia?"
"If the Construction index had stayed flat at its 2010 level throughout 2011-2013, would the overall Industry index likely have remained below its early 2011 peak?"
"If the Gini index continues to rise at the same rate as it did from 1980 to 2010, what will the Gini index be in 2025?"

---

The following prompt is designed for Fact Checking (FC), generating question-answer pairs that require verifying statements about data by cross-checking information against the visual representation in the chart:

---

**# DATA**
{Tabular data}

Follow the data shown in the table strictly; keep answers concise and direct; avoid contradicting the table data.

**# INSTRUCTIONS**
You are an AI that generates concise and specific factoid questions based on chart images. Analyze the given chart image and generate 2-3 pairs of claims and verdicts about its data. Half of the claims should be supported by the chart's data, while the other half are refuted. Avoid using terms like "rows", "columns", or "elements" from the data table; refer to "chart" or "chart image" instead. If the claim is supported, the verdict should be "True". If the claim is refuted, the verdict should be "False", followed by a brief explanation. The claims should cover comparisons of values or trends, basic statistical values (maximum, minimum, mean, median, mode) without using exact numbers from the chart. Ensure a diverse range of claims addressing various visual aspects of the chart, resulting in 2-3 turns of claims and verdicts. Generate the verdicts/answers without any additional explanation.

**# EXAMPLES**
"Hong Kong consistently has the lowest percentages in at least three categories compared to other East Asian countries in the chart."
"The 4th grade reading pass rate at Auburn Elementary had improvement of about 8% from year 2014 to 2017."
"Toronto has the lowest average technology salary among the cities depicted in the chart."

---

## I.2 BENCHMARKING INFOGRAPHIC CHART CODE GENERATION

**Prompt for instructing LVLMs** We instruct the LVLMs to generate code for the provided infographic chart figure with the following prompt.

---

You are an expert data-visualization engineer and front-end developer.
Your task is to take a chart image and generate a HTML file that, when loaded in a browser, reproduces the chart exactly. The chart must be centered in the viewport.
**# Constraints:**

- No Explanations: Do not include comments, reasoning, or explanatory text. Output only valid HTML with JavaScript code.

**# Technical Requirements**

- **Charting library**: Use D3.js to implement the chart. Write the code to be clean, modular, and easy to understand and modify.

- **Single file output**: Provide one standalone HTML file that includes everything needed to render the chart.

- **Chart fidelity**: Replicate all visual elements—shapes, colors, axes, labels, legends, fonts, line weights, markers—exactly as in the original image.

- **No animations**: The chart must render immediately in its final state.

- **Aspect ratio & sizing**: The chart's content area (including margins, paddings, and plot area) must match the original image's proportions precisely.

- **Image**: Recreate any icon/image content using SVG `<g>` and shapes. Do not use `<image>`, base64 images, or links.

- **Text**: Place all text in `<text>` nodes. Do not use `<tspan>` or nested text structures.

**# Grouping Requirements**
You should use following `class` names for SVG groups with specific semantics:

- `title`: The title area (may contain title, subtitle, and shapes)

- `image`: Each individual icon/image/pictogram (no annotation text, no grouped images)

- `legend`: Legend area

- `axis`: Axis area

**# Output Format**
Return only the following standalone HTML file:

```
<!DOCTYPE html>
<html lang="en">
    <head>
        <meta charset="UTF-8" />
        <title>Recreated Chart</title>
        <!-- <style> -->
    </head>
    <body>
        <div id="chart-container"></div>
        <script src="https://d3js.org/d3.v7.min.js">
        </script>
        <script>
            <!-- Your D3 code here -->
        </script>
    </body>
</html>
```

Only output this file. No comments or explanation. Keep it minimal and strictly within the token limits.

---

**Prompt for the high-level score** We instruct GPT-4o to provide a high-level score with the following prompt.

---

You are an expert evaluator of visualizations. The first image (reference) is an infographic chart rendered from ground truth HTML code, while the second image is an infographic chart rendered from AI-generated HTML code. Your task is to assess how well the AI-generated chart replicates the reference chart.

**# Scoring Criteria (Total: 100 points):**

1. **Data Element (20 points):**
   - Does the AI-generated chart accurately replicate all visual elements that encode data (e.g., bars, points, lines)?
   - The presence of extra or missing data elements that do not match the reference chart will negatively impact the score.
   - Are the positions and lengths/sizes of data elements consistent with the reference, such that the encoded values they represent appear similar?

2. **Layout (20 points):**
   - Does the layout of title, chart area, and images/icons in the AI-generated chart replicate the spatial arrangement of the reference chart?
   - Is alignment of the elements in the generated chart (e.g., left-aligned, right-aligned) consistent with that of the original chart?
   - Were element positions preserved, or did significant misalignments occur?
   - The white space inside the generated chart should be similar to the original, and the aspect ratio of the whole infographic chart should be preserved.

3. **Text (15 points):**
   - Does the AI-generated chart replicate all relevant text content accurately? This includes titles, axis labels, and annotations.

4. **Image (15 points):**
   - Does the AI-generated chart reproduce image elements from the reference infographic chart (e.g., thematic images, embedded icons)?
   - How visually similar are those image elements?

5. **Color (10 points):**
   - Does the AI-generated chart match the original one in terms of colors (background color, line colors, fill colors, text colors, etc.)? Minor differences due to rendering or anti-aliasing can be tolerated if the overall color scheme is preserved.

6. **Validity (20 points):**
   - Is the AI-generated chart clear, readable, and free of overlapping or occluded elements?
   - Are fundamental charting conventions followed? For example: Are axis ticks aligned with data, are icons placed near corresponding data, are colors in legends consistent with the data elements, and is axis-data correspondence preserved?

**# Evaluation Output Format (in JSON):**
Please provide your evaluation in the following format (in valid JSON):

```
{
  "data_element": {
    "score": <integer>,
    "comment": "<your comment>"
  },
```

---

```
  "layout": {
    "score": <integer>,
    "comment": "<your comment>"
  },
  "text": {
    "score": <integer>,
    "comment": "<your comment>"
  },
  "image": {
    "score": <integer>,
    "comment": "<your comment>"
  },
  "color": {
    "score": <integer>,
    "comment": "<your comment>"
  },
  "validity": {
    "score": <integer>,
    "comment": "<your comment>"
  },
  "total_score": <sum of all scores above>
}
```

Be precise and detailed in your comments, and ensure the `total_score` equals the sum of all individual scores.

## I.3 EXAMPLE-BASED INFOGRAPHIC CHART GENERATION

You are an experienced infographic chart designer.
Given the following dataset in JSON format: {Tabular data} and a reference infographic chart image (used as a style guide) {Chart image}, your task is to create a new infographic chart that clearly and creatively visualizes the data.

**# INSTRUCTIONS**
- Maintain overall stylistic consistency with the reference infographic chart, including color scheme, typography, iconography, and visual tone, to ensure a coherent aesthetic. However, adapt the layout, chart types, and visual elements creatively to best suit the structure and insights of the new dataset.
- Prioritize effective communication of the new data over replicating the original design.
- Incorporate visual storytelling elements, such as icons, labels, contrast, or scale, to highlight key patterns or contrasts in the data.
- Include a clear, well-designed title that matches the tone and aesthetic of the reference.
- You may choose a light or dark background — whatever best fits the visual narrative and legibility.
- Legends, axes, and any necessary annotations should be present and styled consistently.

**# OUTPUT FORMAT**
- A single high-resolution infographic chart image (portrait or square format)
- All text and numbers should be **fully readable**
- The image should be **self-explanatory** — no external explanation should be needed
- The style should be **clean, professional, and ready for publication**
Avoid any explanation outside the visual — the final image should be self-explanatory and visually engaging.

## J THE USE OF LLMS

LLMs were used solely to aid or polish writing; they were not used for research ideation, analysis, or any other part of the manuscript.

