# OpenReview forum: "ChartGalaxy: A Dataset for Infographic Chart Understanding and Generation"
_ICLR.cc/2026/Conference — ICLR 2026 Poster_

### Official Review · Reviewer_d34b · 2025-10-29

**Soundness:** 2
**Presentation:** 3
**Contribution:** 4
**Rating:** 6
**Confidence:** 4

**Summary:**

This paper introduces ChartGalaxy, a large-scale dataset comprising million-scale synthetic and 61K real infographic charts paired with underlying data tables, aimed at advancing multimodal reasoning and generation in large vision-language models (LVLMs). The dataset is constructed through a two-stage process. The authors demonstrate the dataset's utility via three applications: fine-tuning LVLMs for improved infographic understanding, benchmarking 17 LVLMs on code generation for infographics (with Gemini-1.5-Pro topping), and an example-based generation method that outperforms GPT-4V in user studies on fidelity, aesthetics, and creativity.

**Strengths:**

- The paper addresses a clear and important gap on infographic data. The community has largely focused on plain charts, while complex infographics, which are common in media, business, and education, remain a major challenge for LVLMs and no large-scale high-quality data exists.

- The pipeline for generating synthetic data is a major strength. The authors incorporate D3, which offers much more visualization options than conventional approaches relying on matplotlib or seaborn. The use of template-based generation approach is quite clever and neat.

- The authors do not just present a dataset; they thoroughly demonstrate its utility. All three applications are well-conceived and executed through chart understanding, code generation, and chart generation.

**Weaknesses:**

- The authors did not discuss the performance of the object detection model (InternImage+DINO) used to parse real charts.


- While outperforming baselines in applications, direct comparisons to baselines are missing; e.g., no fine-tuning ablation using only prior datasets. The authors can compare relevant training datasets such as [1][2][3] as baselines. These additional experiments would enhance the soundness of the paper.


[1] Liu et al., MMC: Advancing Multimodal Chart Understanding with Large-scale Instruction Tuning. NAACL 2024

[2] Huang et al., Why Vision Language Models Struggle with Visual Arithmetic? Towards Enhanced Chart and Geometry Understanding. ACL 2025 Findings

[3] Masry et al., ChartGemma: Visual Instruction-tuning for Chart Reasoning in the Wild. COLING 2025 Industry Track

**Questions:**

In section 3.4, you first mention you use gemini-2.0-flash to select templates. However, in the next paragraphs about layout optimization, you mentioned you filter templates based on some criteria. These two parts seem conflicting. Could you elaborate more?

---

> ### Author Response · Authors · 2025-11-24
> **Response to Review**
>
> We sincerely thank you for your constructive suggestions and encouraging feedback.
> We have carefully addressed concerns regarding the performance of the object detection model, ablation studies using previous datasets (Sec. 4.1), and layout template selection/filtering.
>
> > **Q1**: The authors did not discuss the performance of the object detection model (InternImage+DINO) used to parse real charts.
> >
> >
> **A1**:
> Per your suggestion, we evaluate our object detection model on an evaluation set consisting of 1,500 real infographic charts annotated by three co-authors from our dataset.
> On the evaluation set, the model achieves 0.71 mAP,
> demonstrating strong capability in extracting relevant elements and layouts from real infographic charts.
>
> > **Q2**: While outperforming baselines in applications, direct comparisons to baselines are missing; e.g., no fine-tuning ablation using only prior datasets. The authors can compare relevant training datasets such as [1][2][3] as baselines. These additional experiments would enhance the soundness of the paper.
> >
> **A2**:
> Per your suggestion, given the limited time available for additional experiments, we prioritized the two most recent datasets you mentioned, CogAlign [2] and ChartGemma [3]. We fine-tuned on each dataset using the same LoRA adapters, hyperparameters, and training setup as in Sec. 4.1, training for one epoch.
> We used the same two models as in Sec. 4.1 (InternVL3-8B and Qwen2.5-VL-7B) and evaluated performance on ChartQAPro and InfographicVQA. The results are shown below.
>
> |Model|ChartQAPro|InfographicVQA|
> |-----------------|----------|--------------:|
> |InternVL3-8B|38.15|76.19|
> |+ ChartGemma|38.60|76.23|
> |+ CogAlign|37.24|74.19|
> |+ ChartGalaxy|**44.13**|**79.99**|
> |Qwen2.5-VL-7B|37.97|78.59|
> |+ ChartGemma|38.26|78.94|
> |+ CogAlign|37.48|77.73|
> |+ ChartGalaxy|**41.56**|**83.03**|
>
> ChartGalaxy yields the largest performance improvements for both models across both benchmarks.
>
> ChartGemma brings relatively modest improvements, likely because its design, optimized for tasks such as chart summarization, chart-to-Markdown extraction, and program-aided solving, aligns only loosely with the complex analytical reasoning required by ChartQAPro and the visual extraction required of InfographicVQA.
>
> CogAlign results in a performance drop, indicating negative transfer from task discrepancies. It emphasizes low-level geometric cues typical of visual arithmetic, and the resulting gradients likely dominate the shared LoRA updates, thereby suppressing the higher-level reasoning capabilities required for complex QA.
> This observation is consistent with the findings of Li et al. [4], who highlight that the limited trainable capacity of LoRA-based methods can hinder cross-task generalization under divergent task distributions.
>
> We include these additional analyses in Appendix H.1.
>
> > **Q3**: In section 3.4, you first mention you use gemini-2.0-flash to select templates. However, in the next paragraphs about layout optimization, you mentioned you filter templates based on some criteria. These two parts seem conflicting. Could you elaborate more?
> >
> >
> **A3**:
> We would like to clarify that in Sec. 3.4, gemini-2.0-flash is used to select the chart types \& variations, not the layout templates. The next stage of layout optimization operates on layout templates. Specifically, we filter out layout templates that are incompatible with the generated elements (e.g., due to unintended overlaps), and then optimize the layout by reducing unnecessary white space and adjusting element positions. Finally, we select the template with the highest ink ratio.
>
> ### References
>
> [1] F. Liu, et al., "MMC: Advancing Multimodal Chart Understanding with Large-scale Instruction Tuning," NAACL, 2024.
>
> [2] K. Huang, et al., "Why Vision Language Models Struggle with Visual Arithmetic? Towards Enhanced Chart and Geometry Understanding," ACL Findings, 2025.
>
> [3] A. Masry, et al., "ChartGemma: Visual Instruction-tuning for Chart Reasoning in the Wild," COLING Industry Track, 2025.
>
> [4] D. Li, et al., "Mixlora: Enhancing large language models fine-tuning with lora-based mixture of experts," Arxiv, 2024.

---

> > ### Comment · Reviewer_d34b · 2025-11-25
> >
> > Thank you for the response. The rebuttal addressed all my concerns and I have increased the soundness score and rating.

---

> > > ### Author Response · Authors · 2025-11-25
> > > **Thank You Very Much for Your Feedback**
> > >
> > > We sincerely thank you for your feedback! We are glad to hear that our responses address your concerns. We deeply appreciate your time and effort in reviewing our paper and providing insightful and constructive comments, which are valuable for improving our paper. If you have any further suggestions or questions, we will make every effort to address them.

---

### Official Review · Reviewer_NBfT · 2025-10-31

**Soundness:** 4
**Presentation:** 3
**Contribution:** 4
**Rating:** 8
**Confidence:** 5

**Summary:**

This manuscript introduces ChartGalaxy, a large-scale dataset designed for infographic chart understanding and generation. The dataset combines 61,833 real infographic charts collected from 18 publicly available sources with 1.7 million synthetic infographic charts generated through an inductive, template-based synthesis pipeline. Each chart is paired with a tabular dataset, enabling multimodal learning. The authors demonstrate ChartGalaxy’s utility through three applications: (1) fine-tuning LVLMs for infographic chart understanding, (2) benchmarking infographic chart code generation, and (3) example-based infographic chart generation evaluated via a user study.

**Strengths:**

S1: Comprehensive and large-scale dataset. The dataset includes over 1.76 million infographic charts paired with tabular data, exceeding the scale of previous datasets. This scope enables training LVLMs for realistic infographic scenarios, supporting broad generalization. The inclusion of 75 chart types, 440 chart variations, and 68 layout templates reflects high visual and structural diversity. The dual-source construction (real + synthetic) provides both authenticity and scalability, addressing prior dataset limitations.

S2: Well-defined inductive synthesis pipeline. The human-in-the-loop design for extracting and expanding layout templates demonstrates methodological soundness. The pipeline’s iterative detection and clustering stages ensure diversity. The layout optimization step formalizes the spatial packing problem, providing a principled way to ensure high visual quality. The combination of data-to-chart mapping rules, adaptive sampling for variation balance, and semantically resonant color selection enhances the realism of synthetic charts.

S3: Clear evidence of downstream utility. Fine-tuning LVLMs with ChartGalaxy yields large gains on InfographicVQA and ChartQAPro and even larger gains on the authors’ evaluation set, showing measurable improvement. The code generation benchmark evaluates 17 LVLMs with reproducible metrics (SVG-level and GPT-4o-based judgments), positioning the dataset as an evaluation standard. The example-based generation experiment includes a controlled user study showing statistically significant improvements in fidelity, aesthetics, and creativity.

**Weaknesses:**

W1:
The manuscript would benefit from a clearer definition of what qualifies as an infographic chart and a more concrete explanation of how it differs from plain charts in terms of reasoning challenges. The current description in the Introduction could be made more specific, perhaps with brief examples or a sharper comparison to existing VQA or chart QA tasks to better highlight the unique difficulty of infographic chart understanding.

W2:
Table 1 shows that models achieve higher accuracy on InfographicVQA than on ChartQAPro. It would be helpful to include a short discussion on potential factors contributing to this gap, such as differences in question design, dataset composition, or task difficulty. Clarifying this would better support the manuscript’s motivation and help readers interpret the reported results.

W3:
The synthetic infographic chart generation pipeline is a valuable contribution. To further strengthen the work, it could be beneficial to include a small quantitative or qualitative assessment of the fidelity and usefulness of the generated charts beyond the downstream fine-tuning results. Even a brief ablation on key components, such as layout optimization, would provide additional insight.

W4:
The dataset and benchmark are timely and impactful. Adding more analysis or reflections on model behavior, enabled by this dataset, could enhance the contribution for the ICLR audience, which often values methodological insight alongside dataset and system advances.

**Questions:**

Q1: Shouldn’t the original intention and practice of infographic charts, such as adding explanatory text, icons, and descriptive narrative structures as clues, be to make it easier to understand? Is there any empirical evidence that infographic charts are more difficult for VLMs to understand?

Q2: W2.

Q3: W3.

Q4: The setup of the GPT-Image-1 baseline in the example-based infographic chart generation experiment is insufficiently documented. How was GPT-Image-1 prompted? Any layout or structural guidance supplied?

---

> ### Author Response · Authors · 2025-11-24
> **Response to Review (1/2)**
>
> We sincerely thank you for your valuable feedback and thoughtful suggestions.
> We have carefully addressed the concerns regarding the definition and reasoning challenges of infographic charts, the ablation study of the chart generation pipeline, and the analysis model behavior.
>
> > **Q1**: The manuscript would benefit from a clearer definition of what qualifies as an infographic chart and a more concrete explanation of how it differs from plain charts in terms of reasoning challenges. The current description in the Introduction could be made more specific, perhaps with brief examples or a sharper comparison to existing VQA or chart QA tasks to better highlight the unique difficulty of infographic chart understanding.
> >
> **A1**:
> We have revised Sec. 1 to provide a clearer definition of infographic charts and a more concrete explanation of their unique reasoning challenges compared to plain charts and existing chart QA tasks.
> Specifically, we refine our **infographic chart** definition to:
> "By integrating imagery (e.g., icons and metaphorical graphics) alongside charts and textual information, infographic charts present abstract data in a manner that is both engaging and easy to understand, making data more accessible to a broad audience."
> Consequently, unlike plain charts, understanding infographic charts requires models to jointly interpret decorative and metaphorical elements, as well as their relationship to the underlying data. Additionally,
> we add an example for these reasoning challenges in Line 50.
>
> > **Q2**: Table 1 shows that models achieve higher accuracy on InfographicVQA than on ChartQAPro. It would be helpful to include a short discussion on potential factors contributing to this gap, such as differences in question design, dataset composition, or task difficulty...
> >
> **A2**:
> As we can only infer from the design characteristics of the two benchmarks, we attribute the performance gap between InfographicVQA [2] and ChartQAPro [3] to differences in task and question design.
> InfographicVQA mainly focuses on visual extraction tasks, while ChartQAPro emphasizes "complex analytical reasoning," introducing more challenging question types such as "hypothetical questions," "conversational questions," and "unanswerable questions" that require reasoning beyond simple data retrieval.
> This distinction is also reflected in human performance reported in their original papers: InfographicVQA reaches 95.70\% accuracy, while ChartQAPro only achieves 85.02\%.
> Overall, these results suggest that current models handle recognition well, but analytical reasoning remains a major gap.
> We have added the discussion in Appendix H.1.
>
> > **Q3**: The synthetic infographic chart generation pipeline is a valuable contribution. To further strengthen the work, it could be beneficial to include a small quantitative or qualitative assessment of the fidelity and usefulness of the generated charts...
> >
> **A3**:
> Per your suggestion, we have conducted an ablation study to assess the contribution of key modules in our generation pipeline.
> We constructed four ablation variants, each removing one key module and replacing it with a simpler baseline:
>
> 1) Title generation: replace the RAG-based method (Sec. 3.4) with direct LLM generation.
>
> 2) Image recommendation: replace recommended images with randomly selected ones.
>
> 3) Chart rendering: replace programatically implemented charts with chart images generated by GPT-Image-1 using only the chart type and tabular data.
>
> 4) Layout optimization: initialize and adjust layouts without enforcing template-specified spatial constraints.
>
> We evaluated fidelity and semantic consistency with 5 participants on 40 pairs of infographics, each consisting of one output from our full pipeline and one from an ablation variant (balanced across settings).
> Participants rated each infographic on a 5-point Likert scale: fidelity reflects how accurately the chart represents the underlying data, while semantic consistency measures how well the visual content aligns with the intended message.
>
> The results show our full pipeline achieves high fidelity (4.82) and semantic consistency (4.50).
> The chart rendering module is most critical for fidelity: replacing it with GPT-Image-1 reduces the score to 3.08 due to label errors, misalignment, and incorrect data encoding in LLM-generated charts.
> Semantic consistency is most affected by layout optimization, image recommendation, and title generation, with scores dropping to 3.12, 2.90, and 3.66, respectively.
> These results confirm that all four modules play essential roles in the quality and interpretability of the synthesized infographic charts.
> We have added these results to Appendix H.4.
>
> |Method|Fidelity|Semantic consistency|
> |-|-|-|
> |Ours|4.82|4.50|
> |w/o Title generation|4.62|3.66|
> |w/o Image recommendation|4.56|2.90|
> |w/o Chart rendering|3.08|2.58|
> |w/o Layout optimization|4.60|3.12|

---

> ### Author Response · Authors · 2025-11-24
> **Response to Review (2/2)**
>
> > **Q4**: The dataset and benchmark are timely and impactful. Adding more analysis or reflections on model behavior, enabled by this dataset, could enhance the contribution for the ICLR audience, which often values methodological insight alongside dataset and system advances.
> >
> >
> **A4**: We have added discussion in Appendix H.1 and H.2 to provide further insights enabled by our dataset.
>
> **Limitations of pre-trained models in visual design literacy**.
> As shown in Table 2, fine-tuning on our dataset leads to notably larger improvements in visual understanding tasks, such as Style Detection (SD) and Visual Encoding Analysis (VEA), with average gains of +42.5\% for InternVL3 and +42.3\% for Qwen2.5-VL.
> In contrast, the improvements on the remaining tasks are more modest (+20.2\% for InternVL3 and +17.2\% for Qwen2.5-VL).
> This gap indicates that existing pre-training pipelines pay relatively little attention to visual design literacy: understanding how visual attributes like color, style, and other encodings convey information.
> Our dataset helps compensate for this missing capability by providing tasks that require such understanding.
> Incorporating such tasks and data into pre-training could meaningfully strengthen models' visual design literacy.
>
> **Trade-off between code generation performance and code length**.
> As shown in Figure 25 in Appendix H.2, we observe a trade-off between overall code generation performance and code length across different models.
> In general, models generating longer code, which often specify more detailed elements, tend to achieve higher overall scores.
> For example, Gemini-2.5-Pro, which generates the second-longest code (about 7,000 tokens), achieves the highest overall score.
> In contrast, OpenAI's o-series models (o4-mini, o1, and o3) show relatively strong performance despite generating shorter code (about 3,000 tokens), indicating their ability to convey high-quality outputs with fewer tokens.
> This suggests that their code is more concise, capturing the essential structure and visual details with fewer tokens.
> Taken together, this analysis presents a trade-off in real-world applications: whether to prioritize higher performance with longer code (e.g., Gemini-2.5-Pro and GPT-4.1-mini) or slightly lower performance with shorter code (o4-mini, o1, and o3).
>
> > **Q5**: Shouldn’t the original intention and practice of infographic charts, such as adding explanatory text, icons, and descriptive narrative structures as clues, be to make it easier to understand? Is there any empirical evidence that infographic charts are more difficult for VLMs to understand?
> >
> >
> **A5**:
> A recent study by Xie et al. [1] compared VLMs' understanding of paired infographic charts and plain charts constructed from the same data.
> They found that VLMs performed worse on infographic charts, largely due to the increasing number of visual elements that increase perceptual complexity and the models’ limited ability to interpret visual metaphors.
> Thus, although infographic charts include explanatory text, icons, and narrative structures intended to aid the understanding, the benefits are mainly for *human readers*, not for VLMs.
>
> > **Q6**: The setup of the GPT-Image-1 baseline in the example-based infographic chart generation experiment is insufficiently documented. How was GPT-Image-1 prompted? Any layout or structural guidance supplied?
> >
> >
> **A6**:
> The prompt for GPT-Image-1 is provided in Appendix I.3.
> In our setup, GPT-Image-1 is prompted with a reference infographic chart and a new data table, and is instructed to adapt the layout and style of the reference to the new data.
> No additional layout or structural guidance is supplied.
>
> ### References
>
> [1] T. Xie, et al., "InfoChartQA: A benchmark for multimodal question answering on infographic charts," NeurIPS, 2025.
>
> [2] M. Mathew, et al., "InfographicVQA," WACV, 2022.
>
> [3] A. Masry, et al., "ChartQAPro: A more diverse and challenging benchmark for chart question answering," Arxiv, 2025.

---

### Official Review · Reviewer_Z9By · 2025-10-31

**Soundness:** 2
**Presentation:** 3
**Contribution:** 3
**Rating:** 6
**Confidence:** 3

**Summary:**

This paper constructs a large-scale dataset for infographic charts.
While previous infographic datasets were limited in scale, this work builds a massive dataset of 1.7 million synthetic samples, combining web-collected real data with diverse layout templates.
Using the constructed dataset, the authors conduct experiments on three tasks: chart understanding, code generation, and example-based infographic chart generation.

**Strengths:**

- 1. Templates are carefully extracted from real data and utilized for synthetic data generation, resulting in a diverse set of samples.
- 2. A large number of evaluation experiments are conducted in a thorough and detailed manner.
- 3. This work provides a large-scale dataset in the infographic domain, where available data has been scarce until now.

**Weaknesses:**

There are some unclear points regarding the details and procedures of the experiments.
- 1. In Section 3.3, the paper describes the extraction of layout templates. Could you clarify the format in which these templates are stored? While Figure 3 presents visual examples of the template images, it would be helpful to know whether they also contain information such as bounding boxes or other structural annotations.
- 2. In Section 3.4, which discusses Element Generation, could you please clarify the procedure for chart generation? Specifically, are the charts produced using predefined D3.js code that is associated with each layout template, or are they dynamically generated through the use of LLMs or other generative models?
- 3. Does the proposed method described in Section 4.3 generate charts by specifically utilizing the Element Generation/Recommendation and Layout Optimization processes presented in Section 3.4?

**Questions:**

- 1. In Section 4.2 on the infographic chart generation task, only zero-shot evaluation is conducted. Is it also possible to fine-tune the model using the synthetic data?
- 2. In the same Section 4.2, why are the evaluation samples drawn only from the synthetic subset? Since the synthetic data itself was generated by combining chart types and variations using D3.js with LLM-based synthesis, wouldn’t using only LLM-generated data for evaluation introduce a potential bias that makes the generation task easier for the model?

---

> ### Author Response · Authors · 2025-11-24
> **Response to Review (1/2)**
>
> We appreciate your feedback and constructive suggestions.
> We have carefully addressed the concerns regarding layout template representation, the chart generation procedure, example-based generation (Sec. 4.3), fine-tuning with synthetic data (Sec. 4.2), and the use of synthetic data (Sec. 4.2).
>
> > **Q1**: In Section 3.3, the paper describes the extraction of layout templates. Could you clarify the format in which these templates are stored? While Figure 3 presents visual examples of the template images, it would be helpful to know whether they also contain information such as bounding boxes or other structural annotations.
> >
> >
> **A1**:
> We would like to clarify that the layout templates are stored in the JSON format, which describes (1) the existence of elements (chart, image, and text), (2) their pairwise positional relationships, and (3) overlap relationships between their bounding boxes.
> Information such as bounding boxes is not recorded.
> A concrete example is provided below.
>
> ```
> {
>   # Element Existence
>   "title": "yes",     # "yes" or "no"
>   "image": "yes",
>   "chart": "yes",
>
>   # Position Relationships
>   "title-to-chart": "left-top",   # Top, bottom, and 7 other options
>   "image-to-chart": "right-top",
>   "title-to-image": "left",
>
>   # Overlap Relationships Between Bounding Boxes
>   "chart-title-overlap": "no",    # "yes" or "no"
>   "chart-image-overlap": "no",
>   "title-image-overlap": "no"
> }
> ```
>
> > **Q2**: In Section 3.4, which discusses Element Generation, could you please clarify the procedure for chart generation? Specifically, are the charts produced using predefined D3.js code that is associated with each layout template, or are they dynamically generated through the use of LLMs or other generative models?
> >
> >
> **A2**:
> To clarify, the charts are generated purely using predefined, manually crafted D3.js scripts, rather than by LLMs or other generative models.
> We programmatically implemented all 440 chart variations in D3.js.
>
> These D3.js chart scripts are independent of the layout templates.
> Their purpose is to render the charts that later serve as input to the layout optimization stage.
> Specifically, our infographic creation pipeline is designed to decouple Element Generation/Recommendation from Layout Optimization:
>
> 1. Element Generation/Recommendation:
> We first produce the content elements (charts, images, text) based on the input data.
> Chart rendering specifically uses the manually crafted D3.js scripts mentioned above.
>
> 2. Layout Optimization:
> We then determine how these elements are arranged (i.e., their sizes and positions) according to the selected layout template.
>
> > **Q3**: Does the proposed method described in Section 4.3 generate charts by specifically utilizing the Element Generation/Recommendation and Layout Optimization processes presented in Section 3.4?
> >
> >
> **A3**:
>
> The proposed method in Sec. 4.3 uses the same Element Generation/Recommendation method presented in Sec. 3.4, but the Layout Optimization method differs slightly.
> Specifically, in Sec. 4.3, we extract the layout from the reference image with its elements' bounding boxes, which is used in the Layout Optimization to maintain layout similarity between the reference and the generated infographic.
> In Sec. 3.4, we only use predefined layout templates without explicit element coordinates.
> We have clarified this in Sec. 4.3, "We then generate/recommend new elements from the provided tabular data (Sec. 3.4), initialize their positions using the extracted positional information from the example, and refine the element positions through the layout optimization module (Sec. 3.4)."

---

> ### Author Response · Authors · 2025-11-24
> **Response to Review (2/2)**
>
> > **Q4**: In Section 4.2 on the infographic chart generation task, only zero-shot evaluation is conducted. Is it also possible to fine-tune the model using the synthetic data?
> >
> >
> **A4**:
> We agree that fine-tuning LLMs with ChartGalaxy is a promising direction.
> However, this would require constructing a large set of high-quality pairs of cleaned code and infographic charts, which demands substantial additional effort on code cleanup.
> Moreover, the long code sequences (Table 13 in Appendix H.2) also pose great challenges for open-source small models, which require further technical contributions to the model design and training processes for better performance.
> We leave this as an important future work.
>
> > **Q5**: In the same Section 4.2, why are the evaluation samples drawn only from the synthetic subset? Since the synthetic data itself was generated by combining chart types and variations using D3.js with LLM-based synthesis, wouldn’t using only LLM-generated data for evaluation introduce a potential bias that makes the generation task easier for the model?
> >
> >
> **A5**:
> We would like to clarify that we evaluate on the synthetic subset because only the synthetic infographics provide the metadata necessary for fine-grained, low-level similarity assessment.
> Our benchmark requires detailed annotations, including element colors, sizes, positions, and other attributes, which are extremely difficult to reliably extract from real infographics.
> In contrast, synthetic infographics are rendered via D3.js code into SVGs with complete, fine-grained metadata (e.g., bounding boxes and attributes for all elements), enabling precise low-level evaluation that would not be feasible with real charts.
>
> Regarding the concern about potential bias from LLM-generated data, our synthetic data is grounded in real infographic design patterns and programmatically constructed, not primarily synthesized by LLMs.
> LLM involvement is restricted to peripheral components, including:
> (i) assisting with data extraction from a subset of real infographic charts, where results are verified via multi-model agreement and human checking;
> (ii) generating a relatively small portion of synthetic tables (98k), compared with a substantially larger set of real tables (200k); and
> (iii) producing minor textual elements such as titles and subtitles.
> Given that LLM contributions are limited and tightly controlled, the risk of introducing biases that would make the task artificially easier is greatly mitigated.

---

### Official Review · Reviewer_6nvx · 2025-11-01

**Soundness:** 3
**Presentation:** 2
**Contribution:** 3
**Rating:** 6
**Confidence:** 3

**Summary:**

This paper presents ChartGalaxy, a dataset for chart understanding and generation. It contains over 1 million infographics, some of them sourced from the real world and some of them auto-generated. Authors show three use cases with the dataset: improving infographics QA, benchmarking infographics generation, and enabling example-based infographic generation.

**Strengths:**

+ The dataset size is very large, and it seems to span many chart types and designs.
+ The dataset quality seems to be high.
+ The use cases are in general convincing.
+ The supplemental materials are a very helpful addition for presentation.

**Weaknesses:**

- I am a bit lost in reading the curation of machine-generated synthetic infographics. I can see a lot of work went into this, but I feel like it is lacking a big picture. Please see my first question. The other parts of the paper all seem pretty easy to follow.
- Huge improvements on questions based on ChartGalaxy is probably not that unsurprising given the vast majority of infographics in the QA set are based on synthetic charts with fixed templates. Finetuning is expected to help a lot here. Also I believe the QA set is not yet published so we do cannot gauge QA set quality.
- The code generation metric seems a bit questionable. For example, GPT-4.1-mini ranks higher than o4-mini, o1, o3, and GPT-4o. It is better to have a validation process where humans independently score each chart without seeing the model generating it and establish some sort of inter-rater reliability with scores produced by your metric.

**Questions:**

- The authors say templates capture the spatial relation of chart elements, but how are templates represented? Is each template a D3 code snippet that can take in different data tables and images as input? Do you use a VLM to generate these templates? And how does layout optimization work with these templates?
- All Chit Chart charts seem to not wrong urls. They all have https://chitchart.com/ without further specification (e.g., 00060283, 00059496). Could the authors check if this is the case please?
- The authors say "Moreover, we construct an independent, human-verified evaluation set containing 2,176 synthetic charts with 4,975 question-answer pairs." Are these a sampled and verified subset of all VLM-generated questions?

**Details Of Ethics Concerns:**

The paper crawls content from 18 websites. Appendix A Table 5 lists the sources. Even though the authors claim that all websites allow use of content for research purposes, this is not be true:
- Hikaku Sitatter states that users must not republish material from Hikaku Sitatter (https://hikakusitatter.net/terms-of-service/).
- Marketing Charts' policy (https://www.marketingcharts.com/licensing) is that republishing individual charts is fine with attribution, but republishing collections of charts, which the authors are doing, requires contacting the website for licensing. Since the authors state that all websites allow use of charts for publication, I assume they most likely did not contact the website.
- Chit Chart's Terms and Conditions (https://chitchart.com/terms-and-conditions/) state that "reproduction is prohibited other than in accordance with the copyright notice" and explicitly require that users "may not distribute any part of this site without the trademark of Chit chart." For publications, the policy specifically states: "For more information on how to use a chart on any publications please contact us."
These are just some examples. I think an ethics review is probably necessary to ensure fair republishing of materials.

---

> ### Author Response · Authors · 2025-11-24
> **Response to Review (1/2)**
>
> Thank you for your thoughtful feedback.
> We have addressed the concerns regarding the layout template details, the quality of our QA set, the reliability of our metric, URL correctness, and ethical concerns about our dataset.
>
> > **Q1**: The authors say templates capture the spatial relation of chart elements, but how are templates represented? Is each template a D3 code snippet that can take in different data tables and images as input?
> >
> >
> **A1**:
>
> First, we would like to clarify that the layout templates capture the spatial relation of *infographic elements* (i.e., charts, images, and text), not only charts.
> We represent the layout template using the JSON format, which describes (1) the existence of elements (chart, image, and text), (2) their pairwise positional relationships, and (3) overlap relationships between their bounding boxes.
> A concrete example is shown below.
> Appendix F illustrates all 68 layout templates.
>
> ```
> {
>   # Element Existence
>   "title": "yes",     # "yes" or "no"
>   "image": "yes",
>   "chart": "yes",
>
>   # Position Relationships
>   "title-to-chart": "left-top",   # Top, bottom, and 7 other options
>   "image-to-chart": "right-top",
>   "title-to-image": "left",
>
>   # Overlap Relationships Between Bounding Boxes
>   "chart-title-overlap": "no",    # "yes" or "no"
>   "chart-image-overlap": "no",
>   "title-image-overlap": "no"
> }
> ```
>
> Second, we would like to clarify that the D3 code is used only to generate the chart elements.
>
> The layout templates do not contain any D3 code; instead, they are used to determine the final positions of generated elements (charts, images, and text) in the infographic.
>
> We clarify this in the layout optimization in Sec. 3.4.
>
> > **Q2**: Do you use a VLM to generate these templates?
> >
> **A2**: We do not use a VLM to generate these templates.
> Instead, as described in Sec. 3.3, they are derived from the analysis of layouts in 61,833 real infographic charts and summarized by the authors.
>
> > **Q3**: And how does layout optimization work with these templates?
> >
> **A3**:
>
> Given a selected layout template, we treat its spatial relationships as hard constraints during optimization. We first initialize element positions via rejection sampling, where we repeatedly sample random candidate layouts and keep only those that satisfy the spatial relationships specified by the template. This provides a structurally coherent starting point. We then refine the layout by adjusting element positions and sizes while preserving all template-specified constraints. In this way, the template effectively defines the feasible region over which the layout is optimized. We have clarified this in Sec. 3.4.
>
> > **Q4**: Huge improvements on questions based on ChartGalaxy is probably not that unsurprising given the vast majority of infographics in the QA set are based on synthetic charts with fixed templates. Finetuning is expected to help a lot here. Also I believe the QA set is not yet published so we do cannot gauge QA set quality.
> >
> >
> **A4**:
> First, we would like to clarify that we have released the QA set on HuggingFace for reproduction (link: <https://huggingface.co/datasets/ChartGalaxy/ChartGalaxy/blob/main/eval_data.zip>).
>
> Second, although fine-tuning on in-domain data can naturally bring gains, we further extend our evaluation on two public benchmarks that consist entirely of real infographics (i.e., InfographicVQA [1] and ChartQAPro [2]), as shown in Sec. 4.1.
> Results in Table 1 show that fine-tuning on ChartGalaxy consistently improves model performance: InternVL3-8B improves from 76.19 to 79.99 (**+3.80**) on InfographicVQA and 38.15 to 44.13 (**+5.98**) on ChartQAPro, while Qwen2.5-VL-7B improves from 78.59 to 83.03 (**+4.44**) and 37.97 to 41.56 (**+3.59**).
> These gains on the public benchmarks with different chart sources, styles, and questions indicate that ChartGalaxy provides transferable knowledge, rather than merely teaching models to memorize specific synthetic templates.
>
> Finally, regarding your concern about the QA set quality, we carefully design the construction process to ensure high QA quality.
>
> The questions not only cover the categories from previous benchmarks but also extend beyond purely text-based questions to include visual-element-based reasoning and visual style questions (see Appendix H.1 for detailed taxonomy).
> To ensure reliability, all QA pairs were manually verified by three co-authors.

---

> ### Author Response · Authors · 2025-11-24
> **Response to Review (2/2)**
>
> > **Q5**: The code generation metric seems a bit questionable...
> >
> >
> **A5**: To evaluate **the reliability of our metric**, we add human evaluation with three participants to assess consistency between our metric and human preferences.
> We randomly selected 600 pairs of charts generated by different LLMs.
> For each pair, participants were asked to choose the chart with higher overall similarity to the ground-truth chart without knowing which model produced which chart.
> We then compared the human choices with the preferences predicted by our metric (i.e., which chart received a higher overall score).
> The results demonstrate a high agreement rate of 92.33\% and a Cohen's kappa of 0.847 (almost perfect agreement), indicating strong alignment between our metric and human judgment.
> Details are provided in Appendix H.2.
>
> > **Q6**: All Chit Chart charts seem to not wrong urls. They all have https://chitchart.com/ without further specification (e.g., 00060283, 00059496). Could the authors check if this is the case please?
> >
> >
> **A6**: Upon review, we found that the image URLs were mistakenly set to the page URL.
> For example, the image URL for 00060283 was set to <https://chitchart.com/>, whereas the correct image URL should be <https://chitchart.com/sites/default/files/youthobesity.jpeg>.
> We have corrected these image URLs, and verified the entire dataset to ensure all such errors are fixed.
>
> > **Q7**: The authors say "Moreover, we construct an independent, human-verified evaluation set containing 2,176 synthetic charts with 4,975 question-answer pairs." Are these a sampled and verified subset of all VLM-generated questions?
> >
> >
> **A7**: To clarify, our dataset includes both VLM-generated and template-based questions, as detailed in Appendix I.1.
> The evaluation set is a subset sampled from this entire dataset.
> Specifically, we strategically sampled the data to ensure comprehensive coverage of all chart types, layouts, and question types.
> Furthermore, to guarantee high quality, all 2,176 charts and 4,975 QA pairs in this evaluation set were manually verified by the co-authors.
>
> > **Q8**: The paper crawls content from 18 websites. Appendix A Table 5 lists the sources. Even though the authors claim that all websites allow use of content for research purposes, this is not be true...
> >
> **A8**:
> We fully agree with the importance of properly handling copyright.
>
> Before releasing our dataset, we consulted a professional Intellectual Property (IP) attorney, who reviewed the terms and conditions of all 18 websites in our corpus.
> Her assessment is that our dataset, in its current form, **does not violate** the rights of these websites.
>
> The strictest restrictions in these terms apply to:
>
> - **Republish / Redistribute**: hosting and disseminating the actual content (e.g., image files).
>
> - **Reproduce**: creating and distributing copies of the original work.
>
> Our dataset only contains URLs pointing to the original sources. We do not host, copy, or redistribute chart images themselves.
> Therefore, according to the legal review, our dataset does not constitute republishing, redistribution, or reproduction, and is not subject to the corresponding restrictions.
> When users employ this dataset, they can download these images from the shared URLs, under the condition of complying with the copyright usage terms of the respective websites.
>
> Below, we clarify the specific examples mentioned by the reviewer:
>
> 1. **Hikaku Sitatter**:
> As shown in Table 5 in Appendix A, we use <hikaku-sitatter.com>, not <hikakusitatter.net> cited in the comment.
> These are different websites with different terms.
> The former explicitly allows use for "not-for-profit purposes" (<https://hikaku-sitatter.com/en/about/>), which aligns with our non-commercial research use.
> Again, we only store URLs and do not republish any chart images.
>
> 2.  **Marketing Charts**:
> The policy cited by the reviewer concerns republishing chart collections.
> In our dataset, we do not republish or redistribute any chart images.
> Instead, we only link to the original pages.
> Thus, the restriction on republishing collections does not apply to our usage, as confirmed by our IP attorney.
>
> 3. **Chit Chart**:
> Chit Chart's terms allow sharing with attribution (e.g., stating that the content is copyrighted by Chit Chart).
> The restrictions quoted by the reviewer apply to the reproduction and republishing of the charts themselves.
> Since our dataset does not reproduce or host any chart images and only lists source URLs with appropriate attribution, these restrictions do not apply to our dataset, according to the legal review.
>
> ### References
>
> [1] M. Mathew, et al., "InfographicVQA," WACV, 2022.
>
> [2] A. Masry, et al., "ChartQAPro: A more diverse and challenging benchmark for chart question answering," Arxiv, 2025.

---

### Author Response · Authors · 2025-12-02
**Summary of Reviews and Author Rebuttal**

Dear Reviewers, ACs, and SACs,

We sincerely thank you for the constructive feedback and thoughtful suggestions.
We are encouraged by the recognition of the scale, quality, diversity, and value of our dataset, as well as its comprehensive and convincing evaluation and effective generation pipeline.
Overall, there is a general agreement on the following strengths:

**The scale, quality, and diversity of the dataset**:
All reviewers agreed on the large scale, high quality, and diversity of ChartGalaxy.

- R\_6nvx: "The dataset size is very large, and it seems to span many chart types and designs."

- R\_6nvx: "The dataset quality seems to be high."

- R\_Z9By: "This work provides a large-scale dataset in the infographic domain."

- R\_Z9By: "Templates are carefully extracted ... resulting in a diverse set of samples."

- R\_NBfT: "The dataset includes over 1.76 million infographic charts ... high visual and structural diversity ... The dual-source construction (real + synthetic) provides both authenticity and scalability, addressing prior dataset limitations."

- R\_d34b: "*The paper addresses a clear and important gap on infographic data.* The community has largely focused on plain charts, while complex infographics ... remain a major challenge for LVLMs and no large-scale high-quality data exists."

**Comprehensive and convincing evaluation**:
All the reviewers found our evaluation to be comprehensive and convincing.

- R\_6nvx: "The use cases are in general convincing."

- R\_Z9By: "A large number of evaluation experiments are conducted in a thorough and detailed manner."

- R\_NBfT: "Clear evidence of downstream utility. Fine-tuning LVLMs ... showing measurable improvement ... The code generation benchmark ... positioning the dataset as an evaluation standard ... The example-based generation ... showing statistically significant improvements ..."

- R\_d34b: "All three applications are well-conceived and executed through chart understanding, code generation, and chart generation."

**The value/usefulness of the dataset**:
Three reviewers agreed on the value/usefulness of ChartGalaxy.

- R\_Z9By: "... where available data has been scarce until now."

- R\_NBfT: "This scope enables training LVLMs for realistic infographic scenarios, supporting broad generalization."

- R\_d34b: "The paper addresses a clear and important gap on infographic data."

- R\_d34b: "The authors do not just present a dataset; they thoroughly demonstrate its utility."

**Effective infographic generation pipeline**:
The reviewers found our generation pipeline effective and well-defined.

- R\_NBfT: "Well-defined inductive synthesis pipeline ... The pipeline's iterative detection and clustering stages ensure diversity ... The layout optimization step ... ensure high visual quality. The combination of data-to-chart mapping rules, adaptive sampling for variation balance, and semantically resonant color selection enhances the realism of synthetic charts."

- R\_d34b: "The pipeline for generating synthetic data is a major strength. The authors incorporate D3, which offers much more visualization options ... The use of template-based generation approach is quite clever and neat."
-------
We also deeply appreciate the insightful comments of each reviewer and have carefully addressed them. Below, we summarize the main review concerns and our responses/solutions.

**Additional quantitative/qualitative analysis**:

- Conducted an ablation experiment to compare the effectiveness of ChartGalaxy and prior datasets in fine-tuning (R\_d34b).

- Conducted human evaluation to assess consistency between our code generation metric and human preferences (R\_6nvx).

- Conducted an ablation experiment to assess the contribution of four key modules in our infographic chart generation pipeline (R\_NBfT).

- Evaluated the object detection model on 1,500 real infographic charts (R\_d34b).

**Clarifications**:

- Clarified that we **do not violate** the rights of the 18 websites based on the review of their terms and conditions by a professional Intellectual Property attorney (R\_6nvx).

- Clarified the unique reasoning challenges posed by infographic charts compared to plain charts (R\_NBfT).

- Clarified how layout templates are represented, as well as how they are utilized in our infographic chart generation pipeline (R\_6nvx, R\_Z9By, R\_NBfT).

- Clarified the design and quality control of our QA dataset, including coverage of chart types, layouts, and question types, along with rigorous manual verification (R\_6nvx).

- Explained the rationales for the task design and synthetic data selection in our code generation benchmark (R\_Z9By).

- Clarified the settings of our method and the baseline in example-based generation (R\_Z9By, R\_NBfT).

We are confident that these additional analyses and clarifications will effectively address the reviewers' concerns.
We deeply appreciate your careful review, constructive feedback, and valuable time.

---

### Meta-Review · Program_Chairs · 2026-01-07

**Summary:**

The paper proposes ChartGalaxy, a million-scale dataset designed to advance the understanding and generation of infographic charts. The paper has showcased that the proposed dataset can benefit multiple tasks, such as finetuning VLMs for infographic understanding, benchmarking code generation, etc.

Initial reviews are consistently positive. There are also several concerns pointed out by the reviewers, which have been addressed during the rebuttal phase. Therefore, I would recommend acceptance of this work. I encourage the authors to incorporate reviewers' suggestions in their next version.

**This paper is conditionally accepted provided the authors do the following for the camera-ready **:
[Ethics] Authors must include evidence indicating that they are respecting copyright and terms of service, providing details of the legal opinion described in the rebuttal.

**Reviewer Concerns:**

Most of the major concerns are addressed by the rebuttal.

**Reviewer Scores:**

6, 6, 8, 6 -> 6, 6, 8, 6

---

> ### Public Comment · ~Zhen_Li37 · 2026-03-01
> **Response to Meta Review**
>
> Thank you for the conditional acceptance of our paper.
> In the camera-ready revision, we have specifically addressed the requested ethics update by **providing additional details of the legal opinion referenced in our rebuttal**.
> We have made the following updates in our paper.
>
> First, we expand the Ethics Statement (Item 1, "Real infographic charts").
> We clarify that our release for real infographic charts is URL-only and does not include, host, cache, or redistribute any chart/image files.
> In addition, we report the legal assessment by a licensed IP attorney, based on a review of the sources’ licensing/usage pages and our release format.
> We further note that any downstream access to content via these URLs must comply with each source’s terms.
>
> Second, we expand Appendix A to include supporting details of the legal review.
> Appendix A now (i) links each of the 18 chart-rich sources to its publicly available terms/licensing page, and (ii) summarizes the attorney review scope and the corresponding source-specific conditions, grouped by key requirement types.

---

### Decision · Program_Chairs · 2026-01-26

**Decision:**

Accept (Poster)

**Comment:**

Conditions for acceptance have been satisfied.